# LEARNING TO SEGMENT FOR VEHICLE ROUTING PROBLEMS

**Wenbin Ouyang, Sirui Li, Yining Ma,**[*] **Cathy Wu**
MIT
`{oywenbin, siruil, yiningma, cathywu}@mit.edu`

## ABSTRACT

Iterative heuristics are widely recognized as state-of-the-art for Vehicle Routing Problems (VRPs). In this work, we exploit a critical observation: a large portion of the solution remains stable, i.e., unchanged across search iterations, causing redundant computations, especially for large-scale VRPs with long subtours. To address this, we pioneer the formal study of the First-Segment-Then-Aggregate (FSTA) decomposition technique to accelerate iterative solvers. FSTA preserves stable solution segments during the search, aggregates nodes within each segment into fixed hypernodes, and focuses the search only on unstable portions. Yet, a key challenge lies in identifying which segments should be aggregated. To this end, we introduce Learning-to-Segment (L2Seg), a novel neural framework to intelligently differentiate potentially stable and unstable portions for FSTA decomposition. We present three L2Seg variants: non-autoregressive (globally comprehensive but locally indiscriminate), autoregressive (locally refined but globally deficient), and their synergy. Empirical results on CVRP and VRPTW show that L2Seg accelerates state-of-the-art solvers by 2x to 7x. We further provide in-depth analysis showing why synergy achieves the best performance. Notably, L2Seg is compatible with traditional, learning-based, and hybrid solvers, while supporting various VRPs.

## 1 INTRODUCTION

Vehicle Routing Problems (VRPs) have profound applications such as in logistics and ride-hailing, driving advances in combinatorial optimization (Laporte, 2009). As NP-hard problems, they are typically tackled with heuristics approximately. Neural Combinatorial Optimization (NCO) (Kool et al., 2018; Bengio et al., 2021; Luo et al., 2025; Berto et al., 2023) has recently introduced machine learning into VRP solving, enabling data-driven decision-making with minimal domain knowledge while matching and even surpassing the performance of meticulously designed heuristics such as Lin-Kernighan-Helsgaun (LKH)(Helsgaun, 2017) and Hybrid Genetic Search (HGS)(Vidal, 2022).

Generally, state-of-the-art VRP solvers predominantly rely on iterative search to refine solutions through local search (e.g., ruin and repair). However, as noted in Section 3, a significant portion of edges *stabilize*[1] , or their presence in the solution stops changing between iterations, as the search progresses, despite repeated local search. For example, inner edges of neighboring subtours may remain fixed while only boundary edges undergo frequent combinatorial changes. Intuitively, such stability can be inferred from customer spatial distribution and the solution properties through end-to-end learning. Yet, existing solvers overlook such opportunities, leading to redundant computations that hinder their scalability and efficiency, especially in large-scale VRPs with long subtours.

Motivated by this critical observation, we study how learning to identify such *segments* can accelerate iterative search solvers, a perspective yet to be explored to the best of our knowledge. To this end, we formalize a **First-Segment-Then-Aggregate (FSTA)** decomposition framework, which identifies stable segments in a VRP solution and then aggregates them as fixed (one or two) hypernodes with combined attributes (e.g., total demand, min/max time windows). This not only decomposes the

---

[*]Corresponding author.

[1]Specifically, we define *stable edges* as those that consistently remain in the solution across consecutive solver invocations, where each invocation performs a full optimization round (multiple local search operations) within a fixed budget to return a locally optimal solution (see Appendix A.1 for formal definitions).

original large problem into more tractable subproblems but also significantly accelerates the search by leveraging iterative local search to strategically focus on unstable portions. We further show that FSTA preserves solution equivalence and is broadly applicable to VRPs with diverse constraints.

To identify unstable portions for FSTA decomposition, we then introduce **Learning-to-Segment (L2Seg)**, a novel learning-guided framework that leverages deep models to intelligently differentiate potentially stable and unstable portions, allowing dynamic decomposition for accelerated local search. Realizing this, however, is nontrivial: it involves a large combinatorial decision space requiring accurate segment grouping, and demands modeling complex interdependencies among predicted edges, constraints, spatial distribution, solution structures, and both node and edge features.

To address these challenges, L2Seg proposes encoder-decoder-styled neural models. The encoder integrates graph-level and route-level features using attention and graph neural networks, generating node embeddings that guide edge re-optimization predictions. L2Seg offers three decoders: (1) L2Seg-NAR (Non-Autoregressive): which features one-shot fast global prediction; (2) L2Seg-AR (Autoregressive), which enjoys sequential dependency modeling for high-precision local predictions; and (3) L2Seg-SYN (Synergized), which balances the strengths of both NAR and AR. Notably, this represents a pioneering work that explores the joint decision-making between AR and NAR models in neural combinatorial optimization. Our L2Seg models are trained via a weighted cross-entropy loss on datasets labeled using a lookahead procedure: edge stability is classified based on whether its presence in the solution was changed during iterative re-optimization.

Extensive experiments on large-scale CVRPs and VRPTWs show that L2Seg accelerates backbone heuristics by 2x to 7x, enabling them to outperform state-of-the-art classic, neural, and hybrid baselines, while generalizing well across different customer distributions and problem sizes. Notably, L2Seg exhibits strong flexibility in enhancing various solvers, including the classic LKH-3 (Helsgaun, 2017) solver, other orthogonal Large Neighborhood Search (LNS) methods (Shaw, 1998), and learning-guided decomposition method Learning-to-Delegate (L2D) (Li et al., 2021). We further analyze the synergy between AR and NAR models, showing their combination achieves the best performance by integrating NAR's global comprehension with AR's local precision.

Our contributions are: (1) We make a critical yet underexplored insight that stable segments persist across search iterations in large-scale VRPs, causing redundant computations; (2) We formally study and theoretically prove the properties and applicabilities of First-Segment-Then-Aggregate (FSTA) for various VRPs; (3) We develop Learning-to-Segment (L2Seg), a learning-guided framework with bespoke network architecture, training, and inference for segment identification; (4) We propose autoregressive, non-autoregressive, and their synergistic deep models, pioneering the first-of-its-kind study in NCO; (5) L2Seg consistently accelerates state-of-the-art iterative VRP solvers by 2x to 7x, boosting both classic and learning-based solvers, including other decomposition frameworks.

## 2 RELATED WORKS

**VRP Solvers.** Classical VRP solvers include exact methods with guarantees (Baldacci et al., 2012) and practical heuristics (Helsgaun, 2017). Recently, machine learning has been applied to combinatorial optimization, either end-to-end (Kool et al., 2018; Kwon et al., 2020; Fang et al., 2024; Geisler et al., 2022; Gao et al., 2024; Drakulic et al., 2023; Wang et al., 2024; Min et al., 2023; Li et al., 2023a) or learning-guided to unite data-driven insights into human solvers (Li et al., 2021; Lu et al., 2023; Huang et al., 2024; 2023; Hottung et al., 2025). For VRPs, the former could yield competitive performance to classic methods (Drakulic et al., 2023; Luo et al., 2023), while the latter often achieve state-of-the-art performance (Zheng et al., 2024). Among these, most effective VRP solvers rely on iterative search, including classic heuristics such as HGS (Vidal, 2022), LNS (Shaw, 1998) and LKH (Helsgaun, 2017); neural solvers that learn local search (Ma et al., 2021; 2023; Kim et al., 2023; Hottung and Tierney, 2022; Ma et al., 2022); neural constructive solvers integrated with search components (Luo et al., 2023; Hottung et al., 2022; Kim et al., 2021; Sun and Yang, 2023; Chalumeau et al., 2023; Kim et al., 2024; Qiu et al., 2022); and hybrid learning-guided methods like L2D (Li et al., 2021). However, both handcrafted and neural iterative search solvers overlook the redundant computations identified in this paper, particularly in large-scale VRPs.

**Decomposition for Large-scale VRPs.** Scalability in VRP solvers often relies on effective decomposition that operates on solutions partially (Santini et al., 2023). This includes hand-crafted

heuristics, such as LNS (Shaw, 1998) and evolutionary algorithms (Helsgaun, 2017), as well as learning-based methods such as sub-tour grouping (Zong et al., 2022), problem variant reduction (Hou et al., 2023), action space decomposition (Drakulic et al., 2023; Luo et al., 2023; Zhou et al., 2025a), spatial-based decomposition (Zheng et al., 2024; Zhou et al., 2025b; Pan et al., 2025), and hypergraphs decomposition (Li et al., 2025; Fu et al., 2023). In fact, effective search space reduction is fundamental to modern optimization solvers, including LKH-3 (Helsgaun, 2017) (via $k$-nearest reduction), LNS (Shaw, 1998) (via bounded neighborhood reduction), and exact solvers like Gurobi (via branch-and-bound reduction). Different from those hand-crafted ways of reduction, we present FSTA and L2Seg, a learning-based decomposition framework that advances this line by introducing stability estimation to intelligently detect unstable edges and aggregate stable segments in a data-driven way. L2Seg introduces a fundamentally different decomposition paradigm compared to existing methods. Unlike LNS, which isolates a bounded local neighborhood (e.g., 3–5 sub-routes) to destroy and rebuild, L2Seg operates globally across the entire solution, breaking edges and aggregating segments throughout all sub-routes simultaneously without predefined neighborhood boundaries. Compared with ML-enhanced LNS methods like L2D (Li et al., 2021), which operate at the sub-route level (selecting which sub-routes to modify), L2Seg detects unstable edges both within and across sub-routes, enabling finer-grained, edge-level decomposition. Moreover, L2Seg is solver-agnostic and complementary: it can enhance most existing iterative methods such as LKH and LNS, e.g., FSTA can be applied on top of LNS to fix stable edges globally within its selected neighborhood (see Appendix C.1 for detailed comparisons with representative decomposition methods). While another related work (Morabit et al., 2024) explores segment stability for re-optimization in a specific dynamic CVRP setting, our work addresses a different problem, i.e., identifying stable segments across search steps to accelerate iterative solvers. And we formally analyze the solution equivalence of FSTA across broader VRP variants. Moreover, L2Seg uniquely designs and integrates three novel deep learning models (AR, NAR, and synergized) to guide FSTA decomposition during search.

**AR and NAR Models.** In NCO, NAR models make global predictions like edge heatmaps (Sun and Yang, 2023; Li et al., 2023b). However, they struggle to model complex interdependencies, particularly VRP constraints. In contrast, AR models make sequential predictions, e.g., node by node selection in construction solvers (e.g., Luo et al. (2023)). AR offers stronger modeling capacity but might overlook global structure. Recent NCO works combine AR and NAR models in divide-and-conquer frameworks, with NAR for problem splitting and AR for solving (Zheng et al., 2024; Hou et al., 2023; Ye et al., 2024). We are the first to leverage their complementary strengths for joint decision-making, enabling more effective identification of unstable segments in FSTA decomposition.

## 3 FIRST-SEGMENT-THEN-AGGREGATE (FSTA)

### 3.1 VEHICLE ROUTING PROBLEMS (VRPS)

VRPs aim to minimize total travel costs (often distance or travel time) while serving a set of customers under constraints. Formally, a VRP instance $P$ is defined on a graph $G = (V, E)$, where each node $x_i \in V$ represents a customer and each edge $e_{i,j} \in E$ represents traveling from $x_i$ to $x_j$ and is associated with a travel cost. For Capacitated VRP (CVRP), vehicles of capacity $C$ start and end at a depot node $x_0$. The sum of the demands $d_i$ on any route must not exceed $C$, and each customer should be served exactly once. For VRP with Time Windows (VRPTW), each customer is additionally associated with a service time $s_i$ and a time window $[t_i^l, t_i^r]$ within which service must begin. See Appendix A for the formal definitions of CVRP and VRPTW.

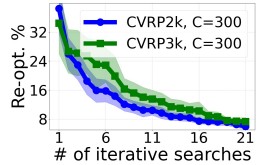

Figure 1: Percentage of re-optimized edges during iterative search using LKH-3 on 100 CVRP instances. Most edges remain unchanged, suggesting redundant calculations.

### 3.2 FSTA DECOMPOSITION

Figure 1 depicts that iterative search solvers perform ***redundant searches***, reoptimizing only a small portion while many edges remain unchanged, especially in large subtours with high capacity $C$. Inspired by Morabit et al. (2024), we formally study the decomposition technique, First-Segment-Then-Aggregate (FSTA), for accelerating iterative search solvers. As shown in the top of Figure 2, FSTA segments the VRP solutions by identifying unstable portions, and then groups the stable

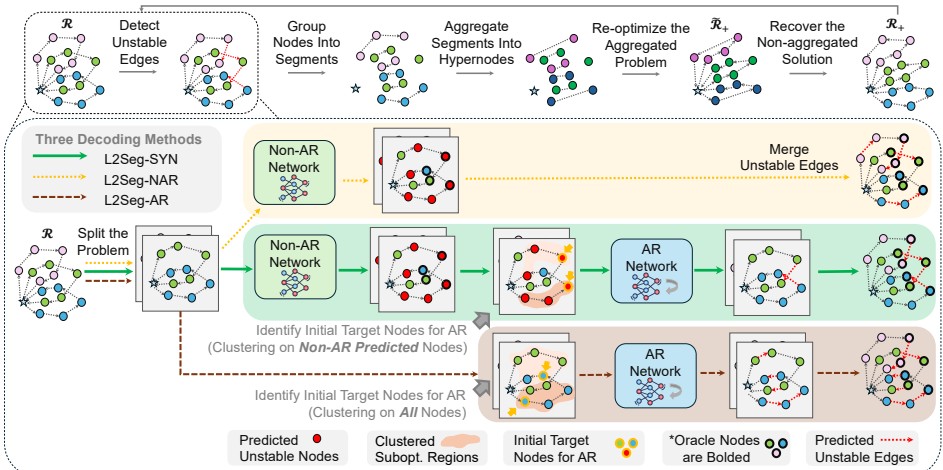

Figure 2: The overview of our FSTA decomposition framework (top) and the three proposed L2Seg models (bottom). L2Seg-SYN employs a four-step synergized approach: (1) problem decomposition into subproblems, (2) unstable nodes detection globally via NAR decoding, (3) clustering of NAR-predicted nodes to localize unstable regions and select initial target nodes, and (4) refining unstable edge predictions locally via AR decoding starting from these identified initial target nodes.

segments into hypernodes. We thus expect more efficient re-optimization on the reduced problems with smaller size. More visualization of FSTA is provided in Appendix B.1.

**Segment Definition**. Denote the solution (set of routes) of a CVRP as $\mathcal{R} = \{R^1, R^2, ...\}$, and each route as $R^i = (x_0 \rightarrow x_1^i \rightarrow x_2^i \rightarrow ... \rightarrow x_0) \in \mathcal{R}$, where the first and the last nodes in $R^i$ are the depot. A segment consists of some consecutive nodes within a route. We denote the segment containing the $j^{\text{th}}$ to $k^{\text{th}}$ nodes of route $i$ as $S_{j,k}^i = (x_j^i \rightarrow ... \rightarrow x_k^i)$. An aggregated segment $\tilde{S}_{j,k}^i$ uses one hypernode ($\tilde{S}_{j,k}^i = \{\tilde{x}_{j,k}^i\}$) or two hypernodes ($\tilde{S}_{j,k}^i = \{\tilde{x}_j^i, \tilde{x}_k^i\}$) with aggregated attributes (e.g. the demand of $\tilde{x}_{j,k}^i$ equals to $d_j^i + ... + d_k^i$) to represent the non-aggregated segment $S_{j,k}^i$.

**FSTA Solution Update**. After identifying unstable edges $\{e_{j_1}^i, e_{j_2}^i, ...\}$ in each route (which will be addressed in Section 4), where each $e_j^i$ denotes the edge starting from the $j^{\text{th}}$ node in route $R^i$, we break these edges and group the remaining stable edges into segments. To preserve a valid depot, edges connecting to the depot are included in the unstable edge set. After unstable edges are removed, each route $R^i$ is then decomposed into multiple disjoint segments $\{x_0, S_{1,j_1}^i, S_{j_1,j_2}^i, ...\}$, where $x_0$ is depot. Each segment $S_{j,k}^i$ is then aggregated into one or two hypernodes $\tilde{S}_{j,k}^i$, leading to a reduced problem $\tilde{P}$. We then obtain the corresponding solution $\tilde{\mathcal{R}}$ for such reduced problem, where for each $\tilde{R}^i \in \tilde{\mathcal{R}}$, we have $\tilde{R}^i = (x_0 \rightarrow \tilde{S}_{1,j_1}^i \rightarrow \tilde{S}_{j_1,j_2}^i ... \rightarrow x_0)$. With fewer nodes than the original problem $P$, re-optimization with a backbone solver becomes more efficient, which is analyzed and confirmed in Appendix B.1. After re-optimization, we obtain a new solution $\tilde{\mathcal{R}}_+$ for the reduced problem $\tilde{P}$, which is then recovered into a solution $\mathcal{R}_+$ for the original problem $P$ by expanding each hypernode(s) back into its original segment of nodes. This relies on our monotonicity theorem, which guarantees that an improved solution in $\tilde{P}$ maps to an improved solution in $P$.

**Theoretical Analysis.** We establish a theorem proving FSTA's feasibility and monotonicity across multiple VRP variants (e.g. CVRP, VRPTW, VRPB, and 1-VRPPD), with the proof in Appendix B.2.

**Theorem (Feasibility)** If the aggregated solution $\tilde{\mathcal{R}}_+$ is feasible to the aggregated problem, then $\mathcal{R}_+$ is also feasible to the original, non-aggregated problem.

**Theorem (Monotonicity).** If two feasible aggregated solutions $\tilde{\mathcal{R}}_+^1$ and $\tilde{\mathcal{R}}_+^2$ satisfy $f(\tilde{\mathcal{R}}_+^1) \leq f(\tilde{\mathcal{R}}_+^2)$, where $f(\cdot)$ denotes the objective function (total travel cost), their corresponding original solutions also preserve this order: $f(\mathcal{R}_+^1) \leq f(\mathcal{R}_+^2)$.

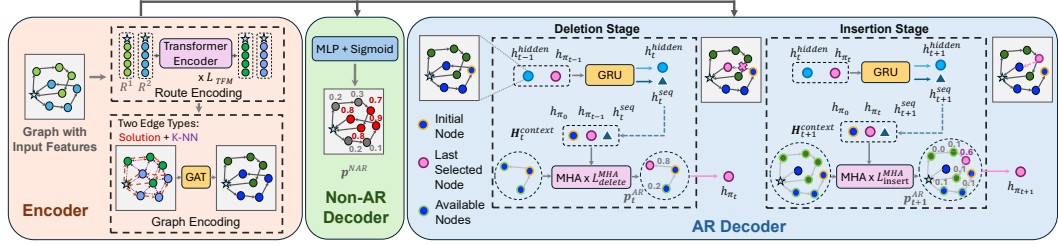

Figure 3: Architecture of L2Seg: encoder (left), NAR decoder (center), and AR decoder (right). NAR predicts unstable nodes for associated edges. AR uses a two-stage process, where the insertion bridges the deletion stage to accurately detect unstable edges locally, akin to the local search behavior.

## 4   LEARNING TO SEGMENT (L2SEG)

We introduce **Learning to Segment (L2Seg)**, a neural framework for predicting unstable edges to guide FSTA. We consider two paradigms: 1) Non-autoregressive (NAR) and 2) Autoregressive (AR) models. *NAR models* offer global predictions with an efficient single forward pass. However, they lack conditional modeling to accurately capture local dependencies. For example, when one edge is unstable, nearby edges often show instability but not all, but NAR models may fail to distinguish them and mark all neighboring edges as unstable. On the other hand, *AR models* can more natively capture local dependencies. Yet, they may miss the crucial global structure. For example, when unstable edges are distributed across distant regions, AR models may struggle to recognize and model these broader patterns. Our approach offers three variants as shown in Figure 2: non-autoregressive (L2Seg-NAR), autoregressive (L2Seg-AR), and a synergized combination of both (L2Seg-SYN).

### 4.1   NEURAL ARCHITECTURE

The autoregressive and non-autoregressive models of L2Seg share the same encoder structure. Next, we first describe the encoder, and then the two decoder architectures.

**Input Feature Design.** We propose enhanced input features for L2Seg to better distinguish unstable and stable edges (see Appendix B.1 for intuitions). Key features include node angularity relative to the depot and node internality, where the latter measures the proportion of nearest nodes within the same route. We consider two edge types: edges in the current solution $\mathcal{R}$ and edges connecting each node to their k-nearest neighbors. Appendix C.2 provides a detailed feature description.

**Encoder.** Given node features $\mathbf{X} = (\mathbf{x_0}, \mathbf{x_1}, \dots)$ and edge features $\mathbf{E} = \{\mathbf{e}_{0,1}, \mathbf{e}_{0,2}, \dots\}$, we compute the initial node embedding as $\mathbf{h}_i^{\text{init}} = \text{Concat}(\mathbf{h}_i^{\text{MLP}}, \mathbf{h}_i^{\text{POS}}) \in \mathbb{R}^{2d_h}$, where $\mathbf{h}_i^{\text{MLP}}$ and $\mathbf{h}_i^{\text{POS}}$ are obtained by passing $\mathbf{x_i}$ through a multilayer perceptron (MLP) and an absolute position encoder (Vaswani, 2017), respectively. Next, we process the embeddings using $L_{\text{TFM}}$ Transformer layers (Vaswani, 2017) with masks to prevent computation between nodes in different routes: $\mathbf{h}_i^{\text{TFM}} = \text{TFM}\left(\mathbf{h}_i^{\text{init}}\right) \in \mathbb{R}^{d_h}$. This step encodes local structural information from the current solution. Finally, we compute the node embeddings $\mathbf{H}^{\text{GNN}} = \{\mathbf{h}_i^{\text{GNN}} \in \mathbb{R}^{d_h} \mid i = 0, \dots, |V|\}$ leveraging the global graph information by using $L_{\text{GNN}}$ layers of a Graph Attention Network (GAT) (Veličković et al., 2017), where $\mathbf{H}^{\text{GNN}} = \text{GNN}\left(\mathbf{H}^{\text{TFM}}, \mathbf{E}\right)$.

**Non-Autoregressive Decoder.** It uses an MLP with a sigmoid function to decode the probability $\mathbf{p}^{\text{NAR}}$ of each node being unstable globally in one shot, so as to identify associated unstable edges:

$$\mathbf{p}^{\text{NAR}} = \text{MLP}_{\text{NAR}}\left(\mathbf{H}^{\text{GNN}}\right) \tag{1}$$

**Autoregressive Decoder.** The autoregressive decoder models unstable edge interdependence by generating them sequentially as $a = \{x_{\pi_0}, x_{\pi_1}, \dots\}$. Following classical local search where $k$ removed edges are reconnected via $k$ new insertions (Funke et al., 2005), the sequence alternates between deletion (identifying unstable edges) and insertion (introducing pseudo-edges that bridge to the next unstable edge), terminating at $x_{\text{end}}$. Note that the "insertion" stage is designed to model dependencies between consecutive unstable edges rather than actually "insert" edges into the solution. Formally, denote the set of edges within the current solution as $E_{\mathcal{R}}$. The decoding alternates between:

(1) **Deletion** ($t = 2k$): Selects an unstable edge $e_{\pi_{2k}, \pi_{2k+1}} \in E_{\mathcal{R}}$ based on a target node, which is either initialized at the first step (see Section 4.3) or the one obtained from the previous insertion step; one of the two edges connected to this node in the current solution is then selected as unstable (more than two candidates may exist if the node is the depot); importantly, this edge is only marked as unstable, not immediately removed from the solution; and (2) **Insertion** ($t = 2k + 1$): Selects an new edge $e_{\pi_{2k+1}, \pi_{2k+2}} \notin E_{\mathcal{R}}$ that links to the endpoint of the last unstable edge removed, exploring $O(|V|)$ potential candidates to serve as a bridge to the next unstable target node (next unstable region). From $a$, we then identify the set of removed edges as the unstable edges, i.e., $E_{\text{unstable}} = \{e_{\pi_0, \pi_1}, e_{\pi_2, \pi_3}, \dots\}$. Both stages employ two principal modules: Gated Recurrent Units (GRUs) (Chung et al., 2014) to encode sequence context, and multi-head attention (MHA) (Vaswani, 2017) for node selection. The GRU's initial hidden state is the average of all node embeddings: $\mathbf{h}_0^{\text{hidden}} = \frac{1}{|V|} \sum_{i=0}^{|V|} \mathbf{h}_i^{\text{GNN}}$. At step $t$, the sequence embedding is updated by $\mathbf{h}_t^{\text{seq}} = \text{GRU}\big(\mathbf{h}_{t-1}^{\text{hidden}}, \mathbf{h}_{\pi_{t-1}}^{\text{GNN}}\big)$, and the context embedding is formed by concatenating the embeddings of the initial node, the previous node, and the new sequence embedding: $\mathbf{H}_t^{\text{context}} = \text{Concat}\big(\mathbf{h}_{\pi_0}^{\text{GNN}}, \mathbf{h}_{\pi_{t-1}}^{\text{GNN}}, \mathbf{h}_t^{\text{seq}}\big)$.

Inspired by the decoder design in LEHD (Luo et al., 2023), we use two distinct MHA modules with $L^{\text{MHA}}$ layers, to decode $x_{\pi_t}$. Specifically, considering the size of the action space (at most 2 for deletion and $O(|V|)$ for insertion), we utilize a shallow decoder ($L_{\text{delete}}^{\text{MHA}} = 1$) during the deletion stage and a deeper decoder ($L_{\text{insert}}^{\text{MHA}} = 4$) during the insertion stage. Let $\mathbf{H}_t^a \subseteq \mathbf{H}^{\text{GNN}}$ denote the set of available nodes at step $t$. During the insertion stage, we also incorporate an additional candidate $\mathbf{h}^{\text{end}} = \alpha \mathbf{h}_{\pi_0}^{\text{GNN}} + (1 - \alpha) \frac{1}{|V|} \sum_{i=0}^{|V|} \mathbf{h}_i^{\text{GNN}}$, where $\alpha$ is a learnable parameter, to indicate termination of decoding, providing the AR model flexibility to determine the number of unstable edges. Formally, the decoding at step $t$ is given as follows; note that the first 3 dimensions of $\mathbf{H}^{(L^{\text{MHA}})}$ corresponds to context embeddings $\mathbf{H}_t^{\text{context}}$ and hence are masked from selection:

$$
\begin{aligned}
\mathbf{H}^{(0)} &= \text{Concat}\big(\mathbf{H}_t^{\text{context}}, \mathbf{H}_t^a\big), \\
\mathbf{H}^{(l)} &= \text{MHA}\big(\mathbf{H}^{(l-1)}\big), \\
u_i &= \begin{cases} (W_q \mathbf{h}^c)^T W_k \mathbf{h}_i^{(L^{\text{MHA}})} / \sqrt{d_h}, & \text{if } i > 3, \\ -\infty, & \text{O.W.}, \end{cases}
\end{aligned} \tag{2}
$$

where $1 \leq l \leq L^{\text{MHA}}$, $W_q$ and $W_k$ are learnable matrices, and $\mathbf{h}^c \in \mathbb{R}^{6d_h}$ concatenates the first three columns of $\mathbf{H}^{(0)}$ and $\mathbf{H}^{(L^{\text{MHA}})}$ along the last axis. The node $x_{\pi_t}$ is sampled from $\mathbf{p}_t^{\text{AR}} = \text{softmax}(\mathbf{u})$.

## 4.2 TRAINING

We employ iterative solvers as look-ahead heuristics to detect unstable edges. We utilize imitation learning to train L2Seg models to replicate the behavior of the look-ahead heuristics.

**Dataset Construction.** Let the edges in $\mathcal{R}$ be $E_{\mathcal{R}}$, and nodes indicated by edge set $E$ be $V_E$. Given $P$ with current solution $\mathcal{R}$, we first employ an iterative solver $\mathcal{S}$ to refine $\mathcal{R}$ and obtain $\mathcal{R}_+$. We then collect differing edges as $\mathcal{R}$ and $\mathcal{R}_+$ as $E_{\text{diff}} = \big(E_{\mathcal{R}} \setminus E_{\mathcal{R}_+}\big) \cup \big(E_{\mathcal{R}_+} \setminus E_{\mathcal{R}}\big)$ (including both the deleted and newly inserted edges). Next, we identify the set of unstable nodes $V_{\text{unstable}} = V_{E_{\text{diff}}}$, i.e., the set of nodes that are end points to some edge in $E_{\text{diff}}$. We empirically observe that solution refinement typically takes place between two adjacent routes. **For the NAR model,** we construct a dataset with binary labels. Each problem-label pair consists of a decomposed problem containing two adjacent routes and binary labels indicating whether each node is unstable (1) or stable (0). Formally, a node $x$ is labeled 1 if $x \in V_{\text{unstable}}$. **For the AR model,** we construct labels as node sequences preserving local dependencies among unstable

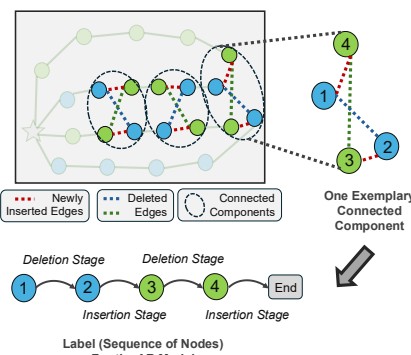

Figure 4: Training data construction for the AR model. Re-optimization reveals deleted edges (blue/green dashed) and inserted edges (red dashed) forming connected components (circles). For each component, depth-first search generates node sequences alternating between deletion and insertion stages, terminated by an end token as the AR model's training label.

edges. Nodes without local dependencies are naturally excluded through connected component partitioning. We obtain connected components $\mathcal{K}$ induced by $E_{\text{diff}}$ and select those spanning at most two routes, denoted $\mathcal{K}_{\text{TR}}$. For each $K \in \mathcal{K}_{\text{TR}}$ containing nodes from routes $R_i$ and $R_j$, we form a subproblem $P_K$ with solution $\mathcal{R}_K = \{R_i, R_j\}$. From each component $K$ (dashed circles in Figure 4), we extract a node sequence $y_K = \{x_{\pi_0}, x_{\pi_1}, \ldots, x_{\pi_m}, x_{\text{end}}\}$ by alternating between edge deletion and insertion operations (shown in Figure 4, second row). These problem-label pairs $(P_K, y_K)$ constitute **the AR model** training data.

**Loss Function.** To balance labels, we use weighted binary cross-entropy for the NAR model ($w_{\text{pos}} > 1$) and weighted cross-entropy for the AR model to balance the two stages ($w_{\text{insert}} > w_{\text{delete}}$).

$$L_{\text{NAR}}(\mathbf{p}^{\text{NAR}}, y^{ij}) = - \sum_{y_{x_k} \in y^{ij}} w_{\text{pos}}\, y_{x_k}\, \log\big(p_k^{\text{NAR}}\big) + \big(1 - y_{x_k}\big) \log\big(1 - p_k^{\text{NAR}}\big)$$

$$L_{\text{AR}}(\mathbf{p}^{\text{AR}}, y_K) = - \sum_{x_{\pi_{2k}} \in y_K} w_{\text{insert}}\, \log\big(p_{\pi_{2k}}^{\text{AR}}\big) \quad - \sum_{x_{\pi_{2k+1}} \in y_K} w_{\text{delect}}\, \log\big(p_{\pi_{2k+1}}^{\text{AR}}\big).$$

### 4.3 Inference

We describe the synergized inference that combines the benefits of global structural awareness from NAR with the local precision from AR, followed by two variants using only NAR or AR.

**Synergized Prediction (L2Seg-SYN).** L2Seg-SYN's inference pipeline for detecting unstable edges consists of four steps: (1) problem decomposition, (2) global unstable node detection via NAR decoding, (3) representative initial node identification for AR decoding based on NAR predictions, and (4) local unstable edge detection using AR decoding.

Given a problem $P$ with solution $\mathcal{R}$, we partition $P$ into approximately $|\mathcal{R}|$ subproblems, $\mathcal{P}_{\text{TR}}$, by grouping nodes from all two adjacent sub-tour pairs. For each subproblem in $\mathcal{P}_{\text{TR}}$, the NAR model predicts unstable nodes as $\hat{y}_{\text{NAR}} = \{x_i \mid p_i^{\text{NAR}} \geq \eta\}$, where $\eta$ is a predefined threshold. We then refine unstable edge detection with the AR model within regions identified by the NAR prediction. To reduce redundant decoding efforts on neighboring unstable nodes, we first group unstable nodes into $n_{\text{KMEANS}}$ clusters using the $K$-means algorithm, and select the node with the highest $p_i^{\text{NAR}}$ within each cluster as the starting point for AR decoding. The AR model then detects unstable edges based on these initial nodes. Finally, we aggregate unstable edges from all subproblems in $\mathcal{P}_{\text{TR}}$ as the final unstable edge set for $P$ given the current solution $\mathcal{R}$.

**Non-Autoregressive Prediction (L2Seg-NAR).** L2Seg-NAR uses only the NAR model for predictions. It identifies unstable nodes and marks all connected edges as unstable.

**Autoregressive Prediction (L2Seg-AR).** L2Seg-AR exclusively uses the AR model. Instead of using the NAR model, it assumes all nodes may be unstable, applying the $K$-means algorithm on all nodes. It then selects the node closest to each cluster center as the initial node for AR-based decoding.

## 5 Experiment

Our decomposition-based FSTA and L2Seg excel on large-scale problems. In this section, we first evaluate how L2Seg-AR, L2Seg-NAR, and L2Seg-SYN accelerate various learning and non-learning iterative solvers on large-capacity CVRPs with long subtours. Next, we compare L2Seg against state-of-the-art baselines on standard benchmark CVRP and VRPTW instances. Finally, we provide in-depth analyses of our pipeline. Additional results on CVRPLib benchmarks, clustered CVRP, heterogeneous-demand CVRP, a case study, and further discussions are presented in Appendix E.

**Backbone Solvers.** We apply L2Seg to three representative backbones: LKH-3 (Helsgaun, 2017) (*classic heuristic*), LNS (Shaw, 1998) (*decomposition framework*), and L2D (Li et al., 2021) (*learning-guided hybrid solvers, or machine-learning enhanced LNS*) to demonstrates the broad applicability. See Appendix D.1 for details.

**Baselines.** We include state-of-the-art classic solvers (LKH-3 (Helsgaun, 2017), HGS (Vidal, 2022)), neural solvers (BQ (Drakulic et al., 2023), LEHD (Luo et al., 2023), ELG (Gao et al., 2024), ICAM (Zhou et al., 2024), L2R (Zhou et al., 2025a), SIL (Luo et al., 2025; 2024)), and learning-based divide-and-conquer methods (GLOP (Ye et al., 2024), TAM (Hou et al., 2023), UDC (Zheng et al.,

Table 1: Performance comparisons of our proposed L2Seg-NAR, L2Seg-AR, and L2Seg-SYN when accelerating three backbone solvers, LKH-3, LNS, and L2D, on the *large-capacity* CVRP instances. We report the objective value, improvement gain (%), and the time. The gains (the higher the better) are w.r.t. the performance of each backbone solver. Time limits were set to be 150s for CVRP2k and 240s for CVRP5k, respectively.

| Methods | CVRP2k | | | CVRP5k | | |
|---|---|---|---|---|---|---|
| | Obj.↓ | Gain↑ | Time↓ | Obj.↓ | Gain↑ | Time↓ |
| LKH-3 (Helsgaun, 2017) | 45.24 | 0.00% | 152s | 65.34 | 0.00% | 242s |
| L2Seg-NAR-LKH-3 | 44.34 | 1.99% | 158s | 64.72 | 0.95% | 246s |
| L2Seg-AR-LKH-3 | 44.23 | 2.23% | 151s | 64.67 | 1.03% | 244s |
| **L2Seg-SYN-LKH-3** | **43.92** | **2.92%** | **152s** | **64.12** | **1.87%** | **248s** |
| LNS (Shaw, 1998) | 44.92 | 0.00% | 154s | 64.69 | 0.00% | 246s |
| L2Seg-NAR-LNS | 44.12 | 1.78% | 154s | 64.38 | 0.48% | 244s |
| L2Seg-AR-LNS | 44.02 | 2.00% | 157s | 64.24 | 0.70% | 249s |
| **L2Seg-SYN-LNS** | **43.42** | **3.34%** | **152s** | **63.94** | **1.16%** | **241s** |
| L2D (Li et al., 2021) | 43.69 | 0.00% | 153s | 64.21 | 0.00% | 243s |
| L2Seg-NAR-L2D | 43.55 | 0.32% | 152s | 64.02 | 0.30% | 243s |
| L2Seg-AR-L2D | 43.53 | 0.37% | 156s | 64.12 | 0.14% | 247s |
| **L2Seg-SYN-L2D** | **43.35** | **0.78%** | **157s** | **63.89** | **0.50%** | **248s** |

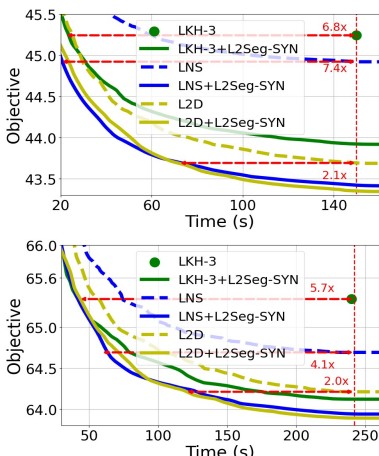

Figure 5: Search curves for L2Seg on three backbone solvers on large capacity CVRP2k (upper) and CVRP5k (lower). L2Seg achieves up to 7x speedups.

2024), L2D (Li et al., 2021), NDS (Hottung et al., 2025)). We rerun LKH-3, LNS, L2D, SIL and NDS and report results from Zheng et al. (2024); Zhou et al. (2025a); Luo et al. (2024) for other baselines using the same benchmarks. See Appendix D.2 for baseline setup details and Appendix D.3 for L2Seg hyperparameters.

**Data Distribution.** We generate all training and test instances following prior works Zheng et al. (2024) for CVRP and Solomon (1987) for VRPTW. See Appendix D.4 for details. For Section 5.1, results are averaged over 100 large-scale CVRP test instances at 2k and 5k scales (capacities 500 and 1,000, respectively). For Section 5.2, we follow standard NCO benchmarks, reporting averaged results on 1k, 2k, and 5k test datasets with 1,000 CVRP and 100 VRPTW instances per scale.

**Evaluation and Metric.** We impose time limits of 150s, 240s, and 300s for CVRP1k, 2k, and 5k, and 120s, 240s, and 600s for VRPTW1k, 2k and 5k, where each solver may finish a few seconds ($< 10$s) beyond its limit. We set $\eta = 0.6$ and $n_{\text{KMEANS}} = 3$ for our L2Seg. We report averaged cost and per-instance solve time for all cases, and report percentage improvements over backbone in Section 5.1 and gaps to HGS (the best heuristic solvers) for both CVRP and VRPTW in Section 5.2.

## 5.1 L2SEG ACCELERATES VARIOUS ITERATIVE BACKBONE SOLVERS

We first verify the effectiveness of the three L2Seg variants to enhance backbone solvers. Table 1 presents results on large-capacity, uniformly distributed CVRPs with long subtours. All L2Seg variants consistently improve each backbone across all problem scales. Also, performance gains are larger for weaker backbones. While L2Seg-AR and L2Seg-NAR each boost performance, their combination (L2Seg-SYN) delivers the best solutions. Figure 5 plots average objective curves over time, which reveal 2x to 7x speedups on the backbone solvers with L2Seg-SYN. Remarkably, L2Seg-augmentation lets weaker solvers surpass stronger ones (e.g., LKH-3 + L2Seg-SYN outperforms vanilla LNS).

## 5.2 L2SEG OUTPERFORMS CLASSIC AND NEURAL BASELINES ON CVRP AND VRPTW

We evaluate the highest-performing L2Seg-SYN implementation with three distinct backbone solvers and compare against state-of-the-art classical and neural approaches. As demonstrated in Table 2, L2Seg surpasses both classical and neural baselines on CVRP and VRPTW benchmarks. For CVRP, L2Seg achieves superior performance within comparable computational time relative to competitive classical solvers, including HGS on larger problem instances. It also outperforms the state-of-the-art learning-based constructive solver SIL (Luo et al., 2025; 2024) and divide-and-conquer solver L2D (Li et al., 2021) across all problem scales. For VRPTW, L2Seg exceeds all classical and learning-based solvers across various scales under identical time constraints, with performance advantages increasing

Table 2: Performance comparisons of our L2Seg-SYN-L2D against baselines on *benchmark* CVRP and VRPTW instances. The gap % (lower the better) is w.r.t. the performance of HGS.

| Methods | CVRP1k | | | CVRP2k | | | CVRP5k | | |
|---|---|---|---|---|---|---|---|---|---|
| | Obj.↓ | Gap↓ | Time↓ | Obj.↓ | Gap↓ | Time↓ | Obj.↓ | Gap↓ | Time↓ |
| HGS (Vidal, 2022) | 41.20 | 0.00% | 5m | 57.20 | 0.00% | 5m | 126.20 | 0.00% | 5m |
| LKH-3 (Helsgaun, 2017) | 42.98 | 4.32% | 6.6m | 57.94 | 1.29% | 11.4m | 175.70 | 39.22% | 2.5m |
| LNS (Shaw, 1998) | 42.44 | 3.01% | 2.5m | 57.62 | 0.73% | 4.0m | 126.58 | 0.30% | 5.0m |
| BQ (Drakulic et al., 2023) | 44.17 | 7.21% | 55s | 62.59 | 9.42% | 3m | 139.80 | 10.78% | 45m |
| LEHD (Luo et al., 2023) | 43.96 | 6.70% | 1.3m | 61.58 | 7.66% | 9.5m | 138.20 | 9.51% | 3h |
| ELG (Gao et al., 2024) | 43.58 | 5.78% | 15.6m | - | - | - | - | - | - |
| ICAM (Zhou et al., 2024) | 43.07 | 4.54% | 26s | 61.34 | 7.24% | 3.7m | 136.90 | 8.48% | 50m |
| L2R (Zhou et al., 2025a) | 44.20 | 7.28% | 34.2s | - | - | - | 131.10 | 3.88% | 1.8m |
| SIL (Luo et al., 2025; 2024) | 42.00 | 1.94% | 1.5m | 57.10 | -0.17% | 3.0m | 123.25 | -2.33% | 6.8m |
| TAM(LKH-3) (Hou et al., 2023) | 46.30 | 12.38% | 4m | 64.80 | 13.29% | 9.6m | 144.60 | 14.58% | 35m |
| GLOP-G(LKH-3) (Ye et al., 2024) | 45.90 | 11.41% | 2m | 63.02 | 10.52% | 2.5m | 140.40 | 11.25% | 8m |
| UDC (Zheng et al., 2024) | 43.00 | 4.37% | 1.2h | 60.01 | 4.9% | 2.15h | 136.70 | 8.32% | 16m |
| L2D (Li et al., 2021) | 42.07 | 2.11% | 2.5m | 57.44 | 0.42% | 4.2m | 126.48 | 0.22% | 5.3m |
| NDS (Hottung et al., 2025) | **41.16** | **-0.01%** | 2.5m | 56.11 | -1.91% | 4m | - | - | - |
| L2Seg-SYN-LKH-3 | 41.42 | 0.53% | 2.5m | 56.37 | -1.45% | 4.4m | 122.34 | -3.16% | 5.1m |
| L2Seg-SYN-LNS | 41.36 | 0.39% | 2.5m | 56.08 | -1.96% | 4.1m | 121.96 | -3.48% | 5.1m |
| **L2Seg-SYN-L2D** | 41.23 | 0.07% | 2.5m | **56.05** | **-2.01%** | 4.1m | **121.87** | **-3.55%** | 5.1m |
| Methods | VRPTW1k | | | VRPTW2k | | | VRPTW5k | | |
| | Obj.↓ | Gap↓ | Time↓ | Obj.↓ | Gap↓ | Time↓ | Obj.↓ | Gap↓ | Time↓ |
| HGS (Vidal, 2022) | 90.35 | 0.00% | 2m | 173.46 | 0.00% | 4m | 344.2 | 0.00% | 10m |
| LKH-3 (Helsgaun, 2017) | 91.32 | 1.07% | 2m | 174.25 | 0.46% | 4m | 353.2 | 2.61% | 10m |
| LNS (Shaw, 1998) | 88.12 | -2.47% | 2m | 165.42 | -4.64% | 4m | 338.5 | -1.66% | 10m |
| L2D (Li et al., 2021) | 88.01 | -2.59% | 2m | 164.12 | -5.38% | 4m | 335.2 | -2.61% | 10m |
| NDS (Hottung et al., 2025) | 87.54 | -3.11% | 2m | 167.48 | -3.45% | 4m | - | - | - |
| L2Seg-SYN-LKH-3 | 88.65 | -1.88% | 2m | 169.24 | -2.43% | 4m | 345.2 | 0.29% | 10m |
| L2Seg-SYN-LNS | 87.31 | -3.36% | 2m | 163.94 | -5.49% | 4m | 334.1 | -2.93% | 10m |
| **L2Seg-SYN-L2D** | **87.25** | **-3.43%** | 2m | **163.74** | **-5.60%** | 4m | **333.4** | **-3.14%** | 10m |

Table 3: Performance of L2Seg-SYN v.s. Random FSTA to accelerate LNS on CVRP instances.

| Methods | LNS (Backbone) | Random FSTA (40%) | Random FSTA (60%) | L2Seg-SYN w/o Enhanced Features | **L2Seg-SYN** |
|---|---|---|---|---|---|
| CVRP2k | 44.92 | 46.24 | 46.89 | 43.65 | **43.42** |
| CVRP5k | 64.69 | 66.72 | 65.92 | 64.22 | **63.94** |

as problem size grows. Notably, L2Seg consistently enhances performance when integrated with any backbone solver, demonstrating its versatility. Additional analyses are provided in Appendix E.

## 5.3 L2Seg Performs and Generalized Well on More Realistic CVRP

To demonstrate L2Seg's robustness beyond uniform distributions, we evaluate both in-distribution and zero-shot generalization on instances with clustered customers and heterogeneous demands—patterns common in real-world logistics. As shown in Table 12, L2Seg maintains strong performance across all settings: zero-shot transfer achieves 0.82%-3.10% improvements over LNS, while in-distribution models reach 1.02%-3.54% gains. These consistent improvements across diverse distributions validate L2Seg's practical applicability. See Appendix E.3 for details.

## 5.4 Further Analysis and Discussions

**Ablation Study.** Table 3 compares the LNS backbone; random FSTA with 40% and 60% of edges arbitrarily marked as unstable; L2Seg-SYN w/o enhanced features; and full L2Seg-SYN. Results show that Random FSTA worsens performance; and only full L2Seg-SYN with enhanced features achieves top performance. This confirms that L2Seg's learnable, feature-guided segmentation is indispensable for preserving high-quality segments in FSTA for boosting backbone solvers.

**High Recall or High TNR?** Higher Recall allows more unstable edges to be reoptimized, potentially improving performance, while higher TNR reduces problem size and runtime. However, due to learning imprecision, pursuing high TNRs often reduces Recall, causing premature convergence. Figure 7 shows that for L2Seg-SYN, fixing too few (left: high Recall, low TNR) or too many (right:

Table 4: Model prediction analysis of L2Seg-LNS on CVRP2k.

| Methods | Recall↑ | TNR↑ | Obj.↓ |
|---|---|---|---|
| L2Seg-SYN | 89.02% | **61.24%** | **43.42** |
| L2Seg-NAR | **91.46%** | 51.79% | 44.02 |
| L2Seg-AR | 74.39% | 54.07% | 44.12 |

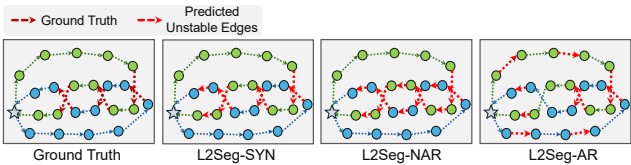

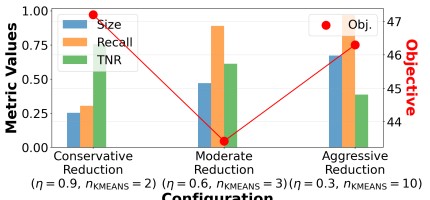

Figure 6: Illustration of L2Seg model behaviors.

high TNR, low Recall) degrades performance. Ours (middle) balances this tradeoff for optimal performance.

**Why NAR+AR Is the Best?** Figure 6 shows a conceptual illustration of the model's behaviour across L2Seg variants (See Appendix E.5 for a real case-study). L2Seg-NAR identifies unstable regions but over-classifies due to the lack of dependency modeling, while L2Seg-AR models dependencies but struggles with initial detection. L2Seg-SYN achieves the complementary synergy. Moreover, Table 4 further shows that L2Seg-SYN achieves the best balanced Recall and TNR for the best performance.

Figure 7: Statistic values of Size (reduced/original ratio), Recall, and TNR across three L2Seg-SYN configurations.

**The Structure of the Stability Labels.** We hypothesize that stability labels are composed of two factors: (1) Inherent Problem Structure (e.g., edges common to the majority of local optima regardless of the solver), and (2) Solver-Dependent Patterns (edges preferred due to specific search biases). We designed an experiment on CVRP1k to compare label similarity across different solvers. We generated labels using HGS (60s time limit) and LKH-3 (1000 local search steps), comparing them against a "ground truth" generated by LKH-3 with a much longer run (3000 steps). Each solver was run with 10 different seeds to remove randomness. Label similarity is defined as the percentage of overlapping stable edges. As shown in Table 5, these results reveal two key insights: (1) Even fundamentally different solvers (HGS vs. LKH-3) share 78.3% of stable edges. This indicates that the majority of the stability is contributed by inherent problem structure rather than solver artifacts; (2) More similar solvers produce more similar labels (LKH-3 variants: 85.1% similarity) compared to fundamentally different solvers (HGS vs LKH-3: 78.3%), suggesting some solver-specific patterns.

Table 5: Label similarity across different solvers

| | HGS | LKH-3 (1000 Local Search Step) | LKH-3 (3000 Local Search Step) |
|---|---|---|---|
| Label Similarity | 78.3% | 85.1% | 100% (by default) |

# 6 CONCLUSION

This work introduces Learning-to-Segment (L2Seg), a novel learning-guided framework that accelerates state-of-the-art iterative solvers for large-scale VRPs by 2x to 7x. We formalize the FSTA decomposition and employ a specialized encoder-decoder architecture to dynamically differentiate potentially unstable and stable segments in FSTA. L2Seg features three variants, L2Seg-NAR, L2Seg-AR, and L2Seg-SYN, pioneering the synergy of AR and NAR models in NCO. Extensive results demonstrate L2Seg's state-of-the-art performance on representative CVRP and VRPTW and flexibility in boosting classic and learning-based solvers, including other decomposition frameworks. One potential limitation is that L2Seg is not guaranteed to boost all VRP solvers across all VRP variants. Future work includes: (1) extending L2Seg to accelerate additional VRP solvers (e.g., Vidal (2022)); (2) applying L2Seg to more VRP variants and other combinatorial optimization problems; and (3) expanding the synergy between AR and NAR models to the broader NCO community. Our code is publicly available at `https://github.com/mit-wu-lab/learning-to-segment`.

ACKNOWLEDGMENT

This research was supported by a gift from Mathworks, as well as partial support from the MIT Amazon Science Hub, the National Science Foundation (NSF) award 2149548 and CAREER award 2239566, and an Amazon Robotics Fellowship. The authors acknowledge the MIT SuperCloud and Lincoln Laboratory Supercomputing Center for providing the high-performance computing resources that contributed to the research results reported in this paper. We particularly thank Andrea Lodi for the insightful discussions throughout this project, especially during its initial stages. We also thank Zhongxia "Zee" Yan, Jianan Zhou, and Samitha Samaranayake for their valuable input.

REPRODUCIBILITY STATEMENT

We provide comprehensive technical details in the appendices: architecture and input features (Appendix D.3), data generation (Appendix D.4), training procedures (Appendix C.4), and experimental setup (Section 5). The complete codebase, including code and pre-trained models, will be released on GitHub under the MIT License upon publication.

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

APPENDICES

CONTENTS

# A  SUPPLEMENTARY DEFINITIONS

## A.1  UNSTABLE EDGES AND STABLE EDGES

We define an **iterative step** $t$ as the $t$-th invocation of the backbone solver. Within each invocation, the solver performs a full round of optimization (involving multiple local search operations) subject to a fixed budget (e.g., time limit or number of steps) to return a locally optimal solution. Unstable edges refer to edges that need to be re-optimized during the iterative re-optimization procedure. We supplement the formal definitions as follows: given a solution $\mathcal{R}_t$ at iterative step $t$, an edge $e \in \mathcal{R}$ is unstable if $e \notin \mathcal{R}_{t+1}$. When we generate the labels for training, we use a lookahead backbone solver for detecting unstable edges. An edge is a stable edge if it's not an unstable edge.

## A.2  CAPACITATED VEHICLE ROUTING PROBLEM

Given a complete graph $G = (V, E)$ where $V = \{x_0, x_1, \ldots, x_n\}$ is the set of nodes with node $x_0$ representing the depot and nodes $x_1$ to $x_n$ representing customers. Each customer $i$ has a demand $d_i > 0$, and each edge $e_{i,j} \in E$ has an associated cost representing the travel distance or travel time between nodes $x_i$ and $x_j$. A fleet of homogeneous vehicles, each with capacity $C$, is available at the depot. The objective is to find a set of routes that minimizes the total travel cost, subject to: (i) each route starts and ends at the depot, (ii) each customer is visited exactly once, (iii) the total demand of customers on each route does not exceed vehicle capacity $C$.

## A.3  VEHICLE ROUTING PROBLEM WITH TIME WINDOWS

Given a complete graph $G = (V, E)$ where $V = \{x_0, x_1, \ldots, x_n\}$ is the set of nodes with node $x_0$ representing the depot and nodes $x_1$ to $x_n$ representing customers. Each customer $i$ has a demand $d_i > 0$, and each edge $e_{i,j} \in E$ has an associated cost representing the travel distance or travel time between nodes $x_i$ and $x_j$. Each customer $i$ has a time window $[t_i^l, t_i^r]$ where $t_i^l$ is the earliest arrival time and $t_i^r$ is the latest arrival time, and requires a service time $s_i$. A fleet of homogeneous vehicles, each with capacity $C$, is available at the depot. The objective is to find a set of routes that minimizes the total travel cost, subject to: (i) each route starts and ends at the depot, (ii) each customer is visited exactly once, (iii) the total demand of customers on each route does not exceed vehicle capacity $C$, (iv) service at each customer begins within their time window $[t_i^l, t_i^r]$.

# B  DETAILS OF FIRST-SEGMENT-THEN-AGGREGATE (FSTA)

## B.1  MORE DISCUSSIONS ON FSTA

### B.1.1  VISUALIZATION OF UNSTABLE EDGE PATTERNS

In this section, we provide visualization and analysis of unstable edge distribution patterns, which serve as foundational motivation for our L2Seg approach. We examine unstable edges on three randomly selected CVRP1k instances solved iteratively using LKH-3. In these visualizations, red dashed lines represent unstable edges, and yellow stars indicate depot locations.

Our visualization reveals two key observations: (1) The number of unstable edges generally decreases as optimization progresses, with more and more edges remaining unchanged between iterations; (2) Edges at route boundaries exhibit higher stability, while unstable edges predominantly concentrate within route interiors. Despite these discernible spatial patterns, no simple heuristic rule appears sufficient to reliably predict unstable edges, as they can be distributed across the start, middle, and end segments of each tour. This complexity motivates our development of L2Seg, a learning-based method designed to capture these intricate patterns more effectively.

### B.1.2  VISUALIZATION OF APPLYING FSTA ON ONE CVRP INSTANCE

To provide a concrete illustration of our FSTA methodology, we present an example of its application to CVRP in Figure 9, which demonstrates the complete FSTA decomposition pipeline (detailed algorithmic specifications are provided in Appendix B.1.4). This example utilizes the lookahead oracle model for unstable edge identification (defined in Appendix B.1.1), employs LKH-3 as the

backbone optimization solver, and operates on a representative small-capacity CVRP1k instance to showcase the framework's efficacy. Red dashed lines indicate detected unstable edges, while blue dashed lines represent re-optimized edges. Note that dual hypernode aggregation substantially reduces the problem size compared to the original instance.

### B.1.3 ASSUMPTION VERIFICATION

Table 6: Oracle Performance on CVRP2k: Time to Reach L2Seg-SYN-LNS Solution Quality

|  | Oracle (LNS) + perfect recall & TNR | Oracle (LNS) + 95% recall & 95% TNR | Oracle (LNS) + 90% recall & 90% TNR | Oracle + 70% recall & 70% TNR | Ref (L2Seg-SYN-LNS) |
|---|---|---|---|---|---|
| **Obj.** | 56.02 | 56.01 | 56.02 | 56.04 | 56.08 |
| **Time** | 39s | 62s | 119s | 324s | 241s |

In Section 3, we hypothesized that effective problem reduction can substantially accelerate re-optimization. We empirically validate this by implementing a look-ahead oracle for unstable edge detection. The oracle performs a 1-step re-optimization using LKH-3 and identifies unstable edges $E_{\text{unstable}}$ as those differing between the original and re-optimized solutions. FSTA then constructs a reduced problem instance based on these oracle-identified edges, which is subsequently re-optimized using the LKH-3 backbone solver. As this is an oracle-based evaluation, the time required for look-ahead computation is excluded from performance measurements.

Table 6 reports the time required to achieve performance equivalent to our learned model on small-capacity CVRP2k instances. Beyond the perfect oracle scenario, we evaluate imperfect oracle

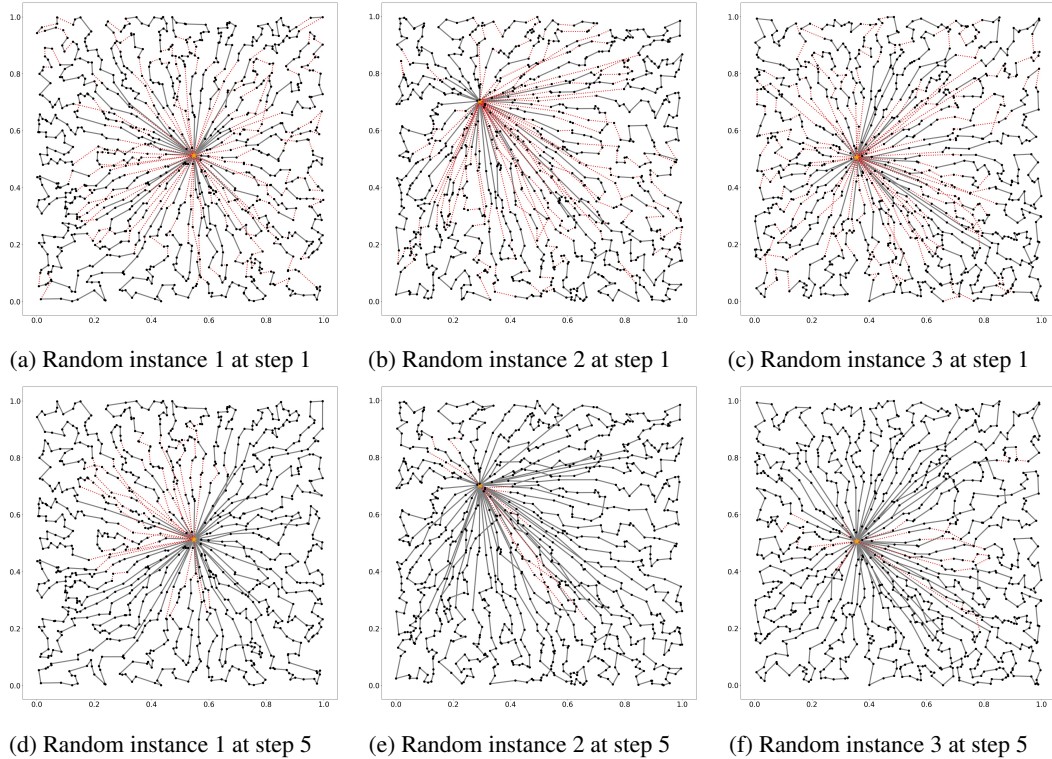

(a) Random instance 1 at step 1   (b) Random instance 2 at step 1   (c) Random instance 3 at step 1

(d) Random instance 1 at step 5   (e) Random instance 2 at step 5   (f) Random instance 3 at step 5

Figure 8: Spatial distribution of unstable edges (dashed red lines) across optimization iterations using LKH-3 solver. Results are presented for three randomly selected CVRP1k instances at iteatvie search steps 1 and 5. While many edges remain unchanged across iterations, unstable edges predominantly emerge within the interiors of routes. In contrast, edges located at route boundaries exhibit higher stability throughout the iterative optimization process.

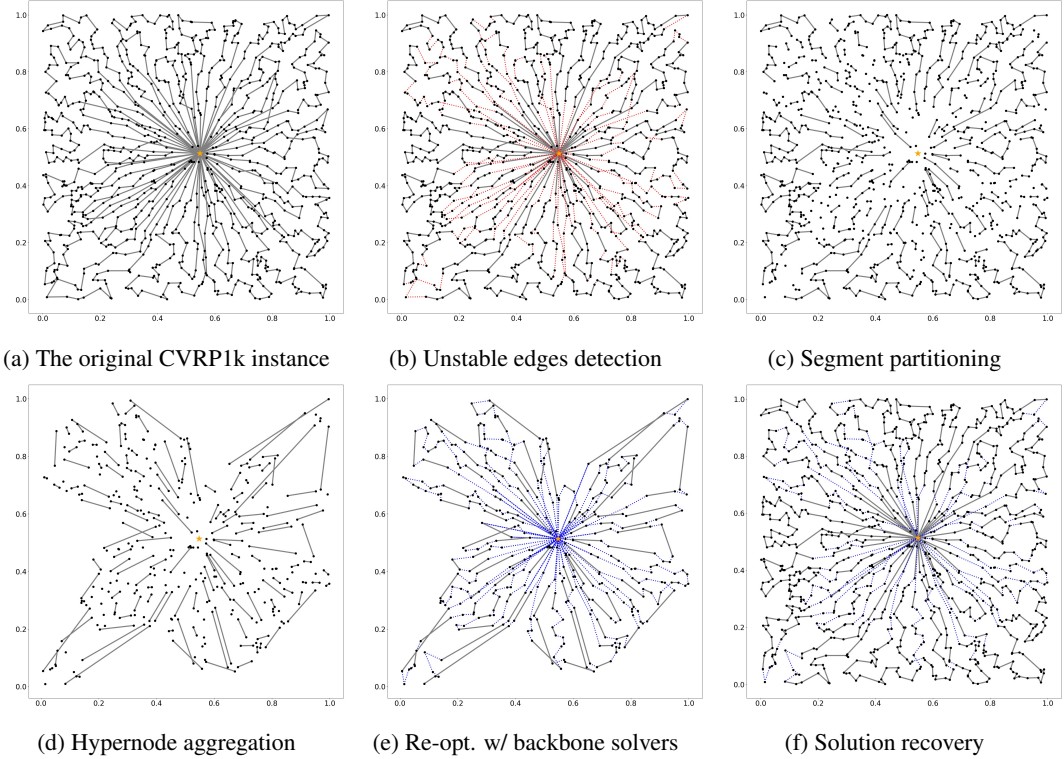

Figure 9: Illustration of our FSTA applied to one CVRP instance. Each FSTA step corresponds to the descriptions in Appendix B.1.4. Red dashed lines: unstable edges; blue dashed lines: re-optimized edges. Note that the subproblem (d) contains substantially fewer nodes than the original instance (a).

configurations where recall and true negative rates fall below 100%. The perfect oracle demonstrates substantially superior efficiency. Performance remains competitive under moderate imperfection levels; however, achieving recall and TNR as high as 90% without oracle access is highly non-trivial. In more practical scenarios, where recall and TNR drop to 70%, the oracle-based approach is outperformed by our L2Seg-SYN-LNS, highlighting the effectiveness of our learned model.

These results provide evidence that accurate identification of unstable edges, coupled with appropriate FSTA-based problem reduction, enables significantly more efficient re-optimization.

### B.1.4 DETAILS OF FSTA DECOMPOSITION FRAMEWORK

In this section, we present the details of the FSTA decomposition framework. Given a routing problem $P$ and an initial solution $\mathcal{R}$, one iterative step of FSTA yields a potentially improved solution $\mathcal{R}_+$. The framework comprises five sequential steps (also illustrated in Algorithm 1 and Figure 2):

1. **Unstable Edges Detection:** We implement effective methods (e.g., our learning-based model L2Seg or random heuristics detailed in Section 5.4) to identify unstable edges $E_{\text{unstable}}$ and obtain the stable edge set $E_{\text{stable}} = E \setminus E_{\text{unstable}}$. This identification challenge is addressed by our L2Seg model, with full details provided in Section 4 and Appendix C.

2. **Segment Partitioning:** After removing unstable edges $E_{\text{unstable}}$, each route decomposes into multiple disjoint segments consisting of consecutive nodes connected by stable edges. Formally, we segment each route into $(x_0, S^i_{1,j_1}, S^i_{j_1,j_2}, ..., x_0) = (x_0, S^i_{(1)}, S^i_{(2)}, ..., x_0) \in R^i$, where $x_0$ is depot and we simplify the notation by using a single index for segments (note that a segment can consist only one single node).

3. **Hypernode Aggregation:** We aggregate each segment $S^i_{j,k}$ and represent it with either one hypernode ($\tilde{S}^i_{j,k} = \{\tilde{x}^i_{j,k}\}$) or two hypernodes ($\tilde{S}^i_{j,k} = \{\tilde{x}^i_j, \tilde{x}^i_k\}$) with aggregated attributes.

This transformation requires that (our feasibility theorem): (a) the reduced problem remains feasible, and (b) a solution in the aggregated problem can be mapped back to a feasible solution in the original problem. These transformations produce a reduced problem $\tilde{P}$ with corresponding solution $\tilde{\mathcal{R}}$.

4. **Re-optimization with Backbone Solvers:** We invoke a backbone solver to improve solution $\tilde{\mathcal{R}}$, yielding an enhanced solution $\tilde{\mathcal{R}}_+$. While theoretically any solver could serve as the backbone solver, practical acceleration requires solvers capable of effectively leveraging existing solutions (e.g., LKH-3 (Helsgaun, 2017)).

5. **Solution Recovery:** With the improved solution $\tilde{\mathcal{R}}_+$ for the reduced problem $\tilde{P}$, we recover a corresponding solution $\mathcal{R}_+$ for the original problem $P$ by expanding each hypernode back into its original segment of nodes. This step relies on our monotonicity theorem, which guarantees that an improved solution in $\tilde{P}$ maps to an improved solution in $P$.

**Selection of Hypernode Aggregation Strategies.** We analyze the trade-offs between single and dual hypernode aggregation strategies: (1) *Dual hypernode aggregation* enables bidirectional segment traversal, potentially improving re-optimization efficiency by expanding the solution search space. However, this approach requires enforcing inclusion of the connecting edge between hypernodes, adding algorithmic complexity. (2) *Single hypernode aggregation* achieves superior problem size reduction but constrains segment traversal to a fixed direction, thereby restricting the re-optimization search space and potentially limiting performance improvements. Additionally, single hypernode aggregation transforms symmetric routing problems into asymmetric variants, which may compromise the efficiency of existing backbone solvers that are typically optimized for symmetric instances.

**Selection of Backbone Solvers.** Our framework is generic to be applied to most existing VRP heuristics by design. In practice, acceleration within our framework requires solvers that can effectively utilize initial solutions as warm starts. Furthermore, if the dual hypernode aggregation is used, the backbone solver needs to fix certain edges during local search. Our framework is readily compatible with a variety of solvers without modifying their source codes, including LKH-3 (Helsgaun, 2017), decomposition-based solvers like LNS (Shaw, 1998), and learning-based methods such as L2D (Li et al., 2021). Incorporating additional solvers such as HGS (Vidal, 2022), would involve extending its current code to accept initial solutions as input, which we leave as future work. Notably, as demonstrated in Section 5, our L2Seg-augmented approach with relatively weaker backbone solvers outperforms HGS in multiple CVRP and VRPTW benchmark scenarios.

**Applicability to Routing Variants.** FSTA is broadly applicable to routing problem variants that support feasible hypernode aggregation and solution recovery, as ensured by the feasibility and monotonicity conditions established in Section 3. In Appendix B.2, we formally prove that many routing variants meet these conditions, demonstrating the versatility of our L2Seg framework. Detailed implementation guidelines for applying hypernode aggregation across different routing variants are provided in Appendix B.1.5.

---

**Algorithm 1:** Iteratively Re-optimize Routing Problems with FSTA

---

**Input:** Routing problem $P$, initial solution $\mathcal{R}$, time limit $T_{TL}$, backbone solver BS, model M to identify unstable edges
**Output:** Improved solution $\mathcal{R}$

1 **while** *time limit $T_{TL}$ is not reached* **do**
2     $E_{\text{unstable}} \leftarrow \text{M}(P, \mathcal{R})$ ;         // Unstable Edges Detection
3     $\{S^i_{j,k}\} \leftarrow \text{GetSegments}(P, \mathcal{R}, E_{\text{unstable}})$ ;      // Segment Partitioning
4     Obtain $\{\tilde{S}^i_{j,k}\}$ and reduced problem $\tilde{P}$ with solution $\tilde{\mathcal{R}}$ ;  // Hypernode Aggregation
5     $\tilde{\mathcal{R}}_+ \leftarrow \text{BS}(\tilde{P}, \tilde{\mathcal{R}})$ ;       // Re-optimization with Backbone Solver
6     $\mathcal{R}_+ \leftarrow \text{RecoverSolution}(P, \tilde{P}, \tilde{\mathcal{R}}_+)$ ;       // Solution Recovery
7     $\mathcal{R} \leftarrow \mathcal{R}_+$ ;         // Update current solution
8 **end while**
9 **return** $\mathcal{R}$

---

B.1.5 APPLYING FSTA ON VARIOUS VRPs

In this section, we present the implementation details of FSTA across diverse routing variants, including the Capacitated Vehicle Routing Problem (CVRP), Vehicle Routing Problem with Time Windows (VRPTW), Vehicle Routing Problem with Backhauls (VRPB), and Single-Commodity Vehicle Routing Problem with Pickup and Delivery (1-VRPPD). Without loss of generality, we denote a segment to be aggregated as $S_{j,k} = (x_j \to \ldots \to x_k)$, and its corresponding hypernode representation as either $\tilde{S}_{j,k} = \{\tilde{x}\}$ (single hypernode) or $\tilde{S}_{j,k} = \{\tilde{x}_j, \tilde{x}_k\}$ (dual hypernodes). The implementation specifications are summarized in Table 7.

**CVRP.** We provide the formal definition of CVRP in Section 3. Each node in CVRP is characterized by location and demand attributes. For CVRP, we employ dual hypernode aggregation where location attributes are preserved as $\tilde{x}_j = x_j$ and $\tilde{x}_k = x_k$, while demand is equally distributed between hypernodes as $\tilde{d}_j = \tilde{d}_k = \frac{1}{2}(d_j + \cdots + d_k)$. We force the solver to include the edge connecting $\tilde{x}_j$ and $\tilde{x}_k$ in the solution.

**VRPTW.** We provide the formal definition of VRPTW in Section 3. In addition to location and demand attributes, VRPTW instances are characterized by time windows $[t^l, t^r]$ and service time $s$ for each node. For VRPTW, we employ adaptive strategies for hypernode aggregation based on temporal feasibility. We first compute the aggregated time windows $\bar{t}^l_j, \bar{t}^r_j$ and aggregated service time $\bar{s}_j$ using the following recursive formulation:

$$
\begin{aligned}
\bar{t}^l_m &= \begin{cases} t^l_k & \text{if } m = k \\ \max\{t^l_m, \bar{t}^l_{m+1} - (s_m + \text{dist}(x_m, x_{m+1}))\} & \text{if } j \le m \le k-1, \end{cases} \\
\bar{t}^r_m &= \begin{cases} t^r_k & \text{if } m = k \\ \min\{t^r_m, \bar{t}^r_{m+1} - (s_m + \text{dist}(x_m, x_{m+1}))\} & \text{if } j \le m \le k-1, \end{cases} \\
\bar{s}_m &= \begin{cases} s_k & \text{if } m = k \\ \bar{s}_{m+1} + s_m + \text{dist}(x_m, x_{m+1}) & \text{if } j \le m \le k-1, \end{cases}
\end{aligned}
\tag{3}
$$

where $[t^l_m, t^r_m]$ denotes the time window for node $x_m$, $s_m$ represents the service time at node $x_m$, and $\text{dist}(x_m, x_{m+1})$ is the travel time from node $x_m$ to node $x_{m+1}$.

If $\bar{t}^l_j \le \bar{t}^r_j$ (feasible time window), we employ single hypernode aggregation with: $\text{dist}(x_i, \tilde{x}) = \text{dist}(x_i, x_j)$, $\text{dist}(\tilde{x}, x_i) = \text{dist}(x_k, x_i)$, $\tilde{d} = d_j + \cdots + d_k$, $\tilde{t}^l = \bar{t}^l_j$, $\tilde{t}^r = \bar{t}^r_j$, and $\tilde{s} = \bar{s}_j$.

If $\bar{t}^l_j > \bar{t}^r_j$ (temporal infeasible time window), we employ dual hypernode aggregation with: $\tilde{x}_j = x_j$, $\tilde{x}_k = x_k$, $\tilde{d}_j = \tilde{d}_k = \frac{1}{2}(d_j + \cdots + d_k)$, time windows $\tilde{t}^l_j = 0$, $\tilde{t}^r_j = \bar{t}^r_j$, $\tilde{t}^l_k = \bar{t}^l_j$, $\tilde{t}^r_k = \infty$, and service times $\tilde{s}_j = 0$, $\tilde{s}_k = \bar{s}_j$. We additionally set $\text{dist}(\tilde{x}_j, \tilde{x}_k) = 0$ and enforce inclusion of the edge connecting $\tilde{x}_j$ and $\tilde{x}_k$ in the solution.

**VRPB.** Compared to the CVRP, the VRPB (Goetschalckx and Jacobs-Blecha, 1989) involves serving two types of customers: linehaul customers requiring deliveries from the depot and backhaul customers providing goods to be collected and returned to the depot. The primary constraint is that all linehaul customers must be visited before any backhaul customers on the same route, while ensuring vehicle capacity is never exceeded during either the delivery or pickup phases. We use $b_i \in \{0, 1\}$ to indicate whether node $i$ is a backhaul customer. For VRPB, we require the edge connecting to a linehaul customer and a backhaul customer included in the $E_{\text{unstable}}$. We employ single hypernode aggregation that $\text{dist}(x_i, \tilde{x}) = \text{dist}(x_i, x_j)$, $\text{dist}(\tilde{x}, x_i) = \text{dist}(x_k, x_i)$, $\tilde{d} = d_j + \cdots + d_k$, and $\tilde{b} = b_j$ (we require customer being the same type within each segment that $b_j = \ldots = b_k$).

**1-VRPPD.** Compared to the CVRP, the 1-VRPPD (Martinovic et al., 2008) deals with customers labeled as either cargo sink ($d_i < 0$) or cargo source ($d_i > 0$), depending on their pickup or delivery demand. Along the route of each vehicle, the vehicle could not load negative cargo or cargo exceeding the capacity of the vehicle $C$. For any segment $S_{j,k}$, we define $D^j = d_j$, $D^{j+1} = d_j + d_{j+1}$, ..., and $D^k = d_j + d_{j+1} + \ldots + d_k$. We further define $D^{\min} = \min\{0, D_j, D_{j+1}, \ldots\}$ and $D^{\max} = \max\{0, D_j, D_{j+1}, \ldots\}$. For 1-VRPPD, we require three hypernodes $\tilde{x}_j = x_j$, $\tilde{x}_{\text{mid}}$, and $\tilde{x}_k = x_k$, where the distances from $\tilde{x}_{\text{mid}}$ to $\tilde{x}_j$ or $\tilde{x}_k$ are 0, and infinity for the other hypernodes. For the aggregated demands, $\tilde{d}_j = D^{\min}$, $\tilde{d}_{\text{mid}} = D^{\max} - D^{\min}$, and $\tilde{d}_k = D^k - D^{\max} - D^{\min}$. Additional constraints are added to ensure the directed edges $\tilde{x}_j \to \tilde{x}_{\text{mid}} \to \tilde{x}_k$ are included in the solutions.

Table 7: Implementation specifications of FSTA hypernode aggregation for CVRP, VRPTW, VRPB variants. Refer to Equation 3 for the definitions of $\bar{s}_j$, $\bar{t}_j^l$ and $\bar{t}_j^r$.

| CVRP | | | | |
|---|---|---|---|---|
| **Type** | **Condition** | **Attribute** | **Aggregation** | **Additional Constraints / Settings** |
| Two Hypernodes | Always | Location/Distance | $\tilde{x}_j = x_j$ $\tilde{x}_k = x_k$ | Include edge $\tilde{x}_j \to \tilde{x}_k$ in the solution |
| | | Demand | $\tilde{d}_j = \tilde{d}_k = \frac{1}{2}(d_j + \cdots + d_k)$ | |

| VRPTW | | | | |
|---|---|---|---|---|
| **Type** | **Condition** | **Attribute** | **Aggregation** | **Additional Constraints / Settings** |
| One Hypernode | $\bar{t}_j^l \leq \bar{t}_j^r$ | Location/Distance | $\mathrm{dist}(x_i, \tilde{x}) = \mathrm{dist}(x_i, x_j)$, $\mathrm{dist}(\tilde{x}, x_i) = \mathrm{dist}(x_k, x_i)$ | None |
| | | Demand | $\tilde{d} = d_j + \cdots + d_k$ | |
| | | Service Time | $\tilde{s} = \bar{s}_j$ | |
| | | Time Windows | $\tilde{t}^l = \bar{t}_j^l, \tilde{t}^r = \bar{t}_j^r$ | |
| Two Hypernodes | $\bar{t}_j^l > \bar{t}_j^r$ | Location/Distance | $\tilde{x}_j = x_j, \tilde{x}_k = x_k$ | Include edge $\tilde{x}_j \to \tilde{x}_k$ in solution; set $\mathrm{dist}(\tilde{x}_j, \tilde{x}_k) = 0$ |
| | | Demand | $\tilde{d}_j = \tilde{d}_k = \frac{1}{2}(d_j + \cdots + d_k)$ | |
| | | Service Time | $\tilde{s}_j = 0, \tilde{s}_k = \bar{s}_j$ | |
| | | Time Windows | $\tilde{t}_j^l = 0, \tilde{t}_j^r = \bar{t}_j^r, \tilde{t}_k^l = \bar{t}_j^l, \tilde{t}_k^r = \infty$ | |

| VRPB | | | | |
|---|---|---|---|---|
| **Type** | **Condition** | **Attribute** | **Aggregation** | **Additional Constraints / Settings** |
| One Hypernode | Always | Location/Distance | $\mathrm{dist}(x_i, \tilde{x}) = \mathrm{dist}(x_i, x_j)$, $\mathrm{dist}(\tilde{x}, x_i) = \mathrm{dist}(x_k, x_i)$ | Require $b_j = \cdots = b_k$ (same customer type) during Unstable Edges Detection Stage |
| | | Demand | $\tilde{d} = d_j + \cdots + d_k$ | |
| | | Is backhaul | $\tilde{b} = b_j$ | |

| 1-VRPPD | | | | |
|---|---|---|---|---|
| **Type** | **Condition** | **Attribute** | **Aggregation** | **Additional Constraints / Settings** |
| Three Hypernodes | Always | Location/Distance | $\tilde{x}_j = x_j, \tilde{x}_k = x_k$ $\mathrm{dist}(\tilde{x}_j, \tilde{x}_{\mathrm{mid}}) = \mathrm{dist}(\tilde{x}_{\mathrm{mid}}, \tilde{x}_k) = 0$ $\tilde{x}_{\mathrm{mid}}$ only connects to $\tilde{x}_j$ and $\tilde{x}_k$ | Include edges $\tilde{x}_j \to \tilde{x}_{\mathrm{mid}} \to \tilde{x}_k$ in the solution |
| | | Demand | $\tilde{d}_j = D^{\min}, \tilde{d}_{\mathrm{mid}} = D^{\max} - D^{\min}$, $\tilde{d}_k = D^k - D^{\max} - D^{\min}$ | |

## B.2 Proof of FSTA

**Theorem.** *(Feasibility)* If the aggregated solution $\tilde{\mathcal{R}}_+$ is a feasible solution to the aggregated problem, then $\mathcal{R}_+$ is a feasible solution to the original, non-aggregated problem. *(Monotonicity)* Let $\tilde{\mathcal{R}}_+^1$ and $\tilde{\mathcal{R}}_+^2$ be two feasible solutions to the aggregated problem, with $f(\tilde{\mathcal{R}}_+^1) \leq f(\tilde{\mathcal{R}}_+^2)$, where $f(\cdot)$ denotes the objective function (total travel cost). Then, for the associated solution in the original space, we also have $f(\mathcal{R}_+^1) \leq f(\mathcal{R}_+^2)$.

***Proof Structure and Notation.*** Without loss of generality, we consider a single-route solution containing one segment $S_{j,k} = (x_j \to \cdots \to x_k)$ with more than one node, i.e., the solution $\mathcal{R}$ contains route $R = (x_0 \to x_1 \to \cdots \to S_{j,k} \to x_{k+1} \to \cdots \to x_0)$. We define the aggregated problem with node set $\tilde{V} = \{x_0\} \cup \{x_p\}_{p<j \text{ or } p>k} \cup \{\tilde{S}_{j,k}\}$, where nodes outside the segment retain their original representation, ensuring their feasibility by construction. Since we enforce the inclusion of the edge connecting $\tilde{x}_j$ and $\tilde{x}_k$ in dual hypernode aggregation within solution $\tilde{\mathcal{R}}_+$, the segment $\tilde{S}_{j,k}$ must be incorporated into some route $\tilde{R}_+^* \in \tilde{\mathcal{R}}_+$ for both hypernode aggregation strategies. We denote the improved route containing this segment after mapping back to the original problem as $R_+^*$.

We present the segment aggregation strategies for different routing variants below, followed by proofs of feasibility and monotonicity for the aggregation scheme. Note that the following analysis naturally extends to multi-route solutions with multiple segments per route.

### B.2.1 CVRP

***Aggregation Strategy (Two Hypernodes).*** The detailed implementation of FSTA on CVRP can be found in Appendix B.1.5 and Table 7. Notice that one single hypernode aggregation is also applicatable for CVRP, and $\tilde{d}_j, \tilde{d}_k$ could take other values as long as $\tilde{d}_j + \tilde{d}_k = d_j + \ldots + d_k$.

***Feasibility Proof [Capacity Constraint].*** Notice that since $\tilde{d}_j + \tilde{d}_k = d_j + ... + d_k$, we have:

$$
\begin{aligned}
\sum_{x_i \in \tilde{R}_+^*} d_i &= \sum_{x_i \in \tilde{R}_+^* \setminus \tilde{S}_{j,k}} d_i + \quad \tilde{d}_j + \tilde{d}_k \\
&= \sum_{x_i \in R_+^* \setminus S_{j,k}} d_i + \quad d_j + ... + d_k = \sum_{x_i \in R_+^*} d_i
\end{aligned}
\tag{4}
$$

Thus, we have:

$$
\sum_{x_i \in \tilde{R}_+^*} d_i \leq C \Rightarrow \sum_{x_i \in R_+^*} d_i \leq C
\tag{5}
$$

Then, we have a feasible $\tilde{\mathcal{R}}_+ \Rightarrow$ a feasible $\mathcal{R}_+$.

$\square$

***Monotonicity Proof.*** Notice that

$$
\begin{aligned}
f(\tilde{\mathcal{R}}_+) &= f(\tilde{\mathcal{R}}_+ \setminus \{\tilde{R}_+^*\}) + f(\{\tilde{R}_+^*\}) = f(\mathcal{R}_+ \setminus \{R_+^*\}) + f(\{\tilde{R}_+^*\}) \\
&= f(\mathcal{R}_+ \setminus \{R_+^*\}) + f(\{R_+^*\}) - \sum_{j \leq q < k} dist(x_q, x_{q+1}) \quad + dist(\tilde{x}_j, \tilde{x}_k) \\
&= f(\mathcal{R}_+) + \text{Const}|_{S_{j,k}}
\end{aligned}
\tag{6}
$$

where $\text{Const}|_{S_{j,k}}$ is a constant once the segment $S_{j,k}$ is decided. Therefore, we have:

$$
f(\tilde{\mathcal{R}}_+^1) \leq f(\tilde{\mathcal{R}}_+^2) \Rightarrow f(\mathcal{R}_+^1) + \text{Const}|_{S_{j,k}} \leq f(\mathcal{R}_+^2) + \text{Const}|_{S_{j,k}} \Rightarrow f(\mathcal{R}_+^1) \leq f(\mathcal{R}_+^2)
\tag{7}
$$

$\square$

We note that the feasibility proof for capacity constraint and the monotonicity proof could be easily extended to the single hypernodes aggregation.

### B.2.2 VRPTW

***Aggregation Strategy (Mixed Strategies).*** The detailed implementation of FSTA on VRPTW can be found in Appendix B.1.5 and Table 7. We denote $s_m^* = s_m + dist(x_m, x_{m+1})$ for $j \leq m < k$ and $s_k^* = s_k$. We further set the service time by $\tilde{s}_m = \sum_{m \leq q \leq k} s_q^*$, and we repeat the temporal time window $[\bar{t}_j^l, \bar{t}_j^r]$ (which could be infeasible) defined by the following recursive relationship:

$$
\begin{aligned}
\bar{t}_m^l &= \begin{cases} t_k^l & m = k \\ \max\{t_m^l, \bar{t}_{m+1}^l - s_m^*\} & j \leq m \leq k-1, \end{cases} \\
\bar{t}_m^r &= \begin{cases} t_k^r & m = k \\ \min\{t_m^r, \bar{t}_{m+1}^r - s_m^*\} & j \leq m \leq k-1, \end{cases}
\end{aligned}
\tag{8}
$$

where $[t_m^l, t_m^r]$ is the time window for a node $x_m$, $s_m$ is the service time at node $x_m$ and $dist(x_m, x_{m+1})$ is the time to travel from node $x_m$ to node $x_{m+1}$.

***Feasibility Proof [Time Window Constraint].*** We first prove for the condition that the temporal time window $[\bar{t}_j^l, \bar{t}_j^r]$ is feasible ($\bar{t}_j^l < \bar{t}_j^r$) and single hypernode aggregation is applied. Then, we extend to the infeasible temporal time window condition where dual hypernode aggregation is applied.

**Condition of Feasible Temporal Time Windows (One Hypernode).** We present an inductive proof based on the *segment length*. Given a feasible solution $\tilde{\mathcal{R}}_+$ for the aggregated problem, we show the following two conditions of the corresponding non-aggregated solution $\mathcal{R}_+$ to satisfy the time window constraint:

- *Condition (1): We visit each node $x_m$ before the end of its time window $t_m^r$.*
- *Condition (2): The total time we spent visiting the entire segment is the same in both aggregated and non-aggregated representations.*

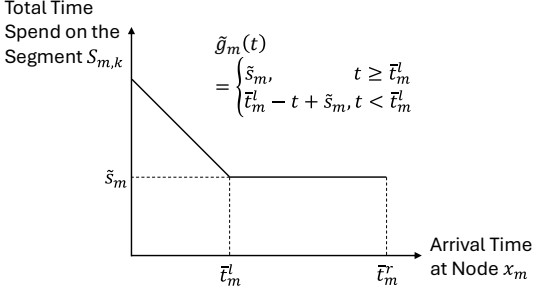 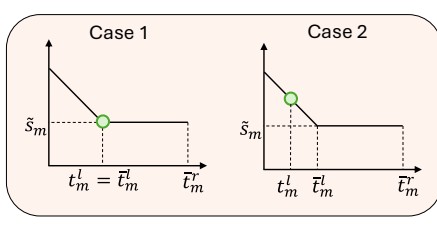

Figure 10: This illustration demonstrates the temporal dynamics of the aggregated segment. The left panel shows the time function characterized by a piecewise linear structure: initially decreasing with slope -1, then transitioning to a constant value corresponding to the aggregated left time window boundary. The right panel presents two distinct scenarios that characterize the relationship between the aggregated left time window ($\bar{t}_m^l$) and the individual non-aggregated left time windows ($t_m^l$).

*Proof of Condition (1):*

- *Base case (segment length = 1).* Suppose the segment $S_{k,k} = (x_k)$ contains a single node $x_k$. Then the aggregated problem is identical to the non-aggregated problem by construction, so condition (1) is trivially satisfied.

- *Inductive Step (segment length = $(k-m)+1 > 1$).* textit the aggregation of the segment $S_{m+1,k} = (x_{m+1} \to ... \to x_k)$ into $\tilde{S}_{m+1,k} = \{\tilde{x}_{m+1,k}\}$ satisfies condition (1). We want to show that the aggregation of the segment $S_{m,k} = (x_m \to ... \to x_k)$ into $\tilde{S}_{m,k} = \{\tilde{x}_{m,k}\}$ also satisfies condition (1).

  Since $\tilde{\mathcal{R}}_+$ is a feasible solution for the aggregated problem, we will visit the hypernode $\tilde{x}_{m,k}$ before the end of its time window $\bar{t}_m^r = \min\{t_m^r, \bar{t}_{m+1}^r - s_m^*\}$. Corresponding, in the associated non-aggregated solution, we visit the node $x_m$ before its time limit $t_m^r$, hence satisfying condition (1) for the node $x_m$. Furthermore, in the associated non-aggregated solution, we visit the next node $x_{m+1}$ before time $\bar{t}_m^r + s_m^* \leq \bar{t}_{m+1}^r$. Based on the inductive hypothesis, condition (1) holds for the rest of the segment $(x_{m+1} \to ... \to x_k)$ if we arrive at node $x_{m+1}$ before its end time. Hence, condition (1) holds for the whole segment $S_{m,k} = (x_m \to x_{m+1} \to ... \to x_k)$.

*Proof of Condition (2):* For all $m$, suppose we arrive at the hypernode $\tilde{x}_{m,k}$ at time $t \leq \bar{t}_m^r$ in the aggregated solution. By definition, the total time spent on the aggregated segment (sum of the waiting time, service time, and the travel time) can be written as the following linear function with $-1$ slope as shown in the first figure in Figure 10.

$$\tilde{g}_m(t) = \begin{cases} \tilde{s}_m & t \geq \bar{t}_m^l \\ \bar{t}_m^l - t + \tilde{s}_m & t < \bar{t}_m^l. \end{cases} \tag{9}$$

*Note: the first condition $t \geq \bar{t}_m$ means we do not need to wait at any node in the segment $S_{m,k}$, and the second condition means we need to wait at some node in the segment $S_{m,k}$.*

It suffices to show that the total time spent on the non-aggregated segment also follows the same function. Again, we prove this by induction.

- *Base case (segment length = 1).* Suppose the segment $S_{k,k} = (x_k)$ contains a single node $x_k$. Then the aggregated problem is identical to the non-aggregated problem by construction, so the total time spent on the non-agggregated segment is exactly Eq. 9 with $m = k$.

- *Inductive Step (segment length = $(k-m)+1 > 1$).* Again, suppose the total time spent on the segment $S_{m+1,k} = (x_{m+1} \to ... \to x_k)$ into $\tilde{S}_{m+1,k} = \{\tilde{x}_{m+1,k}\}$ satisfies the function

$$g_{m+1}(t) = \tilde{g}_{m+1}(t) = \begin{cases} \tilde{s}_{m+1} & t \geq \bar{t}_{m+1}^l \\ \bar{t}_{m+1}^l - t + \tilde{s}_{m+1} & t < \bar{t}_{m+1}^l \end{cases} \tag{10}$$

We now show the total time function $g_m(t)$ for the segment $S_{m,k} = (x_m \to ... \to x_k)$ also equals $\tilde{g}_m(t)$.

By definition of the non-aggregated segment, depending on whether we need to wait at the first node $x_m$, we have:

$$g_m(t) = \begin{cases} s_m^* + g_{m+1}(t + s_m^*) & t \geq t_m^l \\ t_m^l - t + s_m^* + g_{m+1}(t_m^l + s_m^*) & t < t_m^l. \end{cases} \tag{11}$$

*Note: the first condition $t \geq t_m^l$ means we do not need to wait at the first node $x_m$, and the second condition $t < t_m^l$ means we need to wait at the first node $x_m$.*

We split the discussion into the following two cases, based on whether we need to wait *at any node* along the segment $S_{m+1,k}$, if we leave node $x_m$ at $t_m^l$:

1. $t_m^l + s_m^* \geq \bar{t}_{m+1}^l$. In this case, $t_m^l \geq \bar{t}_{m+1}^l - s_m^*$, and hence $\bar{t}_m^l = \max\{t_m^l, \bar{t}_{m+1}^l - s_m^*\}) = t_m^l$ as shown in case 1 of Figure 10. Hence, we have

$$g_m(t) = \begin{cases} s_m^* + g_{m+1}(t + s_m^*) & t \geq \bar{t}_m^l \\ t_m^l - t + s_m^* + g_{m+1}(t_m^l + s_m^*) & t < \bar{t}_m^l. \end{cases} \tag{12}$$

   By inductive hypothesis, we have

$$g_{m+1}(t + s_m^*) = \tilde{s}_{m+1}, \quad t \geq t_m^l = \bar{t}_m^l,$$

   as in this case $t + s_m^* \geq t_m^l + s_m^* \geq \bar{t}_{m+1}^l$.
   Hence, we have

$$\begin{aligned} g_m(t) &= \begin{cases} s_m^* + \tilde{s}_{m+1} & t \geq \bar{t}_m^l \\ t_m^l - t + s_m^* + \tilde{s}_{m+1} & t < \bar{t}_m^l. \end{cases} \\ &= \begin{cases} \tilde{s}_m & t \geq \bar{t}_m^l \\ t_m^l - t + \tilde{s}_m & t < \bar{t}_m^l \end{cases} = \tilde{g}_m(t). \end{aligned} \tag{13}$$

   where we apply the definition of $\tilde{s}_m = s_m^* + \tilde{s}_{m+1}$.

2. $t_m^l + s_m^* < \bar{t}_{m+1}^l$. In this case, $\bar{t}_{m+1}^l - s_m^* > t_m^l$, and hence $\bar{t}_m^l = \max\{t_m^l, \bar{t}_{m+1}^l - s_m^*\} = \bar{t}_{m+1}^l - s_m^*$ as shown in case 2 of Figure 10.
   By inductive hypothesis, we have

$$\begin{aligned} g_{m+1}(t_m^l + s_m^*) &= \bar{t}_{m+1}^l - (t_m^l + s_m^*) + \tilde{s}_{m+1} \\ &= \bar{t}_m^l - t_m^l + \tilde{s}_{m+1} \end{aligned} \tag{14}$$

   We also have, for all $t \geq t_m^l$,

$$\begin{aligned} &g_{m+1}(t + s_m^*) \\ &= \begin{cases} \tilde{s}_{m+1}, & t + s_m^* \geq \bar{t}_{m+1}^l \\ \bar{t}_{m+1}^l - (t + s_m^*) + \tilde{s}_{m+1}, & t + s_m^* < \bar{t}_{m+1}^l \end{cases} \\ &= \begin{cases} \tilde{s}_{m+1}, & t \geq \bar{t}_m^l \\ \bar{t}_m^l - t + \tilde{s}_{m+1}, & t_m^l \leq t < \bar{t}_m^l \end{cases} \end{aligned} \tag{15}$$

   As a result, we have

$$\begin{aligned} g_m(t) &= \begin{cases} s_m^* + \tilde{s}_{m+1} & t \geq \bar{t}_m^l \\ s_m^* + \bar{t}_m^l - t + \tilde{s}_{m+1} & t_m^l \leq t < \bar{t}_m^l \\ t_m^l - t + s_m^* + \bar{t}_m^l - t_m^l + \tilde{s}_{m+1} & t < t_m^l, \end{cases} \\ &= \begin{cases} \tilde{s}_m & t \geq t_m^l \\ \bar{t}_m - t + \tilde{s}_m & t_m^l \leq t < \bar{t}_m^l \\ \bar{t}_m^l - t + \tilde{s}_m & t < t_m^l, \end{cases} \\ &= \begin{cases} \tilde{s}_m & t \geq \bar{t}_m^l \\ \bar{t}_m^l - t + \tilde{s}_m & t < \bar{t}_m^l \end{cases} = \tilde{g}_m(t). \end{aligned} \tag{16}$$

**Condition of Infeasible Temporal Time Windows (Two Hypernodes).** In our time window aggregation, $\bar{t}_j^l$ is responsible for the time expenditure and $\bar{t}_j^r$ is responsible for feasibility. In this case, we have $\bar{t}_j^l > \bar{t}_j^r$, which indicates that to maintain feasibility along the segment, one must arrive at the segment before the aggregated start time $\bar{t}_j^l$, and since one arrives earlier, one must wait at some node within the segment. Since $\bar{t}_j^l > \bar{t}_j^r$ is not permitted according to the definition of VRPTW, we then utilize one additional hypernode to increase the representational capacity such that the first hypernode handles the feasibility component ($\bar{t}_j^r$), and the second hypernode handles the travel time component ($\bar{t}_j^l$). Specifically, $\tilde{t}_j^l = 0$, $\tilde{t}_j^r = \bar{t}_j^r$, $\tilde{t}_k^l = \bar{t}_j^l$, $\tilde{t}_k^r = \infty$ and $\tilde{s}_j = 0$, $\tilde{s}_k = \bar{s}_j$ with the additional constraint that $\text{dist}(\tilde{x}_j, \tilde{x}_k) = 0$.

For time window feasibility (Condition (1)), since $\tilde{t}_j^r = \bar{t}_j^r$, the vehicle must serve the segment before $\bar{t}_j^r$, ensuring the feasibility of serving each customer in the non-aggregated problem. For travel time equivalence (Condition (2)), the time expended before reaching the second node is $\tilde{s}_j + \text{dist}(\tilde{x}_j, \tilde{x}_k) = 0$. Namely, after the vehicle arrives at the segment at time $t$, the travel time is entirely determined by $\tilde{t}_k^l = \bar{t}_j^l$ and $\tilde{s}_k = \bar{s}_j$, whereby in the feasible temporal time window situation, the travel time equivalence is demonstrated.

We complete the time window constraint feasibility proof for VRPTW for both aggregation strategies across all conditions.

□

***Monotonicity Proof.*** For the dual hypernode aggregation, please refer to the *Monotonicity Proof* in B.2.1. For the single hypernode aggregation, notice that

$$f(\tilde{\mathcal{R}}_+) = f(\tilde{\mathcal{R}}_+ \setminus \{\tilde{R}_+^*\}) + f(\{\tilde{R}_+^*\}) = f(\mathcal{R}_+ \setminus \{R_+^*\}) + f(\{\tilde{R}_+^*\})$$
$$= f(\mathcal{R}_+ \setminus \{R_+^*\}) + f(\{R_+^*\}) - \sum_{j \le q < k} dist(x_q, x_{q+1}) \tag{17}$$
$$= f(\mathcal{R}_+) + \text{Const}|_{S_{j,k}}$$

where $\text{Const}|_{S_{j,k}}$ is a constant once the segment $S_{j,k}$ is decided. Therefore, we have:

$$f(\tilde{\mathcal{R}}_+^1) \le f(\tilde{\mathcal{R}}_+^2) \Rightarrow f(\mathcal{R}_+^1) + \text{Const}|_{S_{j,k}} \le f(\mathcal{R}_+^2) + \text{Const}|_{S_{j,k}} \Rightarrow f(\mathcal{R}_+^1) \le f(\mathcal{R}_+^2) \tag{18}$$

□

### B.2.3 VRPB

***Aggregation Strategy (One Hypernode).*** The detailed implementation of FSTA on VRPB can be found in Appendix B.1.5 and Table 7.

***Feasibility Proof [Backhaul Constraint].*** Without loss of generality, we assume all nodes within the segment $S_{j,k}$ are backhaul customers ($b_j = ... = b_k = 1$). Notice that since $\tilde{d} = d_j + ... + d_k$, for the backhaul stage, we have:

$$\sum_{x_i \in \tilde{R}_+^* \text{ and } b_i=1} d_i = \sum_{x_i \in \tilde{R}_+^* \setminus \tilde{S}_{j,k} \text{ and } b_i=1} d_i + \tilde{d}$$
$$= \sum_{x_i \in R_+^* \setminus S_{j,k} \text{ and } b_i=1} d_i + d_j + ... + d_k = \sum_{x_i \in R_+^* \text{ and } b_i=1} d_i \tag{19}$$

For the linehaul stage, we have:

$$\sum_{x_i \in \tilde{R}_+^* \text{ and } b_i=0} d_i = \sum_{x_i \in R_+^* \text{ and } b_i=0} d_i \tag{20}$$

Thus, we have:

$$\sum_{x_i \in \tilde{R}_+^* \text{ and } b_i=0} d_i \le C \quad \Rightarrow \quad \sum_{x_i \in R_+^* \text{ and } b_i=0} d_i \le C$$
$$\sum_{x_i \in \tilde{R}_+^* \text{ and } b_i=1} d_i \le C \quad \Rightarrow \quad \sum_{x_i \in R_+^* \text{ and } b_i=1} d_i \le C \tag{21}$$

Then, we have a feasible $\tilde{\mathcal{R}}_+ \Rightarrow$ a feasible $\mathcal{R}_+$.

$\square$

***Monotonicity Proof.*** Please refer to the monotonicity proof of VRPTW in Appendix B.2.2.

### B.2.4 1-VRPPD.

***Aggregation Strategy (Three Hypernodes).*** The detailed implementation of FSTA on 1-VRPPD can be found in Appendix B.1.5 and Table 7.

***Feasibility Proof [1-Commodity Pickup and Delivery Constraint].*** A feasible $\tilde{\mathcal{R}}_+$ indicates that whenever the vehicle is traveling an aggregated segment $\tilde{S}_{j,k}$, denoted the starting load of the vehicle to be $d_{\text{st}}$ and ending load of the vehicle to be $d_{\text{ed}}$, we have:

$$0 \leq d_{\text{st}} + D^{\min} \leq C$$
$$0 \leq d_{\text{st}} + D^{\min} + D^{\max} - D^{\min} \leq C \tag{22}$$

which requires $-D^{\min} \leq d_{\text{st}} \leq C - D^{\max}$ and $d_{\text{ed}} = d_{\text{st}} + D^k$.

On the other hand, a feasible solution $\mathcal{R}_+$ indicates that whenever the vehicle is traveling a segment $S_{j,k}$, denoted the starting load of the vehicle to be $d_{\text{st}}$ and ending load of the vehicle to be $d_{\text{ed}}$, we have:

$$0 \leq d_{\text{st}} + D^i \leq C, \ \ \forall i \tag{23}$$

which also requires $-D^{\min} \leq d_{\text{st}} \leq C - D^{\max}$ and $d_{\text{ed}} = d_{\text{st}} + D^k$. Then, we have a feasible $\tilde{\mathcal{R}}_+ \Rightarrow$ a feasible $\mathcal{R}_+$.

$\square$

***Monotonicity Proof.*** As $\text{dist}(\tilde{x}_j, \tilde{x}_{\text{mid}}) = \text{dist}(\tilde{x}_{\text{mid}}, \tilde{x}_k) = 0$, we can eliminate the middle hypernode and use a two-hypernode representation when calculating the routing objective. Please refer to the monotonicity proof of CVRP in Appendix B.2.1 for the monotonicity proof of two-hypernode representation.

$\square$

## C L2Seg Details

### C.1 Comparative Analysis of L2Seg Against Existing Methods

**Comparisons with Large Neighborhood Search (LNS)**. (1) LNS (Large Neighborhood Search) operates within a bounded local neighborhood. The algorithm selects a specific region, destroys elements within that boundary, and rebuilds only that portion while keeping the rest of the solution intact. For instance, in Li et al. (2021), LNS selects 3-5 subroutes as its neighborhood, modifying only these routes while leaving all others completely unchanged. There is a clear demarcation between the modified neighborhood and the preserved structure. (2) FSTA (our method), in contrast, operates more globally across the entire solution. It can break existing edges and aggregate segments throughout all subroutes simultaneously, without any predefined neighborhood boundaries. The modifications are distributed across the entire solution rather than confined to a local region, which represents a fundamental departure from existing LNS to more efficiently guide the search. We note that such a flexible framework would not be possible without the proposed ML component, which also constitutes the core novelty and contribution of our work to the field. (3) Moreover, FSTA and LNS are complementary: FSTA can be applied on top of LNS, where LNS first selects a large neighborhood, then FSTA fixes stable edges globally within that selected region.

**Comparisons with Evolutionary Algorithms**. L2Seg framework and evolutionary algorithms (Vidal, 2022)) approach the preservation of solution components from different angles and with distinct goals, and are not interchangeable in use. Evolutionary algorithms (Vidal, 2022)) rely on crossover to merge relatively "good" components from different parents, aiming to promote diversity and generate promising offspring, while our L2Seg framework introduces a learning-guided mechanism to detect unstable edges and aggregates stable edge sequences into hypernodes, enabling a new form of segment-based decomposition that improves scalability and efficiency.

**Comparisons with Path Decomposition Method**. (1) Firstly, path decomposition relies on geometric heuristics (e.g., clustering routes by barycenter distances) to identify decomposition boundaries. In contrast, L2Seg employs deep learning models (synergistic NAR-AR architecture) to intelligently predict which segments should be aggregated, capturing complex patterns that simple heuristics cannot identify. We also propose a novel learning-guided framework with bespoke training and inference processes that are unique to the machine learning method. (2) Secondly, while some prior work explores similar decomposition ideas (e.g., on CVRP only), we are the first to study FSTA decomposition theoretically, providing formal definitions, feasibility theorems, and monotonicity guarantees for various VRPs. (3) Lastly, we empirically demonstrate that by leveraging deep learning in our L2Seg framework, our method consistently achieves significant speedups on state-of-the-art backbones. This provides new insights for the community, highlighting the power of learning-guided optimization in accelerating combinatorial solvers.

**Comparisons with Previous Learning-based Framework L2D (Li et al., 2021)**. (1) Different from the sub-route level, our method detects unstable edges both within and across sub-routes, enabling more global and flexible decomposition. (2) It optimizes beyond localized neighborhoods by identifying improvements that span multiple distant regions simultaneously. (3) It reduces the size of sub-routes by aggregating stable segments into hypernodes, whereas L2D reduces only the number of sub-routes per iteration. This segment-level aggregation allows more adaptive and coarse-grained reduction, offering higher efficiency and solution quality, while remaining complementary to L2D.

**Comparisons with hypergraph decomposition methods Fu et al. (2023) and Li et al. (2025)**. Fu et al. (2023) introduce HDR, a hierarchical destroy-and-repair algorithm that recursively compresses TSP instances to handle problems with millions of cities. While HDR achieves remarkable scalability on very large TSP instances using non-learning heuristics, our approach differs by employing learned policies to identify unstable edges and extending beyond TSP to handle CVRP, VRPTW, and other variants. HDR uses straightforward edge-fixing based on historical local optima, whereas we learn destruction patterns from the lookahead heuristics. Li et al. (2025) propose DRHG, which uses hyper-graphs to reduce consecutive edges and supervised learning for reconstruction. Their approach applies heuristic clustering for destruction followed by ML-based repair of the destroyed segments. Our method takes the opposite approach: we use machine learning to identify unstable edges that should be destroyed, then employ efficient subsolvers for reconstruction. This reversed strategy allows us to leverage learned patterns for the critical decision of what to destroy while using proven optimization techniques for repair. While DRHG demonstrates strong results on TSP and CVRP, our experiments extend to more constrained variants like VRPTW.

## C.2 Input Feature Design Details

Previous works Kool et al. (2018); Li et al. (2021); Kwon et al. (2020) typically utilize only basic input features for routing problems (xy-coordinates and normalized demands for node features, and edge cost for edge features). While neural networks can potentially learn complex patterns from these basic features, tailored feature engineering may lead to enhanced model performance. As illustrated in Appendix B.1, we observe that detecting unstable edges may depend on better capturing local dependencies. We therefore design enhanced node and edge features for our learning task, as shown in Table 8. We also include time windows and service time as node features for VRPTWs.

## C.3 Masking Details

In general, any set of unstable edges could lead to a feasible FSTA problem reduction. However, employing logic-based local search algorithms to select unstable edges can produce more reasonable action space reduction and improved performance. Thus, we design the deletion and insertion stages of L2Seg to emulate a general local search operation.

**For the deletion stage**, given the current node $x$, we mask out nodes that are: (1) not connected to $x$; or (2) part of an edge that has already been deleted during the current deletion stage. Note that the model may select the special ending node $x_{\text{end}}$ to terminate the decoding sequence.

**For the insertion stage**, given the current node $x$, we mask out nodes that are: (1) already connected to $x$; (2) endpoints of two newly inserted edges; or (3) the special ending node $x_{\text{end}}$.

Table 8: Description of enhanced input features for nodes and edges.

| Type | Description | Dimension |
|------|-------------|-----------|
| **Nodes** | The xy coordinates | 2 |
| | The normalized demand | 1 |
| | The centroid of the subtour for each node | 2 |
| | The coordinates of the two nodes connecting to each node | 4 |
| | The travel cost of the two edges connecting to each node | 2 |
| | The relative xy coordinates | 2 |
| | The angles w.r.t. the depot | 1 |
| | The weighted angles w.r.t. the depot by the distances | 1 |
| | The distances of the closest 3 neighbor for each node | 3 |
| | The percentage of the K nearest nodes that are within the same subtour. K=5, 15, 40 | 3 |
| | The percentage of the K% nearest nodes that are within the same subtour. K=5, 15, 40 | 3 |
| **Edges** | The travel cost | 1 |
| | Whether each edge is within the current solution | 1 |
| | The travel cost rank of each edge w.r.t. the corresponding end points | 1 |

### C.4 TRAINING DATA COLLECTION DETAILS

In this section, we present pseudocode that demonstrate the process of generating training labels for both NAR and AR models in Algorithm 2. As a complement to the methodology described in Section 4, we derive our training data from $N_{\mathcal{P}}$ distinct problem instances and extract labels from the first $T_{IS}$ iterative improvement steps. For the AR labels, which emulate feasible local search operations, each label (representing a sequence of nodes) is associated with a quantifiable improvement in solution quality. We retain only those labels that yield improvements exceeding the threshold $\eta_{\text{improv}}$, and we employ stochastic sampling by accepting labels with probability $\alpha_{AC}$. This selective approach ensures both high-quality training signals and sufficient diversity across problem instances and optimization trajectories within the same training budget.

### C.5 INFERENCE DETAILS

In this section, we present the pseudocode that delineates the inference processes of L2Seg-SYN (Algorithm 3), L2Seg-NAR (Algorithm 4), and L2Seg-AR (Algorithm 5). It is important to note that our implementation leverages batch operations for efficient inference across multiple subproblems simultaneously. The K-means clustering algorithm was strategically selected for initial node identification due to its parallelization capabilities. By merging graphs from different subproblems into a unified structure, we can execute the clustering algorithm once for the entire problem space. This parallel clustering approach through K-means significantly enhances decoding efficiency. Notably, within each iterative step, our design requires only a single call of the NAR and AR models, thereby optimizing computational resources.

## D EXPERIMENTAL AND IMPLEMENTATION DETAILS

### D.1 BACKBONE SOLVERS

**LKH-3.** The Lin-Kernighan-Helsgaun algorithm (LKH-3) Helsgaun (2017) represents a strong classical heuristic solver for routing problems, which is widely used in NCO for benchmark. It employs sophisticated $k$-opt moves and effective neighborhood search strategies. For our experiments, we impose time limits rather than local search update limits: 150s and 240s for large-capacity CVRP2k and CVRP5k, respectively, and 2m, 4m, and 10m for VRPTW1k, VRPTW2k, and VRPTW5k, respectively. For small-capacity CVRPs, we adopt the results reported in Zheng et al. (2024).

---

**Algorithm 2:** Training Data Generation

---

**Input:** Solution distribution $\mathcal{P}$, number of instances $N_{\mathcal{P}}$, backbone solver $BS$, number of
iterative steps $T_{IS}$, improvement threshold $\eta_{\text{improv}}$, sample coefficient $\alpha_{AC}$

**Output:** Label sets $\mathcal{Y}_{\text{NAR}}, \mathcal{Y}_{\text{AR}}$

1   $\mathcal{Y}_{\text{NAR}} \leftarrow \emptyset, \mathcal{Y}_{\text{AR}} \leftarrow \emptyset$ **for** $i \leftarrow 1$ **to** $N_{\mathcal{P}}$ **do**

2      Sample $P \sim \mathcal{P}$ and obtain an initial solution $\mathcal{R}$

3      **for** $t \leftarrow 1$ **to** $T_{IS}$ **do**

4          $\mathcal{R}_+ \leftarrow BS(P, \mathcal{R})$                 `// Apply backbone solver`

5          $E_{\text{diff}} \leftarrow (E_{\mathcal{R}} \setminus E_{\mathcal{R}_+}) \cup (E_{\mathcal{R}_+} \setminus E_{\mathcal{R}})$

6          $V_{\text{unstable}} \leftarrow V_{E_{\text{diff}}}$

7          $Y_{\text{NAR}}^P \leftarrow \mathbb{1}\{x \in V_{\text{unstable}}\}$           `// NAR model labels`

8          $\mathcal{Y}_{\text{NAR}} \leftarrow \mathcal{Y}_{\text{NAR}} \cup \{(P, Y_{\text{NAR}}^P)\}$

9          $\mathcal{K}_{\text{TR}} \leftarrow \text{DFS}(P, V_{\text{unstable}}, E_{\text{diff}})$        `// Find sequences`

10         **foreach** $K \in \mathcal{K}_{\text{TR}}$ **do**

11             Obtain $P_K$ with solution $R_K$ and sequence $y_K$ with Improvement

12             **if** *Improvemnet* $\geq \eta_{\text{improv}}$ and with probability $\alpha_{AC}$ **then**

13                $\mathcal{Y}_{\text{AR}} \leftarrow \mathcal{Y}_{\text{AR}} \cup \{(P_K, y_K)\}$      `// AR model labels`

14             **end if**

15                    `// Skip sequences with low improvement or by`
            `probability`

16         **end foreach**

17          $\mathcal{R} \leftarrow \mathcal{R}_+$                 `// Update current solution`

18      **end for**

19   **end for**

20   **return** $\mathcal{Y}_{NAR}, \mathcal{Y}_{AR}$

---

**Algorithm 3:** L2Seg-SYN: Synergized Prediction

---

**Input:** Problem $P$, current solution $\mathcal{R}$, NAR model, AR model, threshold $\eta$, number of clusters
$n_{\text{KMEANS}}$

**Output:** Set of unstable edges $E_{\text{unstable}}$

1   $\mathcal{P}_{\text{TR}} \leftarrow \text{DecomposeIntoSubproblems}(P, \mathcal{R})$       `// Partition into` $\sim |\mathcal{R}|$
   `subproblems`

2   $E_{\text{unstable}} \leftarrow \emptyset$

3   **for** *each subproblem* $P_{\text{TR}} \in \mathcal{P}_{\text{TR}}$ **do**

4      $\mathbf{p}^{\text{NAR}} \leftarrow \text{NARModel}(P_{\text{TR}})$      `// Get NAR predictions for each node`

5      $\hat{y}_{\text{NAR}} \leftarrow \{x_i \mid p_i^{\text{NAR}} \geq \eta\}$     `// Identify unstable nodes via threshold`

6      $\text{Clusters} \leftarrow \text{KMeans}(\hat{y}_{\text{NAR}}, n_{\text{KMEANS}})$      `// Group unstable nodes into`
     `clusters`

7      $\text{InitialNodes} \leftarrow \{x \mid x = \arg\max_{x_i \in c} p_i^{\text{NAR}}, c \in \text{Clusters}\}$

8          `// Select initial node with highest probability for the AR`
    `model`

9      $E_{\text{unstable}}^{P_{\text{TR}}} \leftarrow \emptyset$              `// Unstable edges for this subproblem`

10     **for** *each node* $x_{init} \in$ *InitialNodes with corresponding* $P_{TR}$ **do**

11         $E_{x_{\text{init}}}^{P_{\text{TR}}} \leftarrow \text{ARModel}(P_{\text{TR}}, x_{\text{init}})$      `// Get unstable edges via the AR`
      `model`

12         $E_{\text{unstable}}^{P_{\text{TR}}} \leftarrow E_{\text{unstable}}^{P_{\text{TR}}} \cup E_{x_{\text{init}}}^{P_{\text{TR}}}$

13     **end for**

14      $E_{\text{unstable}} \leftarrow E_{\text{unstable}} \cup E_{x_{\text{init}}}^{P_{\text{TR}}}$          `// Aggregate unstable edges`

15   **end for**

16   **return** $E_{unstable}$

---

---

**Algorithm 4:** L2Seg-NAR: Non-Autoregressive Prediction

---

**Input:** Problem $P$, current solution $\mathcal{R}$, NAR model, threshold $\eta$
**Output:** Set of unstable edges $E_{\text{unstable}}$

1   $\mathcal{P}_{\text{TR}} \leftarrow \text{DecomposeIntoSubproblems}(P, \mathcal{R})$      `// Partition into ∼ |R|`
     `subproblems`

2   $E_{\text{unstable}} \leftarrow \emptyset$

3 **for** *each subproblem $P_{TR} \in \mathcal{P}_{TR}$* **do**

4     $\mathbf{p}^{\text{NAR}} \leftarrow \text{NARModel}(P_{\text{TR}})$      `// Get NAR predictions for each node`

5     $\hat{y}_{\text{NAR}} \leftarrow \{x_i \mid p_i^{\text{NAR}} \geq \eta\}$      `// Identify unstable nodes via threshold`

6     $E_{\text{unstable}}^{P_{\text{TR}}} \leftarrow \{(x_i, x_j) \mid x_i \in \hat{y}_{\text{NAR}} \text{ or } x_j \in \hat{y}_{\text{NAR}}, \text{ and } (x_i, x_j) \in E_{P_{\text{TR}}}\}$

7     `// Mark all edges connected to the unstable nodes as`
     `unstable`

8     $E_{\text{unstable}} \leftarrow E_{\text{unstable}} \cup E_{\text{unstable}}^{P_{\text{TR}}}$      `// Aggregate unstable edges`

9 **end for**

10 **return** $E_{\text{unstable}}$

---

**Algorithm 5:** L2Seg-AR: Autoregressive Prediction

---

**Input:** Problem $P$, current solution $\mathcal{R}$, AR model, number of clusters $n_{\text{KMEANS}}$
**Output:** Set of unstable edges $E_{\text{unstable}}$

1   $\mathcal{P}_{\text{TR}} \leftarrow \text{DecomposeIntoSubproblems}(P, \mathcal{R})$      `// Partition into ∼ |R|`
     `subproblems`

2   $E_{\text{unstable}} \leftarrow \emptyset$

3 **for** *each subproblem $P_{TR} \in \mathcal{P}_{TR}$* **do**

4     $\text{Clusters} \leftarrow \text{KMeans}(\text{AllNodes in } P_{\text{TR}}, n_{\text{KMEANS}})$      `// Cluster all nodes`

5     $\text{Centroids} \leftarrow \{\text{ComputeCentroid}(c) \mid c \in \text{Clusters}\}$

6     $\text{InitialNodes} \leftarrow \{x \mid x = \arg\min_{x_i \in c} \text{Distance}(x_i, \text{centroid}_c), c \in \text{Clusters}\}$

7     `// Select node closest to each cluster centroid for the AR`
     `model`

8     $E_{\text{unstable}}^{P_{\text{TR}}} \leftarrow \emptyset$      `// Unstable edges for this subproblem`

9     **for** *each node $x_{init} \in \text{InitialNodes}$ with corresponding $P_{TR}$* **do**

10       $E_{x_{\text{init}}}^{P_{\text{TR}}} \leftarrow \text{ARModel}(P_{\text{TR}}, x_{\text{init}})$      `// Get unstable edges via the AR`
      `model`

11       $E_{\text{unstable}}^{P_{\text{TR}}} \leftarrow E_{\text{unstable}}^{P_{\text{TR}}} \cup E_{x_{\text{init}}}^{P_{\text{TR}}}$

12     **end for**

13     $E_{\text{unstable}} \leftarrow E_{\text{unstable}} \cup E_{\text{unstable}}^{P_{\text{TR}}}$      `// Aggregate unstable edges`

14 **end for**

15 **return** $E_{\text{unstable}}$

---

**LNS.** Local Neighborhood Search (LNS) Shaw (1998) is a powerful decomposition-based metaheuristic that iteratively improves solutions by destructively and constructively exploring defined search neighborhoods. We implement LNS following the approach in Li et al. (2021), where neighborhoods consisting of three adjacent subroutes are randomly selected for re-optimization. We establish time limits of 150s and 240s for large-capacity CVRP2k and CVRP5k, respectively; 2.5m, 4m, and 5m for small-capacity CVRP1k, CVRP2k, and CVRP5k, respectively; and 2m, 4m, and 10m for VRPTW1k, VRPTW2k, and VRPTW5k, respectively. LKH-3 serves as the backbone solver with a 1,000 per-step local search updates limit.

**L2D.** Learning to Delegate (L2D) Li et al. (2021) is the state-of-the-art learning-based optimization framework that integrates neural networks with classical optimization solvers to intelligently delegate subproblems to appropriate solvers. The framework employs a neural network trained to identify the most promising neighborhoods for improvement. For comparative fairness, we apply identical time limits and backbone solver configurations as used in our LNS implementation. When augmented by L2Seg, training proceeds in two stages: we first train the L2D models following the methodology in Li et al. (2021), then train the L2Seg model using the resulting pre-trained L2D models.

**Initial Solution Heuristics.** For both training data generation and inference, we employ the initial solution heuristic inspired by (Li et al., 2021). Our method partitions nodes according to their angular coordinates with respect to the depot. We begin by selecting a reference node, marking its angle as 0, and incrementally incorporate additional nodes into the same group until the collective demand approaches the capacity threshold ($\alpha_{\text{init}} K_{\text{veh}} C \approx \sum d_i$), where approximately $K_{\text{veh}}$ vehicles would be required to service the group. This process continues sequentially, forming new groups until all customers are assigned. Finally, we apply LKH-3 in parallel to solve each subproblem independently. In our implementation, we set $K_{\text{veh}} = 6$ and $\alpha_{\text{init}} = 0.95$ as the controlling parameters.

## D.2 BASELINES

In this section, we provide further clarification regarding the baselines used in our comparative analysis, beyond the backbone solvers. We independently executed LKH-3, LNS, and L2D using consistent parameters. Results for SIL were sourced from Luo et al. (2024), L2R from Zhou et al. (2025a), and all other baselines from Zheng et al. (2024). When multiple variants of a baseline were presented in the original publications, we selected the configuration that achieved the best objective values. Since the original implementations of SIL (Luo et al., 2025; 2024) and NDS (Hottung et al., 2025) was evaluated on NVIDIA GeForce RTX 3090 GPU and NVIDIA A100 GPUs whereas our experiments use NVIDIA V100 GPUs, we re-ran them on our hardware for fair comparison. Note that the instance settings used in Table 2 follow Table 2 of SIL in its earlier version (Luo et al., 2024), which was removed in the latest version (Luo et al., 2025). We therefore cite both versions when comparing results.

It is important to note that all reported results were evaluated on identical test instances (for CVRPs) or on instances sampled from the same distribution (for VRPTWs), ensuring fair comparison. Moreover, our experiments were conducted on hardware with less powerful GPUs compared to those utilized in Zheng et al. (2024); Zhou et al. (2025a); Luo et al. (2024). This hardware discrepancy suggests that the performance advantages demonstrated by our proposed model would likely persist or potentially increase if all methods were evaluated on identical computing infrastructure.

We re-implemented the backbone solvers and L2D (Li et al., 2021) to ensure a fair and strong comparison. Notably, prior studies (Zheng et al., 2024; Ye et al., 2024) did not explore configurations optimized for L2D's full potential. Specifically, they imposed overly conservative limits (e.g., only allowing 1 trail) on LKH-3 local search updates and did not supply current solution information to the LKH-3 solver during the resolution process. This significantly weakened L2D's performance in their benchmarks. In contrast, our comparison reflects L2D's best achievable performance.

## D.3 PARAMETERS AND TRAINING HYPERPARAMETERS

**Parameters.** Table 9 lists the values of parameters used in training data generation and inference. **Training Hyperparameters.** For model training, we optimize both NAR and AR architectures using the ADAM optimizer with a consistent batch size of 128 across 200 epochs for all problem variants. The learning rate is calibrated at $10^{-3}$ for large-capacity CVRPs and $10^{-4}$ for small-capacity CVRPs

Table 9: A list of parameters and their values used in our experiments for training and inference.

| Training Data Generation | |
|---|---|
| **Parameter** | **Value** |
| # of instances $N_{\mathcal{P}}$ | 1000 |
| # of iterative steps $T_{IS}$ | 40 |
| Improvement threshold $\eta_{\text{improv}}$ | 0 |
| Sample coefficient $\alpha_{AC}$ | 0     for small-capacity CVRPs and VRPTWs |
| | 0.4    for large-capacity CVRPs |

| Inference | |
|---|---|
| **Parameter** | **Value** |
| Threshold $\eta$ for NAR model | 0.6 |
| # of K-MEANS clusters $n_{\text{KMEANS}}$ | 3 |
| # of LKH-3 local search updates limit per iterative step | 1000 |
| Solve time limits | 150s, 240s    for large-capacity CVRP2k, 5k |
| | 2.5m, 4m, 5m    for small-capacity CVRP1k, 2k, 5k |
| | 2m, 4m, 10m    for VRPTW1k, 2k, 5k |

and VRPTWs. The loss function employs weighted components with $w_{\text{pos}} = 9$, $w_{\text{insert}} = 0.8$, and $w_{\text{delete}} = 0.2$. All computational experiments are conducted on a single NVIDIA V100 GPU, with training duration ranging from approximately 0.5 to 1.5 days, scaling with problem dimensionality.

Regarding network architecture, our encoder maps node features $\mathbf{X} \in \mathbb{R}^{n \times 25}$ for standard problems ($\mathbf{X} \in \mathbb{R}^{n \times 28}$ for VRPTWs) to node embeddings via $\mathbf{h}_i^{\text{init}} = \text{Concat}(\mathbf{h}_i^{\text{MLP}}, \mathbf{h}_i^{\text{POS}}) \in \mathbb{R}^{2d_h}$, where $d_h = 128$. They then undergo processing through $L_{\text{TFM}} = 2$ Transformer layers (Vaswani, 2017) with route-specific attention masks, followed by a Graph Attention Network to derive the final node embeddings $\mathbf{H}^{\text{GNN}}$. The transformer implementation utilizes 2 attention heads, 0.1 dropout regularization, ReLU activation functions, layer normalization, and feedforward dimensionality of 512. Our GNN employs a transformer convolution architecture with 2 layers ($L_{\text{GNN}} = 2$) and a single attention head.

Supplementary to the specifications in Section 4, we delineate additional hyperparameters for our decoder modules. The NAR decoder computes $\mathbf{p}^{\text{NAR}}$ (node instability probabilities) via an MLP with sigmoid activation for final probability distribution. The AR decoder incorporates single-layer Gated Recurrent Units (GRUs), complemented by a single-layer/single-head transformer for the deletion mechanism and a four-layer/single-head transformer for the insertion procedure.

All the training hyperparameters are summarized in Table 10.

### D.4 INSTANCE GENERATION

In general, we generate all training and test instances following established methodologies: Zheng et al. (Zheng et al., 2024) for CVRP and Solomon (Solomon, 1987) for VRPTW. Specifically, For small-capacity CVRPs, nodes are uniformly distributed within the $[0, 1]$ square, with integer demands ranging from 1 to 9 (inclusive). Vehicle capacities are set to $C = 200$, 300, and 300 for problem sizes 1k, 2k, and 5k, respectively. For large-capacity CVRPs, we maintain identical configurations except for increased vehicle capacities of $C = 500$ and 1000 for CVRP1k and CVRP5k, respectively. For VRPTWs, we adopt the same spatial distribution, demand structure, and capacity constraints as the small-capacity CVRPs. Service times are uniformly set to 0.2 time units for each customer and 0 for the depot. Time windows are generated according to the methodology outlined in Solomon (Solomon, 1987).

Our experimental framework comprises distinct datasets for training, validation, and testing:

- **Training:** 1,000 instances for each problem type and scale to generate training labels
- **Validation:** 30 instances per problem configuration

Table 10: A list of hyperparameters and their values used in our model architecture and training.

| Training Configuration | |
|---|---|
| **Parameter** | **Value** |
| Optimizer | ADAM |
| Batch size | 128 |
| # of epochs | 200 |
| Learning rates | $10^{-3}$    for large-capacity CVRPs |
| | $10^{-4}$    for small-capacity CVRPs and VRPTWs |
| Weight of unstable nodes $w_{\text{pos}}$ | 9 |
| Weight of prediction in insert stage $w_{\text{insert}}$ | 0.8 |
| Weight of prediction in delete stage $w_{\text{delete}}$ | 0.2 |
| Computing Resource | Single NVIDIA V100 GPU |
| **Model Architecture** | |
| **Parameter** | **Value** |
| Hidden dimension | 128 |
| ***Encoder Transformer*** | |
| # of layers $L_{\text{TFM}}$ | 2 |
| # of attention heads | 2 |
| Dropout regularization | 0.1 |
| Activation function | ReLU |
| Feedforward dimension | 512 |
| Normalization | Layer normalization |
| ***Encoder GNN*** | |
| Architecture | Transformer Convolution Network |
| # of layers $L_{\text{GNN}}$ | 2 |
| # of attention heads | 1 |
| ***Decoder Components*** | |
| NAR decoder activation function | Sigmoid |
| # of layers in GRUs | 1 |
| ***AR Transformer in Deletion Stage*** | |
| # of layers $L_{\text{delete}}^{\text{MHA}}$ | 1 |
| # of attention heads | 1 |
| ***AR Transformer in Insertion Stage*** | |
| # of layers $L_{\text{insert}}^{\text{MHA}}$ | 4 |
| # of attention heads | 1 |

- **Testing:** For small-capacity CVRPs, we utilize the 1,000 test instances from Zheng et al. (Zheng et al., 2024); for large-capacity CVRPs and VRPTWs, we evaluate on 100 instances sampled from the same distribution as the training data

# E    ADDITIONAL EXPERIMENTS AND ANALYSIS

## E.1    HYPERPARAMETER STUDY

Figure 11 depicts the effects of $n_{\text{KMEANS}}$ and $\eta$. We observe that the best performance is when $n_{\text{KMEANS}} = 3$ and $\eta = 0.6$, suggesting that designating a moderate proportion of edges as unstable represents the most effective strategy.

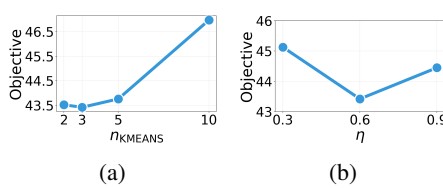

(a)        (b)

Figure 11: Analysis of key hyperparameters: (a) number of clusters $n_{\text{MEANS}}$, and (b) balancing factor $\eta$.

### E.2 RESULTS ON REALISTIC ROUTING DATASETS

We further evaluate L2Seg on the CVRPLib realistic routing dataset (Uchoa et al., 2017; Arnold et al., 2019), adhering to the settings established in Zheng et al. (2024), which incorporates instances from CVRP Set-X [54] and the very large-scale CVRP dataset Set-XXL in the test set. The instances within CVRPLib exhibit more realistic spatial distributions (distinct from simplistic uniform or clustered patterns), greater diversity, and better representation of real-world logistical challenges. For this evaluation, we employ models trained on synthetic small-capacity CVRP2k and CVRP5k datasets and zero-shot transfer them to CVRPLib. Time constraints of 240s and 600s are implemented for L2Seg during testing. Additional methodological details are provided in Appendix D. As demonstrated in Table 11, LNS augmented with L2Seg-SYN surpasses all other learning-based methods in performance. Significantly, the computational time required by LNS+L2Seg-SYN (600s) is substantially less than that of the previously best-performing learning-based model, UDC-$x_{250}$. These results further substantiate L2Seg's exceptional generalizability across varied problem distributions.

Table 11: CVRPLib results. We present the gap to the best known solutions (%).

| Dataset, $N \in$ | LEHD | ELG aug×8 | GLOP-LKH3 | TAM(LKH3) |
|---|---|---|---|---|
| Set-X,(500,1,000] | 17.4% | 7.8% | 16.8% | 9.9% |
| Set-XXL,(1,000,10,000] | 22.2% | 15.2% | 19.1% | 20.4% |

| Dataset, $N \in$ | UDC-$x_2$ | UDC-$x_{250}$ | LNS+L2Seg-SYN (240s) | LNS+L2Seg-SYN (600s) |
|---|---|---|---|---|
| Set-X,(500,1,000] | 16.5% | 7.1% | 7.5% | 6.9% |
| Set-XXL,(1,000,10,000] | 31.3% | 13.2 % | 12.5% | 12.0% |

### E.3 RESULTS ON CLUSTERED CVRP AND HETEROGENEOUS-DEMAND CVRP

Table 12: Results on clustered CVRP and heterogeneous-demand CVRP. We present gains to the backbone solver LNS and the performance of LKH-3 for reference.

| Methods | Clustered CVRP2k | | | Clustered CVRP5k | | |
|---|---|---|---|---|---|---|
| | Obj.↓ | Gain↑ | Time↓ | Obj.↓ | Gain↑ | Time↓ |
| LKH-3 (Helsgaun, 2017) (for reference) | 42.06 | - | 150s | 62.33 | - | 240s |
| LNS (Shaw, 1998) | 41.54 | 0.00% | 150s | 61.42 | 0.00% | 240s |
| L2Seg-SYN-LNS (zero-shot transfer) | 41.03 | 1.23% | 150s | 60.87 | 0.90% | 240s |
| L2Seg-SYN-LNS | **40.73** | **1.95%** | 150s | **60.11** | **2.13%** | 240s |

| Methods | Hetero-demand CVRP2k | | | Hetero-demand CVRP5k | | |
|---|---|---|---|---|---|---|
| | Obj.↓ | Gain↑ | Time↓ | Obj.↓ | Gain↑ | Time↓ |
| LKH-3 (Helsgaun, 2017) (for reference) | 46.02 | - | 150s | 65.89 | - | 240s |
| LNS (Shaw, 1998) | 45.77 | 0.00% | 150s | 64.81 | 0.00% | 240s |
| L2Seg-SYN-LNS (zero-shot transfer) | 44.35 | 3.10% | 150s | 64.28 | 0.82% | 240s |
| L2Seg-SYN-LNS | **44.15** | **3.54%** | 150s | **64.15** | **1.02%** | 240s |

To demonstrate L2Seg's robustness across diverse and more realistic scenarios beyond uniform distributions, we provide in-distribution and zero-shot generalization evaluation of our L2Seg on instances with different customer and demand distributions.

Following Li et al. (2021), we generate clustered CVRP instances with 7 clusters. For heterogeneous-demand scenarios, we employ a skewed distribution where high and low demands ($d \in \{1, 2, 8, 9\}$) occur with probability 0.2 each, while others ($d \in \{3, 4, 5, 6, 7\}$) occur with probability 0.04 each. All experiments use LNS as the backbone solver, with LKH-3 included for reference.

Table 12 presents the comprehensive results. L2Seg demonstrates consistent improvements across all settings: zero-shot transfer achieves 1.23% to 3.10% gains over LNS, while in-distribution testing reaches 1.02% to 3.54% improvements depending on problem size and variant. These experiments

demonstrate that L2Seg maintains consistent improvements across diverse real-world conditions, from uniform spatial layouts to clustered distributions and heterogeneous demands.

### E.4 STANDARD DEVIATION COMPARISON

In this section, we provide standard deviation statistics for L2Seg-SYN across three different backbone solvers on large-capacity CVRPs. We conduct 5 independent trials using different random seeds for each method. All experiments are terminated at the specified time limit, and we report the standard deviations of the objective values for all 6 methods. The results are presented in Table 13. While LKH-3 exhibits the lowest variance among baseline methods, our L2Seg approach also demonstrates consistently low variance across different problem types and backbone solvers, confirming both the effectiveness and stability of our method.

Table 13: Performance comparison of backbone solvers with and without L2Seg-SYN on large-scale CVRP instances. Results represent mean objective values $\pm$ standard deviation across 5 independent trials of testing. L2Seg-SYN demonstrates consistent performance improvements with low variance, indicating both effectiveness and stability of the approach.

| | CVRP2k | | | CVRP5k | | |
|---|---|---|---|---|---|---|
| **Methods** | **Obj.↓** | **Gain↑** | **Time↓** | **Obj.↓** | **Gain↑** | **Time↓** |
| LKH-3 Helsgaun (2017) | $45.24 \pm 0.17$ | 0.00% | 152s | $65.34 \pm 0.29$ | 0.00% | 242s |
| LKH+L2Seg-SYN | $43.92 \pm 0.20$ | 2.92% | 152s | $64.12 \pm 0.34$ | 1.87% | 248s |
| LNS Shaw (1998) | $44.92 \pm 0.24$ | 0.00% | 154s | $64.69 \pm 0.37$ | 0.00% | 246s |
| LNS+L2Seg-SYN | $43.42 \pm 0.22$ | 3.34% | 152s | $63.94 \pm 0.35$ | 1.16% | 241s |
| L2D Li et al. (2021) | $43.69 \pm 0.21$ | 0.00% | 153s | $64.21 \pm 0.32$ | 0.00% | 243s |
| L2D+L2Seg-SYN | $43.35 \pm 0.23$ | 0.78% | 157s | $63.89 \pm 0.34$ | 0.50% | 248s |

### E.5 CASE STUDY: COMPARISON OF PREDICTIONS OF THREE L2SEG APPROACHES

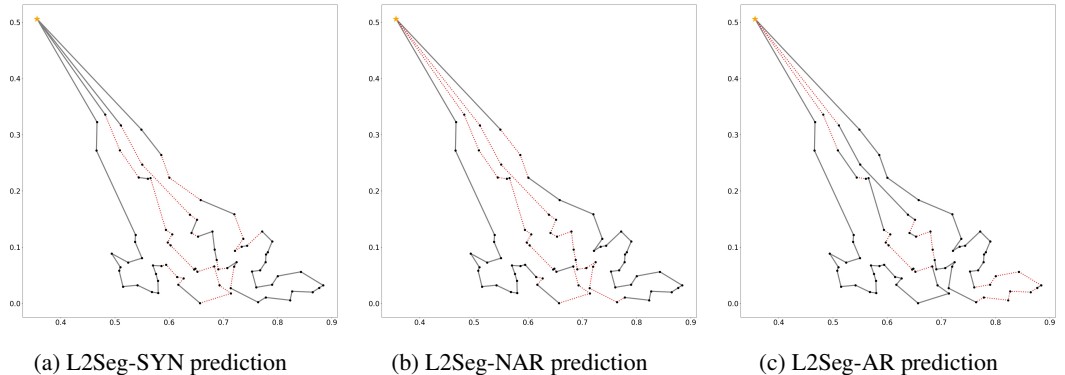

(a) L2Seg-SYN prediction   (b) L2Seg-NAR prediction   (c) L2Seg-AR prediction

Figure 12: Prediction comparison of L2Seg-SYN, L2Seg-NAR, and L2Seg-AR on two adjacent routes from a small-capacity CVRP1k solution. Red dashed lines indicate predicted unstable edges. L2Seg-SYN provides the most reasonable predictions, while L2Seg-NAR over-predicts unstable edges and L2Seg-AR fails to identify unstable regions.

We present a case study on a small-capacity CVRP1k instance to analyze model prediction behavior. Since the learned model ultimately predicts on two adjacent routes, we visualize unstable edge predictions (red dashed lines) for two such routes using L2Seg-SYN, L2Seg-NAR, and L2Seg-AR in Figure 12. L2Seg-SYN demonstrates selective prediction behavior, avoiding boundary edges while targeting specific unstable edges within route interiors—a pattern consistent with our observations in Appendix B.1.1. L2Seg-NAR successfully identifies unstable regions (route interiors) but lacks discrimination, predicting nearly all edges within these regions as unstable without capturing local dependencies. L2Seg-AR exhibits selective prediction within regions but fails to properly identify unstable regions, as many predictions occur at boundaries. These results provide insight into

Table 14: Unstable and stable edges convergence at the first 10 iterations

| Round # | 1 | 2 | 3 | 5 | 7 | 9 |
|---|---|---|---|---|---|---|
| Unstable Edge Overlapping Percentage | 28.2% | 33.5% | 41.2% | 49.2% | 48.8% | 54.1% |
| Stable Edge Overlapping Percentage | 47.2% | 58.2% | 60.5% | 64.7% | 67.3% | 69.4% |
| Avg Segment Length | 2.45 | 2.57 | 2.44 | 3.04 | 2.87 | 2.73 |

L2Seg-SYN's hybrid approach: the NAR component first identifies unstable regions, while the AR component leverages local information to make accurate predictions within each identified region.

### E.6  Unstable and Stable Edges Convergence

We conducted experiments measuring overlapping predicted edges between adjacent iterations over the first 10 rounds, revealing interesting dynamics: The overlap of predicted unstable edges increases from 28% to 54%, while stable edge overlap increases from 47% to 69% across iterations, shown in the Table 14. This indicates gradual but not rapid convergence, allowing our method to continuously explore new regions for re-optimization rather than getting trapped in fixed segments.

### E.7  The Neural Network Overheads of L2Seg

We measured the overhead across CVRP and VRPTW instances ranging from 1k to 5k nodes in the Table 15. Even with our most complex model (L2Seg-SYN), the overhead consistently remains below 10% of the total iteration time (ranging from 7.2% to 9.6%). This indicates that the overhead scales efficiently and predictably. Generally speaking, the overhead primarily stems from neural network inference, which is driven by two factors: input data size and network call frequency. Regarding the former, L2Seg employs a Batched Sub-Route Processing design. During embedding, we split problems into adjacent route pairs and use batch processing. This avoids memory bottlenecks and ensures that inference time scales efficiently for large-scale cases. Regarding the latter, the frequency of calling L2Seg is a tunable hyperparameter.

Table 15: Computational overhead analysis of L2Seg variants across problem scales

| Method | CVRP1k | | | CVRP2k | | | CVRP5k | | |
|---|---|---|---|---|---|---|---|---|---|
| | Avg L2Seg Time/Iter | Avg Total Time/Iter | Overhead Rate | Avg L2Seg Time/Iter | Avg Total Time/Iter | Overhead Rate | Avg L2Seg Time/Iter | Avg Total Time/Iter | Overhead Rate |
| L2Seg-NAR-LNS | 0.38s | 8.4s | 4.5% | 0.62s | 11.4s | 5.4% | 0.80s | 12.9s | 6.2% |
| L2Seg-AR-LNS | 0.63s | 9.8s | 6.4% | 0.90s | 10.8s | 8.3% | 1.24s | 13.9s | 8.9% |
| L2Seg-SYN-LNS | 0.76s | 10.5s | 7.2% | 1.14s | 12.5s | 9.1% | 1.33s | 14.1s | 9.4% |
| | VRPTW1k | | | VRPTW2k | | | VRPTW5k | | |
| L2Seg-NAR-LNS | 0.37s | 8.9s | 4.2% | 0.61s | 10.4s | 5.9% | 0.83s | 12.8s | 6.5% |
| L2Seg-AR-LNS | 0.62s | 9.1s | 6.8% | 0.96s | 11.9s | 8.1% | 1.14s | 12.4s | 9.2% |
| L2Seg-SYN-LNS | 0.76s | 10.1s | 7.5% | 0.97s | 10.8s | 9.0% | 1.36s | 14.2s | 9.6% |

### E.8  Time of Training L2Seg

We give the time spent on data collection, training the NAR, AR, and total time when training L2Seg. Empirically, our training pipeline scales efficiently, requiring under 3 days for CVRP5k. We project that scaling to 10k instances would take approximately 6 days, which is still feasible.

## F  Broader Impacts

On one hand, the integration of deep learning into discrete optimization offers promising advances for real-world domains such as public logistics and transportation systems, where additional considerations for social equity and environmental sustainability can be incorporated. On the other

Table 16: Training time breakdown for L2Seg across problem scales

|  | Data Collection | Training time NAR | Training time AR | Total (parallel) |
|---|---|---|---|---|
| CVRP1k | 6.3h | 6.8h | 13.7h | 20.0h |
| CVRP2k | 10.4h | 10.4h | 22.4h | 32.8h |
| CVRP5k | 19.2h | 18.1h | 40.7h | 59.9h |

hand, the application of deep learning methodologies in discrete optimization necessitates substantial computational resources for model training, potentially leading to increased energy consumption and carbon emissions. The quantification and mitigation of these environmental impacts represent critical areas for ongoing research and responsible implementation.

## G   LARGE LANGUAGE MODELS USAGE

We used LLMs to assist with manuscript revision. After completing the initial draft without LLM assistance, we consulted LLMs for suggestions on improving specific text passages. All LLM-generated advice was carefully reviewed to ensure accuracy before incorporation. LLMs were not used for research tasks or any purpose beyond text refinement.

