# OpenReview forum: "Learning to Segment for Vehicle Routing Problems"
_ICLR.cc/2026/Conference — ICLR 2026 Oral_

### Official Review · Reviewer_L8xN · 2025-10-30

**Soundness:** 2
**Presentation:** 3
**Contribution:** 2
**Rating:** 4
**Confidence:** 2

**Summary:**

This paper proposes First-Segment-Then-Aggregate (FSTA), a decomposition framework for large-scale VRPs, motivated by the observation that in iterative optimization methods, a large portion of the intermediate solution structure tends to remain stable across iterations.
To effectively identify which parts of the solution should be modified and which should remain fixed, the authors introduce a neural network–based module called Learning-to-Segment (L2Seg), which enables efficient segmentation within FSTA.
Through experiments on CVRP and TSPTW, the authors demonstrate that integrating L2Seg can accelerate existing iterative solvers by a factor of approximately 2×–7×, while maintaining or improving solution quality.

**Strengths:**

- The proposed method successfully restricts the search space and achieves higher solution quality within the same computational budget on large-scale CVRP and TSPTW instances.

- The framework is solver-agnostic and can be applied to multiple backbone iterative solvers, demonstrating its flexibility.

- The experimental evaluation is comprehensive, including comparisons against diverse baselines from different methodological categories.

**Weaknesses:**

- Since L2Seg is trained in a supervised manner, its performance may degrade when the distribution of intermediate solutions during inference differs significantly from that of the training data.

- The behavior of L2Seg appears conceptually closer to search-space restriction rather than genuine problem decomposition. Consequently, there is a risk that the restricted search could lead to premature convergence to local optima, preventing the discovery of globally superior solutions.

- The model is trained and evaluated on problem instances of the same size distribution, leaving it unclear whether L2Seg generalizes to unseen problem scales. If it fails to generalize, collecting sufficient training data for larger instances could become computationally prohibitive, limiting the practical applicability of the approach.

- (Minor) Typo found at line 130.

- (Minor) References in lines 73–75 should be enclosed in parentheses.

**Questions:**

- How do the authors precisely define “iterative solvers” in the context of this paper?

- The paper states that “a large portion of the solution remains stable.” Does this mean that most parts of the solution are not updated because they do not significantly contribute to objective improvement, or because only a small subset of solution elements is actually targeted for update by the iterative process?

- Does L2Seg generalize to problem sizes that were not seen during training? If not, how feasible is it to train the model for significantly larger problem instances?

---

> ### Author Response · Authors · 2025-11-25
> **Response to Reviewer L8xN (1/4)**
>
> We thank the reviewer for recognizing our method's efficiency on large-scale instances and its solver-agnostic flexibility. We have carefully noted your concerns regarding search space restriction, premature convergence, and generalization. Below, we address these points in detail to clarify the robustness of our approach.
>
> ---
>
> > **W1:** Since L2Seg is trained in a supervised manner, its performance may degrade when the distribution of intermediate solutions during inference differs significantly from that of the training data.
> >
>
> Thanks for raising this concern. While we acknowledge the potential risk of performance degradation for out-of-distribution data, we have conducted comprehensive generalization experiments that demonstrate L2Seg's robustness.
>
> In Table 11 (also copied below), we provide both in-distribution and zero-shot generalization evaluation on instances with fundamentally different customer and demand distributions (detailed settings in Appendix E.3). For zero-shot transfer, not only the intermediate representations but also the entire problem characteristics differ from our training data. The results show that even under zero-shot transfer, L2Seg consistently outperforms the backbone LNS solver with gains ranging from 0.82% to 3.10%, demonstrating strong generalizability across diverse problem distributions. The performance gap between zero-shot and in-distribution settings is not large, suggesting that our learned stability patterns transfer well across different VRP characteristics.
>
> | Methods | Clustered CVRP2k |  |  | Clustered CVRP5k |  |  |
> | --- | --- | --- | --- | --- | --- | --- |
> |  | Obj.↓ | Gain↑ | Time↓ | Obj.↓ | Gain↑ | Time↓ |
> | LKH-3 (Helsgaun, 2017) (for reference) | 42.06 | - | 150s | 62.33 | - | 240s |
> | LNS (Shaw, 1998)  | 41.54 | 0.00% | 150s | 61.42 | 0.00% | 240s |
> | L2Seg-SYN-LNS (zero-shot transfer) | 41.03 | 1.23% | 150s | 60.87 | 0.90% | 240s |
> | L2Seg-SYN-LNS (re-trained) | **40.73** | **1.95%** | 150s | **60.11** | **2.13%** | 240s |
>
> | Methods | Hetero-demand CVRP2k |  |  | Hetero-demand CVRP5k |  |  |
> | --- | --- | --- | --- | --- | --- | --- |
> |  | Obj.↓ | Gain↑ | Time↓ | Obj.↓ | Gain↑ | Time↓ |
> | LKH-3 (Helsgaun, 2017) (for reference) | 46.02 | - | 150s | 65.89 | - | 240s |
> | LNS (Shaw, 1998)  | 45.77 | 0.00% | 150s | 64.81 | 0.00% | 240s |
> | L2Seg-SYN-LNS (zero-shot transfer) | 44.35 | 3.10% | 150s | 64.28 | 0.82% | 240s |
> | L2Seg-SYN-LNS (re-trained) | **44.15** | **3.54%** | 150s | **64.15** | **1.02%** | 240s |
>
> Additionally, we have performed new experiments **generalizing our model to problems with larger sizes**. We kindly refer the reviewer to our response to **Weakness 3** and **Question 3**, where we demonstrate that benefits persist even when scaling to problem sizes beyond those seen during training.

---

> > ### Author Response · Authors · 2025-11-25
> > **Response to Reviewer L8xN (2/4)**
> >
> > > **W2:** The behavior of L2Seg appears conceptually closer to search-space restriction rather than genuine problem decomposition. Consequently, there is a risk that the restricted search could lead to premature convergence to local optima, preventing the discovery of globally superior solutions.
> > >
> >
> > Thanks for this valuable feedback.
> >
> > **While we agree that L2Seg can be viewed as a novel learning-based search-space restriction method, we regard this as a significant contribution rather than a limitation.** In fact, despite the risks of premature convergence to local optima, effective search space reduction is fundamental to modern optimization solvers to tackle the large-scale NP-hard problems. Both heuristic and exact solvers require effective search space reduction to achieve high performance. Examples include LKH-3 [5] (via $\alpha$-nearest reduction), LNS [1] (via bounded neighborhood reduction), and exact solvers like Gurobi [6] (via branch-and-bound reduction). Different from those hand-crafted ways of reduction, L2Seg advances this research line by replacing static heuristics with data-driven stability estimation. We argue that this also constitutes genuine problem decomposition: by learning to identify and fix stable edges, L2Seg naturally decomposes the original large-scale graph into smaller, independent sub-problems that are significantly easier to optimize. Thus, L2Seg functions as both a novel search-space restriction method and an effective decomposition framework.
> >
> > **Secondly, L2Seg can continue optimizing for a long time budget.** Empirically, in Figure 5, the search curves continue improving throughout the time limit without plateauing, indicating that the method maintains exploration capability rather than converging prematurely. The curves demonstrate steady progress rather than the flat convergence typical of local optima. Moreover, we conducted additional experiments extending the runtime from 5 minutes to 1 hour on the first 25 instances (limited by computational resources), comparing L2Seg-SYN-L2D against HGS. The extended runtime results demonstrate that L2Seg continues to find improvements over the full hour, ultimately outperforming HGS on CVRP2k and CVRP5k. This continued improvement over extended periods confirms that our method avoids premature convergence and maintains effective exploration.
> >
> > | Method | CVRP1k |  |  | CVRP2k |  |  | CVRP5k |  |  |
> > | --- | --- | --- | --- | --- | --- | --- | --- | --- | --- |
> > |  | Obj. | Gap | Time | Obj. | Gap | Time | Obj. | Gap | Time |
> > | HGS | 40.05 | 0.00% | 1h | 54.62 | 0.00% | 1h | 121.94 | 0.00% | 1h |
> > | L2Seg-SYN-L2D (short) | 41.23 | 2.95% | 2.5m | 56.05 | 2.62% | 4.1m | 121.87 | -0.06% | 5.1m |
> > | L2Seg-SYN-L2D (long) | 40.35 | 0.75% | 1h | 54.23 | -0.71% | 1h | 120.24 | -1.39% | 1h |
> >
> > **Why does L2Seg address premature convergence?** L2Seg employs a **dynamic search space restriction mechanism.** At each iteration, L2Seg predicts stable edges from scratch based on the current solver state. This design inherently prevents over-restriction: edges incorrectly classified as stable in one iteration can be identified as unstable and 'unlocked' in subsequent steps, allowing the solver to escape local optima and correct previous errors. We conducted ablation studies to empirically verify this. We compared three configurations on CVRP2k with LNS as the subsolver: (1) static search space restriction where unstable edges are predicted only once at the beginning, (2) our standard dynamic approach with predictions of 1000 search steps limit, and (3) reduced frequency with predictions of 2000 search steps (effectively doubling the search steps between re-evaluations). The results reveal critical insights: Static search space restriction indeed suffers from premature convergence, performing worse than the baseline LNS solver (1.73% vs 0.73% gap). This confirms the risk you identified. However, with dynamic re-evaluation, performance dramatically improves to -1.96% gap, surpassing the baseline. The similar performance between 1000 and 2000 step intervals (-1.96% vs -1.91%) demonstrates the robustness of our dynamic approach. The key is not the exact frequency but rather the ability to adapt predictions as the solution evolves.
> >
> > | Configuration | Search Step Limit per Step | Obj. | Gap (w.r.t. HGS) | Time |
> > | --- | --- | --- | --- | --- |
> > | LNS (baseline) | - | 57.62 | 0.73% | 4.0m |
> > | Static (predict once) | - | 58.19 | 1.73% | 4m |
> > | Dynamic (standard) | 1000 | 56.08 | -1.96% | 4.1m |
> > | Dynamic (reduced freq.) | 2000 | 56.11 | -1.91% | 4m |

---

> > > ### Author Response · Authors · 2025-11-25
> > > **Response to Reviewer L8xN (3/4)**
> > >
> > > > **W3:** The model is trained and evaluated on problem instances of the same size distribution, leaving it unclear whether L2Seg generalizes to unseen problem scales. If it fails to generalize, collecting sufficient training data for larger instances could become computationally prohibitive, limiting the practical applicability of the approach.
> > > >
> > > > **Q3:** Does L2Seg generalize to problem sizes that were not seen during training? If not, how feasible is it to train the model for significantly larger problem instances?
> > > >
> > >
> > > Thanks for raising this important concern.
> > >
> > > We conducted zero-shot generalization experiments using models trained on CVRP2k and tested on CVRP5k (2.5x larger). The results demonstrate strong generalization capabilities. Even under zero-shot transfer, L2Seg maintains significant advantages over baseline methods, with gaps of -1.52% to -1.81% compared to HGS. While there is some performance drop compared to in-distribution testing, the benefits remain substantial.
> > >
> > > |  | CVRP5k |  |  |
> > > | --- | --- | --- | --- |
> > > |  | Obj.↓  |  Gap↓ | Time↓ |
> > > | HGS | 126.20  | 0.00%  | 5m |
> > > | LKH-3 |  175.70  | 39.22%  | 2.5m |
> > > | LNS | 126.58  | 0.30%  | 5.0m |
> > > | L2D | 126.48 | 0.22% |  5.3m |
> > > | L2Seg-SYN-LKH-3 | 122.34 | -3.16% |  5.1m |
> > > | L2Seg-SYN-LKH-3 (zero-shot generalization) | 128.24 | 1.62% | 5m |
> > > | L2Seg-SYN-LNS | 121.96 | -3.48% | 5.1m |
> > > | L2Seg-SYN-LNS (zero-shot generalization) | 123.92 | -1.81% | 5m |
> > > | L2Seg-SYN-L2D | 121.87 | -3.55% | 5.1m |
> > > | L2Seg-SYN-L2D (zero-shot generalization) | 124.65 | -1.52% | 5m |
> > >
> > > In Appendix E.2, we further validate L2Seg's generalization capabilities through zero-shot testing on the CVRPLib benchmark, which includes realistic routing instances ranging up to 10,000 nodes. Our results demonstrate that L2Seg-SYN-LNS achieves the best performance among all learning-based methods on these challenging real-world instances. Notably, L2Seg-SYN-LNS not only achieves superior solution quality (6.9% vs 7.1% gap on Set-X, 12.0% vs 13.2% gap on Set-XXL) but does so with substantially reduced computational time (600s vs >1 hour for UDC-x250). These results on instances up to 10k nodes, with fundamentally different spatial distributions than our synthetic training sets, provide strong evidence of L2Seg's exceptional generalization across both problem scales and distributions.
> > >
> > > **Why does our L2Seg potentially scale/generalize well across sizes?** We list several reasons.
> > >
> > > - **Batched Route Processing**: During embedding, L2Seg splits problems into adjacent route-pairs and uses batch processing, avoiding memory issues and excessive computation regardless of problem scale.
> > > - **Scale-Invariant Local Patterns:** Crucially, the size difference between adjacent route-pairs remains relatively constant across problem scales. While a 5k-node problem is 2.5x larger than a 2k-node problem, the individual route-pairs maintain similar sizes (determined by vehicle capacity, not problem scale). This scale invariance enables excellent transfer learning; patterns learned on 2k instances should transfer effectively to 5k+ instances.
> > > - **Solver Flexibility**: For very large problems, one can employ efficient linear-time solvers (e.g., FILO [6]) as the lookahead solver, enabling L2Seg training on problems with 10k+ nodes. Recent linear-time solvers FILO handle 10k+ node instances efficiently (5 minutes for 12k-node instances), enabling L2Seg training on problems well beyond current benchmarks. Based on competition results, we estimate running times for this type of solvers:
> > >     - 10k nodes: ~5-10 minutes
> > >     - 20k nodes: ~20-30 minutes
> > > - **Training pipeline scales efficiently.** Empirically, our training pipeline scales efficiently, requiring under 3 days for CVRP5k. We project that scaling to 10k instances would take approximately 6 days, which is still feasible.
> > >
> > > |  | Data Collection | Training time NAR | Training time AR | Total (parallel) |
> > > | --- | --- | --- | --- | --- |
> > > | CVRP1k | 6.3h | 6.8h |  13.7h  | 20.0h  |
> > > | CVRP2k | 10.4h | 10.4h  | 22.4h | 32.8h |
> > > | CVRP5k | 19.2h | 18.1h | 40.7h | 59.9h |
> > >
> > > ---
> > >
> > > > **W4 (Minor):** Typo found at line 130.
> > > >
> > > > **W5 (Minor):** References in lines 73–75 should be enclosed in parentheses.
> > > >
> > >
> > > Thank you for your careful review. These corrections improve the clarity of our presentation. We have corrected those typos as shown in red.
> > >
> > > ---
> > >
> > > > **Q1:** How do the authors precisely define “iterative solvers” in the context of this paper?
> > > >
> > >
> > > Thanks for raising this question. We follow the definitions used in [1], [2] and [3], where **Iterative solvers** are optimization algorithms that start with an initial solution and repeatedly modify it through a series of local changes to improve the objective function value. We will supplement this definition in Appendix A.4.

---

> > > > ### Author Response · Authors · 2025-11-25
> > > > **Response to Reviewer L8xN (4/4)**
> > > >
> > > > > **Q2:** The paper states that “a large portion of the solution remains stable.” Does this mean that most parts of the solution are not updated because they do not significantly contribute to objective improvement, or because only a small subset of solution elements is actually targeted for update by the iterative process?
> > > > >
> > > >
> > > > This is an excellent question! We attribute the observed stability primarily to the **first factor**: most parts of the solution are not updated because they have reached a state where no significant objective improvement can be found by the solver, not because they are excluded from the search. We support this with two reasons:
> > > >
> > > > 1. **Stable edges reflect local optimality**: Current iterative solvers (e.g., LKH-3 [4], HGS [6]) and the sub-solvers used in LNS [5] are designed to improve the entire solution even though they may leverage some search space reduction mechanism such as “candidate edge set”. While we cannot guarantee universality across all iterative solvers, for the subsolvers and baselines used in the papers (LKH-3 [4], LNS [5], L2D [5], and HGS [6]), the solver would actively evaluate every edge in the current solution for potential removal or rewiring (e.g., via k-opt moves) by exhaustively checking the potential gains of replacing the edge with all feasible new edges contained in the candidate edge set. Consequently, when we observe an edge remaining unchanged ("stable") during our look-ahead analysis, it does not imply that the edge was ignored for search. Rather, it implies that the solver attempted to improve it by evaluating alternatives but found that modifying the edge did not yield a superior objective value. Thus, stability reflects local optimality, not a lack of search coverage.
> > > > 2. **Empirical Evidence of Decreased Re-optimization Rate**. Our empirical data in Figure 1 strongly support this. We observe that the percentage of edges being re-optimized drops dramatically from ~40% in early stages to ~6% in later stages. If stability were merely due to the solver "ignoring" subsets of edges, this percentage would remain relatively constant. The observed decay indicates a convergence process: as the solution quality improves, fewer and fewer edges contribute to further gains, and the "stable" portion naturally grows.
> > > >
> > > > This observation is precisely what **motivates our L2Seg**. By identifying these stable segments that persist across iterations, we can produce meaningful problem decompositions that allow the subsolver to focus computational effort on the genuinely improvable portions of the solution.
> > > >
> > > > We thank the reviewer again for the insightful comments in shaping our paper!
> > > >
> > > > ---
> > > >
> > > > **References**
> > > >
> > > > [1] Aarts et al. "Local search in combinatorial optimization." Princeton University Press, 2003.
> > > >
> > > > [2] Lin et al. "An effective heuristic algorithm for the traveling-salesman problem." Operations Research, 1973.
> > > >
> > > > [3] Lourenço et al. "Iterated local search." Handbook of Metaheuristics, 2003.
> > > >
> > > > [4] Helsgaun et al, “An Extension of the Lin-Kernighan-Helsgaun TSP Solver for Constrained Traveling Salesman and Vehicle Routing Problems.” Technical Report, Roskilde University, 2017.
> > > >
> > > > [5] Li et al. "Learning to delegate for large-scale vehicle routing." *Neurips 2021*
> > > >
> > > > [6] Accorsi et al. A Fast and Scalable Heuristic for the Solution of Large-Scale Capacitated Vehicle Routing Problems. *Transportation Science*, 55(4), 832–856.

---

> > > > > ### Comment · Reviewer_L8xN · 2025-11-26
> > > > >
> > > > > Dear Authors,
> > > > >
> > > > > Thank you for the comprehensive additional experiments and the clear answers to my questions.
> > > > >
> > > > > The concern raised in W2 is largely resolved because the additional experiments provided in Response (2/4) show that restricting the search space using the proposed method helps improve the solutions rather than causing the search to stagnate. For the concerns raised in W3 and Q3, I consider them largely resolved, as the additional experiments and analyses in Response (3/4) empirically demonstrate the generalization capability to unseen problem sizes and provide insights into the characteristics of problems for which generalization is relatively easy.
> > > > > For Q1 and Q2, your responses have deepened my understanding of the assumptions underlying this work.
> > > > >
> > > > > However, I still have the concern raised in W1. This concern arises not so much from a mismatch between the training and test data, but rather from the fact that the construction of the stability labels depends on existing heuristic optimization methods. This concern may also be related to reviewer Kw4B’s comments and to the authors’ response to reviewer b73r’s Q2. It is important to consider whether the structure of the stability labels has low dependence on the choice of solver.

---

> ### Author Response · Authors · 2025-11-27
>
> We thank the reviewer for the prompt and detailed feedback! We appreciate your acknowledgment that our responses have addressed your concerns regarding W2, W3, Q1, Q2, and Q3, and we thank you for the thoughtful follow-up. We now fully understand your remaining concern regarding W1 and are glad for the opportunity to further clarify this point.
>
> ---
>
> **Regarding the structure of the stability labels:** We fully agree with you that this is a great research question. We hypothesize that stability labels are composed of two factors: (1) **Inherent Problem Structure** (e.g., edges common to the majority of local optima regardless of the solver), and (2) **Solver-Dependent Patterns** (edges preferred due to specific search biases). Following your suggestion to investigate the dependence of such a stability label on the choice of solver, we designed an experiment on CVRP1k to compare label similarity across different solvers. We generated labels using **HGS** (60s time limit) and **LKH-3** (1000 local search steps), comparing them against a "ground truth" generated by **LKH-3** with a much longer run (3000 steps). Each solver was run with 10 different seeds to remove randomness. Label similarity is defined as the percentage of overlapping stable edges.
>
> |  | HGS | LKH-3 (1000 Local Search Step) | LKH-3 (3000 Local Search Step) |
> | --- | --- | --- | --- |
> | Label Similarity | 78.3% | 85.1% | 100% (by default) |
>
> These results reveal two key insights: (1) Even fundamentally different solvers (HGS vs. LKH-3) share **78.3%** of stable edges. This indicates that the majority of the stability is contributed by inherent problem structure rather than solver artifacts; (2) More similar solvers produce more similar labels (LKH-3 variants: 85.1% similarity) compared to fundamentally different solvers (HGS vs LKH-3: 78.3%), suggesting some solver-specific patterns.
>
> We believe these results strongly support the design rationale of L2Seg. By training on data from a specific backbone (e.g., LKH), L2Seg captures **both** the universal inherent problem structure and the solver-specific patterns. **We argue this is beneficial: it allows the model to maximally reduce redundant calculations for that specific backbone,** achieving large speedups (2-7x). On the other hand, if the user prefers a more generalizable model applicable to boost various solvers that has low dependence on the choice of solver, we could train on the intersection of labels from multiple solvers to zero out the effects of solver-specific patterns and better capture purely universal patterns, as supported by the high intersection (78.3%) above. Furthermore, as noted in our response to **Reviewer b73r’s Q1**, L2Seg trained on LKH-3 data already demonstrates zero-shot generalization, boosting a different solver LNS’s performance by 1.86% under a 4-min solving time limit. This confirms that the model has learned robust structural features beyond simple solver overfitting.
>
> ---
>
> **Regarding distribution shift during inference:** To further investigate whether our L2Seg’s prediction accuracy would degrade when intermediate solutions shift during inference (since we are using supervised learning), we measured L2Seg's stable edge prediction accuracy at different iterative steps (1, 5, 10, 15) on CVRP2k:
>
> | Step | 1 out of 25 | 5 out of 25 | 10 out of 25 | 15 out of 25 |
> | --- | --- | --- | --- | --- |
> | L2Seg-SYN-LNS | 76.6% | 80.2% | 74.2% | 69.2% |
> | L2Seg-SYN-L2D | 77.1% | 76.1% | 74.8% | 68.5% |
>
> Our results indicate a gradual and manageable decline in accuracy: from \~77% at step 1 to \~69% at step 15, with accuracy remaining above 74% through the first 10 steps, where on average there are 25-30 iterative steps in total. This performance remains comparable to the training accuracy (\~82%, Figure 7), demonstrating that L2Seg maintains robustness despite the expected distribution shift. Critically, even with this slight degradation, the model successfully drives 2-7x speedups and 1.96%-3.55% solution quality improvements. We identify the mitigation of this distribution shift as a promising direction for future work, where techniques such as DAgger or Reinforcement Learning could be employed to further align the training and inference distributions. Nevertheless, we note that our contributions, including the principled FSTA decomposition framework with theoretical support, the proposed L2Seg (AR, NAR, SYN) models, and our extensive experiments, represent a solid contribution that is worth sharing with the community.
>
> ---
>
> We thank the reviewer again for the constructive and inspiring dialogue. We are more than happy to answer any further questions, if any. We hope our clarification may lead to a more positive assessment of our work. Thank you!

---

> > ### Comment · Reviewer_L8xN · 2025-11-28
> >
> > Thank you for the additional experiments and the detailed responses regarding the solver dependence of the labels and the analysis of distributional shift during inference. In my view, the experimental result showing that most stability labels constructed using solvers with different operating principles coincide supports the validity and reliability of the stability labels, which are a key component of the proposed method. As my concerns have now been largely resolved, I will update my score accordingly.

---

> > > ### Author Response · Authors · 2025-11-28
> > > **Thank You for Your Support!**
> > >
> > > We sincerely thank the reviewers for engaging with our rebuttal and for updating the scores accordingly. We appreciate your thoughtful questions and are glad to hear that the new experiments and clarifications greatly helped address your concerns. We will make sure to incorporate all our discussions into the final manuscript. Thank you again for your valuable guidance and support!

---

### Official Review · Reviewer_Kw4B · 2025-10-30

**Soundness:** 3
**Presentation:** 3
**Contribution:** 2
**Rating:** 4
**Confidence:** 3

**Summary:**

This paper proposes Learning-to-Segment (L2Seg), a learning-based framework that accelerates iterative heuristics for Vehicle Routing Problems (VRPs). The key idea is to detect stable and unstable segments in existing solutions, aggregate stable portions into hypernodes via a First-Segment-Then-Aggregate (FSTA) decomposition, and then re-optimize only the unstable parts.
The authors design three neural variants (NAR, AR, and SYN)  and demonstrate empirical speedups over baseline solvers such as LKH-3, LNS, and L2D, on both CVRP and VRPTW benchmarks.

**Strengths:**

- The empirical study is well executed, covering multiple VRP variants, backbone solvers, and problem scales (1k–5k). Ablation studies, oracle comparisons, and visual analyses provide convincing evidence that the proposed framework accelerates iterative search.

- The proposed framework is well documented, including pseudocode and architectural details. The authors have made strong efforts to ensure reproducibility and practical relevance.

**Weaknesses:**

- Limited conceptual novelty: the proposed framework can be viewed as a natural neural extension of existing decomposition-based heuristics. While the idea of stability is interesting, the framework essentially replaces a search space of LNS.
- Restricted theoretical contribution: the theoretical analysis is limited to the monotonicity of the FSTA reduction — that improving a reduced problem implies improving the original one. However, the paper lacks theoretical analysis regarding solution quality guarantees or learnability.
- Dependence on heuristic supervision: the training labels rely on the lookahead procedure with heuristic solvers. This raises concerns about the correctness of the label.

**Questions:**

- Are there any justifications of the label about stability? It is not entirely clear whether the notion of stability—as defined by differences between consecutive heuristic solutions—is reliable.
- Are there any baselines using LNS with ML-based neighborhood selection? If such a baseline exists, the efficiency of proposed method can be more convincingly positioned as an improved (or generalized) version of ML-enhanced LNS.

---

> ### Author Response · Authors · 2025-11-25
> **Response to Reviewer Kw4B (1/4)**
>
> We thank the reviewer for the thoughtful and constructive feedback. We appreciate your recognition that our well-executed and comprehensive empirical study provides convincing evidence to support our main claim: learning to segment to accelerate iterative solvers. We have carefully noted your concerns regarding conceptual novelty, theoretical contributions, and label reliability. Below, we address them in detail to clarify the significance of our contributions.
>
> ---
> > **W1:** Limited conceptual novelty: ... as a natural neural extension of existing decomposition-based heuristics. ..., the framework essentially replaces a search space of LNS.
>
> Thank you for the comment. Here, we appreciate the opportunity to clarify that there might be some misunderstanding of the significance and the applicability of our method.
>
> **Firstly, while we agree that L2Seg redefines the search space from a new perspective (based on stability), we regard this as a significant contribution rather than a limitation.** In fact, effective reduction is fundamental to modern optimization solvers. Both heuristic and exact solvers require effective search space reduction to achieve high performance. Examples include LKH-3 [5] (via $\alpha$-nearest reduction), LNS [1] (via bounded neighborhood reduction), and exact solvers like Gurobi [6] (via branch-and-bound reduction). Different from those hand-crafted ways of reduction, L2Seg advances this line of research by introducing a **novel way of learning-based stability estimation** to intelligently focus computational resources in a data-driven way, representing the state-of-the-art in learning-based reduction. Moreover, L2Seg offers general applicability: It is designed to be solver-agnostic, which is able to enhance most existing iterative methods (such as LKH and LNS) rather than competing with them; as demonstrated by the theoretical proofs and implementation details in Appendix B.1.5, it also extends to a broad spectrum of routing variants, including CVRP, VRPTW, VRPB, and 1-VRPPD.
>
> **Secondly, we clarify that the search space induced by our L2Seg is fundamentally different from any existing decomposition method.** L2Seg introduces a fundamentally different decomposition paradigm based on learned stability patterns rather than predefined neighborhoods, and can be applied to most iterative solvers, not limited to LNS, across many routing variants. Unlike LNS or other decompositions, L2Seg enables stable edge detection and search space restriction at the **edge level** across the entire graph, instead of predefined local regions or only at the sub-route level (searching or not searching whole sub-routes). We provide a detailed analysis of L2Seg against existing methods in Appendix C.1, including comparisons with LNS, Evolutionary Algorithms, Path Decomposition, and the previous ML-based L2D. Here, we highlight the comparisons against the LNS and ML-enhanced LNS method L2D [1].
>
> - **Compared with hand-crafted LNS**: (1) since LNS operates within a bounded local neighborhood, it isolates a specific region (e.g., 3–5 sub-routes) to destroy and rebuild, leaving the rest of the solution frozen. In contrast, our L2Seg operates **globally** across the entire solution. It can break existing edges and aggregate segments throughout all subroutes simultaneously, without predefined neighborhood boundaries. The modifications are distributed across the entire solution rather than confined to a local region, representing a fundamental departure from existing LNS. (2) We emphasize that such a flexible framework would not be possible without the proposed ML component, which is another core novelty and contribution. (3) Moreover, FSTA and LNS are complementary: FSTA can be applied on top of LNS, where LNS first selects a large neighborhood, then FSTA fixes stable edges globally within that selected region.
> - **Compared with ML-based LNS**: (1) Unlike L2D's sub-route level operation, our method detects unstable edges both within and across sub-routes, enabling more global and flexible decomposition. (2) It optimizes beyond localized neighborhoods by identifying improvements that span multiple distant regions simultaneously. (3) It reduces the size of sub-routes by aggregating stable segments into hypernodes, whereas L2D only reduces the number of sub-routes per iteration. This segment-level aggregation allows more adaptive and coarse-grained reduction, offering higher efficiency and solution quality while remaining complementary to L2D.
>
> As shown in Figure 5 (e.g., on CVRP2K), our proposed L2Seg achieves speedups of **7.4x** over the existing LNS method and **2.1x** over the L2D method. These results validate L2Seg as a unique contribution with distinct mechanisms from LNS. We hope the above discussions are helpful to clarify that our L2Seg is not an extension of existing LNS, and achieving SOTA performance through this novel, solver-agnostic method represents a meaningful contribution to the field.

---

> ### Author Response · Authors · 2025-11-25
> **Response to Reviewer Kw4B (2/4)**
>
> > **W2:** Restricted theoretical contribution: the theoretical analysis is limited to the monotonicity of the FSTA reduction — that improving a reduced problem implies improving the original one. However, the paper lacks theoretical analysis regarding solution quality guarantees or learnability.
> >
>
> Thank you for these valuable suggestions!
>
> Firstly, we would like to clarify that, other than the **monotonicity** as mentioned by the reviewer, we also have another important theoretical contribution in proving the **feasibility** of FSTA decomposition across multiple VRP variants (CVRP, VRPTW, VRPB, and 1-VRPPD). We emphasize that the most non-trivial contribution lies primarily in the **feasibility** proof part of FSTA. For example, computing time windows for aggregated problems in VRPTW is complex, as it requires careful recursive formulation (Equation 3) to maintain temporal feasibility throughout the aggregation process. By establishing both feasibility and monotonicity theorems, we are the first to rigorously formalize such a decomposition technique with theoretical support and have extended this decomposition paradigm to broader routing variants. This creates a systematic framework to extend this pipeline to broader routing variants, moving beyond ad-hoc heuristics.
>
> Secondly, we seek the understanding of the reviewer that many highly impactful neural combinatorial optimization papers [1, 2, 3] do not provide any theoretical contribution. The field is currently at a stage where demonstrating strong empirical results is paramount, largely because deriving tight theoretical guarantees for complex decomposition methods integrated with deep neural networks remains a significant open challenge.
>
> Nevertheless, to address the concern regarding L2Seg’s potential solution quality guarantees and learnability, we provide some in-depth analyses as follows.
>
> **Learnability:**
>
> - Figure 1 shows that the percentage of reoptimized edges drops from ~40% to just 6% as optimization progresses, particularly in large subtours (high capacity $C$). This increasing dominance of stable edges validates the learnability.
> - Figure 8 visualizes unstable edge distribution patterns: (1) the number of unstable edges consistently decreases over time; and (2) instability concentrates within route interiors, while edges at route boundaries exhibit higher stability. We conducted additional experiments quantifying how "interior" stable edges are. We examined the 10 nearest neighbors of each node and calculated the percentage of these neighbors belonging to the same subroute. A node is classified as stable if both edges connecting to it are stable. Analyzing 1,000 cases using LKH-3 as the subsolver (shown in the table above), we found that the 14.5% difference (75.8% vs 61.3%) clearly shows that stable nodes are more interior to their routes. This demonstrates two distinct patterns of edge stability in iterative solver LKH-3, providing strong evidence for the **learnability of L2Seg,** where our neural networks can potentially capture these spatial patterns through the proposed enhanced input features.
>
> |  | Unstable nodes | Stable Nodes |
> | --- | --- | --- |
> | Avg Percentages | 89.4% | 10.6% |
> | Avg Percentages of Nearest Neighbors in the Same Subroute | 61.3% | 75.8% |
>
> **Solution Quality:**
>
> - The solution quality may be affected by the prediction accuracy of the learned model. To quantify this, we conducted a 'look-ahead' sensitivity analysis with controlled levels of Recall and True Negative Rates (TNR). Table 5 reports the time required to achieve performance equivalent to our learned model on small-capacity CVRP2k instances. We evaluated scenarios ranging from a Perfect Oracle to imperfect configurations. While the Perfect Oracle demonstrates superior efficiency (39s vs. 241s for L2Seg-SYN), achieving Recall and TNR as high as 90% without oracle access is highly non-trivial. Crucially, in practical scenarios where simulated Recall and TNR drop to 70%, the oracle-based approach is significantly outperformed by our actual learned model, L2Seg-SYN-LNS (324s vs. 241s).  The strong performance of both perfect Oracle and Oracle with imperfect predictions indicates the solution quality guarantee potential of our pipeline.
>
> |  | Oracle (LNS) + perfect recall & TNR | Oracle (LNS) + 95% recall & 95% TNR | Oracle (LNS) + 90% recall & 90% TNR | Oracle (LNS) + 70% recall & 70% TNR | Ref (L2Seg-SYN-LNS) |
> | --- | --- | --- | --- | --- | --- |
> | Obj. (Sol quality) | 56.02 | 56.01 | 56.02 | 56.04 | 56.08 |
> | Time | 39s | 62s | 119s | 324s | 241s |
>
> We believe this empirical evidence, combined with our theoretical proofs of feasibility and monotonicity, comprehensively establishes the validity and robustness of our framework.

---

> ### Author Response · Authors · 2025-11-25
> **Response to Reviewer Kw4B (3/4)**
>
> > **W3:** Dependence on heuristic supervision: the training labels rely on the lookahead procedure with heuristic solvers. This raises concerns about the correctness of the label.
> >
> > **Q1:** Are there any justifications of the label about stability? It is not entirely clear whether the notion of stability—as defined by differences between consecutive heuristic solutions—is reliable.
>
> This is an insightful question! To address this systematically, we first clarify the data collection process and the definition of an "iteration" to resolve any ambiguity.
>
> **Why our label is reliable.** We define an "iterative step $t$" as the $t$-th invocation of the backbone solver rather than a single heuristic move. **Within each invocation, the solver performs a full round of local search subject to a fixed budget (e.g., time limit or number of steps) and returns a local optimal solution**. Moreover, to reduce the randomness, **we execute the look-ahead process multiple times using different random seeds and select the union of all detected unstable edges**. This further ensures a high-recall labeling process, minimizing the risk of accidentally freezing edges that might be improvable under randomness.
>
> **Empirical Validation:** Accordingly, we assessed label reliability on CVRP2k (using LNS) by varying local search budgets (100-3000) and random seeds (1-5). We measured consistency against a high-fidelity reference (3000 updates, 5 seeds) using two metrics: direct label similarity and the final solution quality of L2Seg models trained on each configuration. As shown below, while lower search limits or single seeds naturally yield slightly lower fidelity, our sensitivity analysis demonstrates that unstable edge labels exhibit **strong convergence**. Specifically, **92.9%** of stable edges remain identical when tripling the search limit, and **95.9%** persist when increasing from 3 to 5 seeds. Crucially, this consistency translates to downstream effectiveness: models trained on these varying labels achieve nearly the same performance (from -1.91% to -1.98%). This confirms that our label generation pipeline is robust and stable.
>
> | Local Search Limit per Iterative Step | Label Similarity | Obj. | Gap (w.r.t. HGS) | Time |
> | --- | --- | --- | --- | --- |
> | 100 | 77.5% | 57.22 | -0.00% | 4m |
> | 500 | 90.1% | 56.32 | -1.54% | 4m |
> | 1000 (used in the paper) | 92.9% | 56.08 | -1.96% | 4.1m |
> | 2000 | 93.4% | 56.11 | -1.91% | 4m |
> | 3000 | 100% (by default) | 56.07 | -1.98% | 4m |
>
> | # of Seeds | Label Similarity | Obj. | Gap (w.r.t. HGS) | Time |
> | --- | --- | --- | --- | --- |
> | 1 | 87.2% | 56.25 | -1.66% | 4m |
> | 3 (used in the paper) | 95.9% | 56.08 | -1.96% | 4.1m |
> | 5 | 100% (by default) | 56.10 | -1.92% | 4.1m |
>
> ---
>
> **Why is dynamic stability preferred?** We then discuss why we define 'stable edges' as a **dynamic stability** relative to consecutive iteration steps, rather than a static alignment with the fixed global optimal solution. The rationale is threefold:
>
> 1. **Tractability:** according to [4](they proved that finding improving k-opt moves is PLS-complete), identifying globally optimal edges is computationally intractable (NP-hard).  Therefore, it's intractable to identify stable edges w.r.t. the optimal solution precisely.
> 2. **Robustness:** Dynamic prediction prevents premature convergence. Unlike static methods that permanently fix edges, our approach re-evaluates stability at every step. This allows the model to correct previous prediction errors in subsequent iterations, preventing the solver from getting trapped in local optima due to over-restriction.
> 3. **Search Alignment:** Iterative solvers operate locally on the current solution. The dynamic stability could better capture such local properties.
>
> **Regarding the reliance on heuristic supervision:** while we acknowledge the reliance on heuristic supervision, our pipeline is highly efficient: label generation requires only **6.3–19.2 hours**, and total training (20–60 hours) consumes substantially fewer resources than typical Reinforcement Learning (RL) approaches.
>
> |  | Data Collection | Training time NAR | Training time AR | Total (parallel) |
> | --- | --- | --- | --- | --- |
> | CVRP1k | 6.3h | 6.8h | 13.7h | 20.0h |
> | CVRP2k | 10.4h | 10.4h | 22.4h | 32.8h |
> | CVRP5k | 19.2h | 18.1h | 40.7h | 59.9h |

---

> ### Author Response · Authors · 2025-11-25
> **Response to Reviewer Kw4B (4/4)**
>
> > **Q2:** Are there any baselines using LNS with ML-based neighborhood selection? If such a baseline exists, the efficiency of proposed method can be more convincingly positioned as an improved (or generalized) version of ML-enhanced LNS.
> >
>
> Thanks for raising this question! Note that the L2D [1] is indeed an ML-enhanced LNS method, which serves as both a backbone solver and baseline in our paper. Notably, L2Seg is more accurately positioned as a data-driven decomposition paradigm based on learned stability patterns that can augment various solvers, rather than just an improved ML-enhanced LNS. We kindly refer the reviewer to refer to our response to W1 for details.
>
> Regarding empirical comparison (the table below), we list two key conclusions:
>
> 1. **Superior enhancement over LNS**: While L2D represents ML-guided LNS, and L2Seg-SYN-LNS also augments LNS, our approach demonstrates significantly better improvement. L2Seg-SYN methods achieve -5.60% to 0.39% gaps compared to HGS (vs. L2D's -5.38% to 2.11% gaps) across different problem sizes and types, consistently outperforming L2D, the SOTA ML-enhanced LNS methods, across all problem scales.
> 2. **Universal applicability**: L2Seg can also be leveraged to further accelerate the ML-based L2D method. Specifically, L2Seg-SYN-L2D achieves additional improvements of 2.04%-3.77% on CVRP and 0.22%-0.84% on VRPTW. This shows that our edge-level stability learning provides orthogonal benefits that complement L2D designs.
>
> | Methods | CVRP1k |  |  | CVRP2k |  |  | CVRP5k |  |  |
> | --- | --- | --- | --- | --- | --- | --- | --- | --- | --- |
> |  | Obj.↓ | Gap↓ | Time↓ | Obj.↓ | Gap↓ | Time↓ | Obj.↓ | Gap↓ | Time↓ |
> | HGS (Vidal, 2022) | 41.20 | 0.00% | 5m | 57.20 | 0.00% | 5m | 126.20 | 0.00% | 5m |
> | LNS (Shaw, 1998) | 42.44 | 3.01% | 2.5m | 57.62 | 0.73% | 4.0m | 126.58 | 0.30% | 5.0m |
> | L2D (Li et al., 2021) | 42.07 | 2.11% | 2.5m | 57.44 | 0.42% | 4.2m | 126.48 | 0.22% | 5.3m |
> | L2Seg-SYN-LNS | 41.36 | 0.39% | 2.5m | 56.08 | -1.96% | 4.1m | 121.96 | -3.48% | 5.1m |
> | L2Seg-SYN-L2D | 41.23 | 0.07% | 2.5m | 56.05 | -2.01% | 4.1m | 121.87 | -3.55% | 5.1m |
> | Methods | **VRPTW1k** |  |  | **VRPTW2k** |  |  | **VRPTW5k** |  |  |
> |  | Obj.↓ | Gap↓ | Time↓ | Obj.↓ | Gap↓ | Time↓ | Obj.↓ | Gap↓ | Time↓ |
> | HGS (Vidal, 2022) | 90.35 | 0.00% | 2m | 173.46 | 0.00% | 4m | 344.2 | 0.00% | 10m |
> | LNS (Shaw, 1998) | 88.12 | -2.47% | 2m | 165.42 | -4.64% | 4m | 338.5 | -1.66% | 10m |
> | L2D (Li et al., 2021) | 88.01 | -2.59% | 2m | 164.12 | -5.38% | 4m | 335.2 | -2.61% | 10m |
> | L2Seg-SYN-LNS | 87.31 | -3.56% | 2m | 163.94 | -5.49% | 4m | 334.1 | -2.93% | 10m |
> | L2Seg-SYN-L2D | 87.25 | -3.43% | 2m | 163.74 | -5.60% | 4m | 333.4 | -3.14% | 10m |
>
> We thank the reviewer again for the insightful comments in shaping our paper!
>
> ---
> **References**
>
> [1] Li et al. "Learning to delegate for large-scale vehicle routing." *Neurips 2021.*
>
> [2] Fu et al. “A hierarchical destroy and repair approach for solving very large-scale traveling salesman problem.” *AAAI 2025.*
>
> [3] Hottung et al. “Neural Deconstruction Search for Vehicle Routing Problems.” *TMLR 2025.*
>
> [4] Lenstra at al. "Complexity of vehicle routing and scheduling problems." *Networks, 11(2), 221-227.*
>
> [5] Helsgaun et al, “An Extension of the Lin-Kernighan-Helsgaun TSP Solver for Constrained Traveling Salesman and Vehicle Routing Problems.” Technical Report, Roskilde University, 2017.
>
> [6] Gurobi Optimization, LLC. "Gurobi Optimizer Reference Manual." 2023. URL: [https://www.gurobi.com](https://www.gurobi.com/)

---

> > ### Comment · Reviewer_Kw4B · 2025-11-28
> > **Official Comment by Reviewer Kw4B**
> >
> > Thank you for clarifying the points I had misunderstood.
> > The authors’ response resolved the main issues underlying my concerns.
> > I appreciate the detailed explanations regarding feasibility and the comparisons with ML-based LNS baselines.
> >
> > Overall, I place value on the empirical contributions demonstrated in the paper, and I will raise my score from 4 to 6.
> > For future work, I would be interested in a more theoretical analysis of stability.

---

> > > ### Author Response · Authors · 2025-11-28
> > > **Thank You For Your Support!**
> > >
> > > We sincerely thank the reviewer for recognizing our empirical contributions and additional clarifications. Thank you for raising the score to support the acceptance of our paper! We also share your interest in a theoretical analysis of stability. While our current focus was empirical, we believe such analysis is a valuable next step. We will expand our discussion section to outline potential ideas and insights based on both our exchange and the discussions with other reviewers. Thank you once again for your score increase and support!

---

### Official Review · Reviewer_b73r · 2025-10-30

**Soundness:** 3
**Presentation:** 3
**Contribution:** 3
**Rating:** 6
**Confidence:** 4

**Summary:**

This paper aims to address the problem of redundant computation in iterative VRP solvers. The authors observe that during the iterative search process, a large portion of the solution stabilizes and no longer changes, yet the solver still consumes computational resources on these stable parts. To address this, the paper proposes FSTA (First-Segment-Then-Aggregate), a formalized decomposition framework with a full theoretical proof, to identify stable segments in the solution, aggregating their nodes into fixed hypernodes and then focusing the search only on the reduced unstable parts. Furthermore, this paper proposes L2Seg (Learning-to-Segment), a novel neural network framework for segmentation identification, including network architecture, training, and inference processes. Empirical results show that L2Seg can achieve speedups of 2 to 7 times for existing classical and learning-based solvers.

**Strengths:**

1. The paper's proposed method of learning to identify and freeze stable segments to accelerate iterative search is intuitive and novel. The proposed method is promising and achieves SOTA performance on most of the testing cases.
2. The FSTA framework is technically sound and empirically robust. It is formalized, and the authors provide theoretical proofs of its feasibility and monotonicity across various VRP variants (CVRP, VRPTW, VRPB, etc.).
3. The authors tested L2Seg on three different and representative backbone solvers (LKH-3, LNS, L2D), demonstrating its flexibility and versatility. Ablation experiments strongly support the necessity of the learned component (compared to stochastic FSTA).

**Weaknesses:**

1. The L2Seg-SYN process seems a bit complicated. It is unclear how much time is consumed by the L2Seg-SYN prediction step. If the L2Seg-SYN prediction step itself is costly, the 2x-7x speedup may only be noticeable over long runs.
2. While L2Seg has been successfully applied to LKH-3, LNS, and L2D, Appendix B.1.4 mentions that applying it to HGS (another top-level solver) requires modifying the HGS source code, which is left for future work. This is a reasonable limitation, but it means that L2Seg is not "plug and play" for all iterative solvers.

**Questions:**

1. Would better look-ahead heuristics lead to a better L2Seg model and ultimately better solution quality?
2. How does the time spent on the L2Seg-SYN prediction step compare to the time spent running the backbone solver in a single iteration?

Minor:

3. Some double quotes are different, such as the one in line 252.
4. Lines 47-48 write: FSTA identifies stable segments and then aggregates them as fixed hypernodes. But lines 146-147 write: FSTA segments the VRP solutions by identifying unstable portions, and then groups them into hypernodes. It seems there is a typo.

---

> ### Author Response · Authors · 2025-11-25
> **Response to Reviewer b73r (1/2)**
>
> Thank you for your support of our paper! We greatly appreciate your recognition of our technical soundness with formal proofs across VRP variants, as well as L2Seg's flexibility demonstrated through three distinct backbone solvers and the effectiveness of our ablation studies validating the learned components. We hope our responses throughout this review process will address the remaining concerns or questions about our work.
>
> ---
>
> > **W1:** The L2Seg-SYN process seems a bit complicated. It is unclear how much time is consumed by the L2Seg-SYN prediction step. If the L2Seg-SYN prediction step itself is costly, the 2x-7x speedup may only be noticeable over long runs.
> >
> > **Q2:** How does the time spent on the L2Seg-SYN prediction step compare to the time spent running the backbone solver in a single iteration?
>
> Thank you for your valuable feedback. First, we clarify that L2Seg provides dynamic predictions at each iteration rather than making a single static prediction. Specifically, L2Seg generates new stable edge predictions based on the current solution at every iteration step, followed by calling the backbone solver to improve the reduced problem for a few local search steps.
>
> While L2Seg-SYN integrates both NAR and AR models, which may increase pipeline complexity, we also present simpler alternatives in Table 1 (L2Seg-NAR and L2Seg-AR) that use only one model type and still outperform backbone solvers. L2Seg-SYN achieves the best performance as a reasonable combination of both approaches. Additionally, the training process is quite manageable: less than 3 days total (under 1 day for data collection and 1.5 days for training both models), which doesn't significantly increase overall complexity.
>
> Regarding the overhead of our L2Seg prediction, we measured the overhead across CVRP and VRPTW instances ranging from 1k to 5k nodes in the Table below. Even with our most complex model (L2Seg-SYN), the overhead consistently remains **below 10%** of the total iteration time (ranging from 7.2% to 9.6%). This indicates that the overhead scales efficiently and predictably. Generally speaking, the overhead primarily stems from neural network inference, which is driven by two factors: input data size and network call frequency. Regarding the former, L2Seg employs a **Batched Sub-Route Processing** design. During embedding, we split problems into adjacent route-pairs and use batch processing. This avoids memory bottlenecks and ensures that inference time scales efficiently for large-scale cases. Regarding the latter, the frequency of calling L2Seg is a tunable hyperparameter.
>
> **However, even though the overhead is manageable and adjustable, we emphasize that overhead may be viewed for reference rather than as a pure performance indicator.** For example, while L2Seg-SYN incurs the highest overhead (7.2%-9.6%) compared to the simpler variants such as L2Seg-NAR (4.2-6.5%), L2Seg-SYN achieves the highest solution quality, where better guidance from NN outweighs the additional overhead. In another example, for a larger-scale problem, the overhead increases due to larger input data. However, the absolute benefits also tend to increase with problem size, since larger problems contain more redundant computations that can be eliminated. This creates a scaling dynamic: even though both overhead and benefits increase with problem scale, the speedup ratios may remain consistent or even improve for larger instances (as shown in our 2-7x speedups across different scales).
>
> |Method|CVRP1k|||CVRP2k|||CVRP5k|||
> |---|---|---|---|---|---|---|---|---|---|
> ||NN|Total|%|NN|Total|%|NN|Total|%|
> |NAR-LNS|0.38s|8.4s|4.5|0.62s|11.4s|5.4|0.80s|12.9s|6.2|
> |AR-LNS|0.63s|9.8s|6.4|0.90s|10.8s|8.3|1.24s|13.9s|8.9|
> |SYN-LNS|0.76s|10.5s|7.2|1.14s|12.5s|9.1|1.33s|14.1s|9.4|
> ||**VRPTW1k**|||**VRPTW2k**|||**VRPTW5k**|||
> |NAR-LNS|0.37s|8.9s|4.2|0.61s|10.4s|5.9|0.83s|12.8s|6.5|
> |AR-LNS|0.62s|9.1s|6.8|0.96s|11.9s|8.1|1.14s|12.4s|9.2|
> |SYN-LNS|0.76s|10.1s|7.5|0.97s|10.8s|9.0|1.36s|14.2s|9.6|
>
> Finally, to address the concern that L2Seg may not be beneficial for a short run time budget, we extracted speedup measurements at different time points from Figure 5 to quantify this consistency in the Table below. The search curves together with the table reveal several important insights:
>
> - **Benefits over short and long runs:** Even at 40 seconds (approximately 3-5 iterations), we observe 3.1-4.5x speedups. The speedup benefits are noticeable in both short and long runs.
> - **Sustained acceleration:** Speedups remain consistently high throughout the search (2.8-8.1x)
> - **No degradation:** Benefits persist or even increase over time, confirming no premature convergence
>
> | Time | 40s  | 60s | 80s | 100s | 140s | 200s | 250s |
> | --- | --- | --- | --- | --- | --- | --- | --- |
> | Speed-up of L2Seg-SYN-LKH-3 (CVRP2k) | 4.5x | 7.9x | 7.3x | 8.1x | 7.1x | - | - |
> | Speed-up of L2Seg-SYN-LKH-3 Speed-up (CVRP5k) | 3.1x | 3.3x | 3.4x | 2.8x | 3.5 | 3.9x | 4.2x |

---

> ### Author Response · Authors · 2025-11-25
> **Response to Reviewer b73r (2/2)**
>
> > **W2:** While L2Seg has been successfully applied to LKH-3, LNS, and L2D, Appendix B.1.4 mentions that applying it to HGS (another top-level solver) requires modifying the HGS source code, which is left for future work. This is a reasonable limitation, but it means that L2Seg is not "plug and play" for all iterative solvers.
> >
>
> Thanks for raising this concern. While it's true that L2Seg requires some adaptations, we believe this limitation is not particularly significant in practice. The required modifications are typically straightforward (primarily adding edge-fixing implementation during solution construction), which is a standard capability that most modern VRP solvers already support or can easily implement. Moreover, the three solvers we've already integrated (LKH-3, LNS, and L2D) represent diverse optimization paradigms, demonstrating that our approach is adaptable across different solver architectures. The fact that we achieved successful integration with minimal modifications across these diverse approaches suggests that extending to other solvers like HGS is more a matter of engineering effort than fundamental incompatibility. We thus leave this as promising
> future work, given the limited time and GPU resources for training available during the short rebuttal period.
>
> > **Q1:** Would better look-ahead heuristics lead to a better L2Seg model and ultimately better solution quality?
> >
>
> That’s a good observation! To address this comprehensively, we analyze "higher quality look-ahead" through two distinct dimensions: (1) Increasing Look-Ahead Accuracy (via a larger local search step limit) and (2) using a Stronger Solver to generate data.
>
> We first analyzed the effect of varying look-ahead solver strength by adjusting the local search limit. The results show that weak look-ahead solvers (limit=100) yield poor performance, but the benefits plateau beyond a certain strength (1000-3000 limits achieve similar -1.91% to -1.98% gaps). This suggests that while a minimum quality threshold is necessary for effective, stable edge identification, excessive look-ahead computation yields no additional gains. The key is achieving sufficient accuracy in stability prediction rather than perfect optimization.
>
> | Local Search Limit per Iterative Step | Obj. | Gap (w.r.t. HGS) | Time |
> | --- | --- | --- | --- |
> | 100 | 57.22 | -0.00% | 4m |
> | 500 | 56.32 | -1.54% | 4m |
> | 1000 (used in the paper) | 56.08 | -1.96% | 4.1m |
> | 2000 | 56.11 | -1.91% | 4m |
> | 3000 | 56.07 | -1.98% | 4m |
>
> We further investigated whether employing a higher-performance solver (e.g., LNS) for data collection could improve inference for a lower-performance solver (e.g., LKH-3).  Interestingly, as shown in the table below, we discovered that **optimal performance requires synchronization** (using the same iterative solver for both look-ahead during training and testing). This is likely because different solvers exhibit distinct optimization patterns and search trajectories. While synchronized models achieve the best performance (-1.45% to -1.96% gaps), we note that cross-validated models still provide substantial benefits over baselines (e.g., -1.13% gap vs. 0.73% baseline). This demonstrates that L2Seg potentially captures robust, transferable stability patterns even when the solver architectures are mismatched.
>
> |**Configuration**|Model Training|Obj.|Gap (w.r.t. HGS)|Time|
> |-|-|-|-|-|
> |LKH-3 |-|57.9|1.29%|11.4m|
> |LNS|-| 57.62 | 0.73% |4.0m|
> |L2Seg-SYN-LKH-3(synchronized model)|LKH-3 labels|56.37|-1.45%|4.4m|
> |L2Seg-SYN-LNS (synchronized model)|LNS labels|56.08|-1.96%|4.1m|
> |L2Seg-SYN-LKH-3 (Cross-validated)|LNS labels|57.88| 1.12% | 4.0m |
> |L2Seg-SYN-LNS (Cross-validated) | LKH-3 labels | 56.55 | -1.13% | 4.0m |
>
> **Key Insights:**
>
> 1. Better look-ahead heuristics generally improve L2Seg performance, but with no additional benefits beyond a quality threshold
> 2. Synchronization between training and deployment solvers maximizes benefits
> 3. Even with solver mismatch, L2Seg maintains meaningful improvements, confirming the generality of learned stability patterns
>
> > **Q.Minor.1**: Some double quotes are different, such as the one in line 252.
> >
> > **Q.Minor.2**: Lines 47-48 write: FSTA identifies stable segments and then aggregates them as fixed hypernodes. But lines 146-147 write: FSTA segments the VRP solutions by identifying unstable portions, and then groups them into hypernodes. It seems there is a typo.
> >
>
> Thank you for your detailed review! These corrections improve the clarity of our presentation.
> For Q.Minor.1, we have corrected all inconsistent quotation marks throughout the paper to ensure proper formatting.
> For Q.Minor.2, we have revised line 147 to: "FSTA segments the VRP solutions by identifying unstable portions, then groups the stable segments into hypernodes." This now correctly aligns with lines 47-48 and accurately describes our methodology.
>
> ---
>
> We thank the reviewer again for the insightful comments in shaping our paper!

---

### Official Review · Reviewer_SSqq · 2025-11-01

**Soundness:** 3
**Presentation:** 3
**Contribution:** 3
**Rating:** 6
**Confidence:** 4

**Summary:**

This paper introduces Learning-to-Segment (L2Seg), a novel learning-guided framework designed to accelerate iterative solvers for large-scale VRPs. It first formalizes a First-Segment-Then-Aggregate (FSTA) decomposition technique, which identifies stable segments in a solution and aggregates them into hypernodes, thereby reducing the problem size for re-optimization. It also employs neural models to predict unstable edges. The paper presents three variants of L2Seg: a non-autoregressive (NAR) model for global prediction, an autoregressive (AR) model for local precision, and a synergized (SYN) model that combines their strengths. Extensive experiments on CVRP and VRPTW show that L2Seg accelerates state-of-the-art solvers by 2x to 7x while maintaining or improving solution quality.

**Strengths:**

(1) The paper is generally well-written and structured.

(2) The novel FSTA framework and the specific problem of learning to segment for decomposition are a fresh perspective.

(3) The FSTA framework is well-motivated, and its theoretical properties (feasibility and monotonicity) are formally proven for multiple VRP variants.

(4) Experiments are comprehensive, testing on large-scale problems, multiple backbone solvers (classic, neural, hybrid), and various VRP types, demonstrating robust performance and generalizability.

**Weaknesses:**

1. While the paper demonstrates broad applicability, a more explicit discussion of the boundaries of FSTA/L2Seg's effectiveness would be beneficial. For example, under what conditions (e.g., problem size, structure, solver type) might the overhead of segmentation and aggregation outweigh the benefits?
2. It is unclear that the boundary of the acceleration, as I notice that the HGS and LKH3 only run for a short time (5m and 10m). If the solving time extends, how will the acceleration benefit change in terms of both solution quality and time?
3. Some works are also based on hypergraphs, but this paper does not discuss [1, 2].


[1] A hierarchical destroy and repair approach for solving very large-scale travelling salesman problem. https://arxiv.org/pdf/2308.04639.

[2] Destroy and Repair Using Hyper-Graphs for Routing. https://ojs.aaai.org/index.php/AAAI/article/view/34018

**Questions:**

1. Can the author explain how to deal with the disconnected node after the insertion (shown as the initial node in the last subfigure in Figure 3)?

2. Can the author explain why not evaluate the proposed L2Segment on another two problems, VRPB and 1-VRPPD, as the FSTA is already theoretically verified on these two problems?

3. Please also see the weaknesses and clarify them.

---

> ### Author Response · Authors · 2025-11-25
> **Response to Reviewer SSqq (1/3)**
>
> Thank you for your support in accepting our paper and the insightful comments! We greatly appreciate your recognition of the novel FSTA framework, its strong theoretical foundations, and the comprehensive experimental validation across diverse settings. We hope our responses below may address any remaining concerns about our work.
>
> > **W1:** While ..., a more explicit discussion of the boundaries of FSTA/L2Seg's effectiveness would be beneficial. For example, under what conditions ... might the overhead of segmentation and aggregation outweigh the benefits?
>
> Thank you for this insightful question. We measured the overhead across CVRP and VRPTW instances ranging from 1k to 5k nodes in the Table below. Even with our most complex model (L2Seg-SYN), the overhead consistently remains **below 10%** of the total iteration time (ranging from 7.2% to 9.6%). This indicates that the overhead scales efficiently and predictably. Generally speaking, the overhead primarily stems from neural network inference, which is driven by two factors: input data size and network call frequency. Regarding the former, L2Seg employs a **Batched Sub-Route Processing** design. During embedding, we split problems into adjacent route-pairs and use batch processing. This avoids memory bottlenecks and ensures that inference time scales efficiently for large-scale cases. Regarding the latter, the frequency of calling L2Seg is a tunable hyperparameter.
>
> However, even though the overhead is manageable and adjustable, we emphasize that overhead may be viewed for reference rather than as a pure performance indicator. For example, while L2Seg-SYN incurs higher overhead (7.2%-9.6%) than L2Seg-NAR (4.2%-6.5%), it achieves superior solution quality: the better neural guidance outweighs the additional computational cost. In another example, for a larger-scale problem, the overhead increases due to larger input data. However, the absolute benefits also tend to increase with problem size, since larger problems contain more redundant computations that can be eliminated. This creates a scaling dynamic: even though both overhead and benefits increase with problem scale, the speedup ratios may remain consistent or even improve for larger instances (as shown in our 2-7x speedups across different scales). As a result, larger overheads may not be an indicator of the performance boundary of L2Seg.
>
> |  | CVRP1k |  |  | CVRP2k |  |  | CVRP5k |  |  |
> | -- | -- | -- | --| -- | -- | -- | -- | -- | -- |
> |  | Avg L2Seg Time per Iteration | Avg Total Time per Iteration | Overhead rate | Avg L2Seg Time per Iteration | Avg Total Time per Iteration | Overhead rate | Avg L2Seg Time per Iteration | Avg Total Time per Iteration | Overhead rate |
> | L2Seg-NAR-LNS | 0.38s | 8.4s | 4.5% | 0.62s | 11.4s | 5.4% | 0.80s | 12.9s | 6.2% |
> | L2Seg-AR-LNS | 0.63s | 9.8s | 6.4% | 0.90s | 10.8s | 8.3% | 1.24s | 13.9s | 8.9% |
> | L2Seg-SYN-LNS | 0.76s | 10.5s | 7.2% | 1.14s | 12.5s | 9.1% | 1.33s | 14.1s | 9.4% |
> |  | **VRPTW1k** ||| **VRPTW2k** ||| **VRPTW5k** ||  |
> |  | Avg L2Seg Time per Iteration | Avg Total Time per Iteration | Overhead rate | Avg L2Seg Time per Iteration | Avg Total Time per Iteration | Overhead rate | Avg L2Seg Time per Iteration | Avg Total Time per Iteration | Overhead rate |
> | L2Seg-NAR-LNS | 0.37s | 8.9s | 4.2% | 0.61s | 10.4s | 5.9% | 0.83s | 12.8s | 6.5% |
> | L2Seg-AR-LNS | 0.62s | 9.1s | 6.8% | 0.96s | 11.9s | 8.1% | 1.14s | 12.4s | 9.2% |
> | L2Seg-SYN-LNS | 0.76s | 10.1s | 7.5% | 0.97s | 10.8s | 9.0% | 1.36s | 14.2s | 9.6% |
>
> We then discuss the boundaries of FSTA/L2Seg's effectiveness. L2Seg is particularly well-suited for problems with specific characteristics as follows:
>
> 1. First, it excels on large-scale instances where the overhead of neural network prediction is amortized over substantial computational savings from solving reduced problems.
> 2. Second, problems with longer subroutes (higher vehicle capacity) benefit more, as they provide more opportunities for stable segment identification. For example, when routes contain 50+ customers, the interior segments have higher stability compared to problems with only 10-15 customers per route.
> 3. Third, as mentioned by the reviewer, the relative benefit decreases as the overhead approaches the savings; however, based on our discussions and results above, this boundary is quite high, with benefits persisting even up to at least 10% overhead rates.
>
> We acknowledge scenarios where L2Seg may be less effective: extremely tight capacity constraints yielding small subroutes (5-10 customers each) limit stable segment identification, and very small problems (<500 nodes) may not justify the ~1s neural network overhead. However, L2Seg demonstrates robust performance across the broad range of practical logistics applications, where problems with 1k-5k nodes across different VRP variants. Our consistent 2-7x speedups with <10% overhead confirm that the effective operating range covers most real-world routing scenarios.

---

> ### Author Response · Authors · 2025-11-25
> **Response to Reviewer SSqq (2/3)**
>
> > **W2:** It is unclear that the boundary of the acceleration, as I notice that the HGS and LKH3 only run for a short time (5m and 10m). If the solving time extends, how will the acceleration benefit change in terms of both solution quality and time?
> >
>
> Thank you for raising this concern.
>
> **Benchmark Settings**: We did not cherry-pick time limits but followed established protocols from prior work. For VRPTW, we used similar settings from L2D [1], which employed time limits of roughly 5mins (ranging from 76s to 6.8 minutes), and for Table 2 comparisons, we followed SIL [2] settings to ensure fair comparisons with existing literature. Additionally, our iterative solving curves for CVRP (Figure 5) demonstrate consistent improvements (up to 7x acceleration), providing further evidence that our benefits are not dependent on specific time limit choices.
>
> **Extended Runtime Analysis**: We also conducted additional experiments extending the runtime from 5 minutes to 1 hour on the first 25 instances (limited by computational resources and time), comparing our best model, L2Seg-SYN-L2D, against HGS:
>
> | Method | CVRP1k |  |  | CVRP2k |  |  | CVRP5k |  |  |
> | --- | --- | --- | --- | --- | --- | --- | --- | --- | --- |
> |  | Obj. | Gap | Time | Obj. | Gap | Time | Obj. | Gap | Time |
> | HGS | 40.05 | 0.00% | 1h | 54.62 | 0.00% | 1h | 121.94 | 0.00% | 1h |
> | L2Seg-SYN-L2D (short) | 41.23 | 2.95% | 2.5m | 56.05 | 2.62% | 4.1m | 121.87 | -0.06% | 5.1m |
> | L2Seg-SYN-L2D (long) | 40.35 | 0.75% | 1h | 54.23 | -0.71% | 1h | 120.24 | -1.39% | 1h |
>
> **Key Findings**: While longer runtimes increase performance gaps, the absolute improvements remain meaningful. Importantly, our method continues to surpass HGS on larger instances, demonstrating consistent benefits across different computational settings.
>
> > **W3:** Some works are also based on hypergraphs, but this paper does not discuss.
> >
> > 1. Fu et al. A hierarchical destroy and repair approach for solving very large-scale traveling salesman problem.
> >
> > 2. Li et al. Destroy and Repair Using Hyper-Graphs for Routing.
> >
>
> Thank you for this valuable feedback. These two papers are indeed relevant and should have been discussed in our related work section. We highlight what distinguishes L2Seg from these important works.
>
> - **Broader Applicability:** L2Seg provides a unified framework across multiple VRP variants (CVRP, VRPTW, VRPB, 1-VRPPD) with theoretical guarantees, while existing hypergraph methods (Fu et al. and Li et al.) focus primarily on TSP or basic CVRP.
> - **Novel AR-NAR Synergy:** We pioneer the first joint decision-making between autoregressive and non-autoregressive models in NCO, enabling both global understanding and local precision in stability detection.
> - **Learning-Guided Decomposition:** Unlike fixed heuristic rules (Fu et al. and Li et al.), L2Seg learns problem-specific decomposition patterns from data, adapting to different problem structures and solver behaviors.
> - **Theoretical Foundation:** We provide formal feasibility and monotonicity theorems for hypergraph aggregation across multiple variants, establishing rigorous theoretical grounding.
>
> We then give detailed comparisons between L2Seg and the two papers below:
>
> Fu et al. (2023) introduce HDR, a hierarchical destroy-and-repair algorithm that recursively compresses TSP instances to handle problems with millions of cities. While HDR achieves remarkable scalability on very large TSP instances using non-learning heuristics, our approach differs by employing learned policies to identify unstable edges and extending beyond TSP to handle CVRP, VRPTW, and other variants. HDR uses straightforward edge-fixing based on historical local optima, whereas we learn destruction patterns from the lookahead heuristics.
>
> Li et al. (2025) propose DRHG, which uses hypergraphs to reduce consecutive edges and supervised learning for reconstruction. Their approach applies heuristic clustering for destruction, followed by ML-based repair of the destroyed segments. Our method takes the opposite approach: we use machine learning to identify unstable edges that should be destroyed, then employ efficient subsolvers for reconstruction. This reversed strategy allows us to leverage learned patterns for the critical decision of what to destroy while using proven optimization techniques for repair. While DRHG demonstrates strong results on TSP and CVRP, our experiments extend to more constrained variants like VRPTW.
>
> **We have updated the above discussions in the appendix (highlighted in red)** to properly acknowledge these important advances and clarify our positioning relative to them.

---

> > ### Author Response · Authors · 2025-11-25
> > **Response to Reviewer SSqq (3/3)**
> >
> > > **Q1:** Can the author explain how to deal with the disconnected node after the insertion (shown as the initial node in the last subfigure in Figure 3)?
> > >
> >
> > Thank you for raising this concern. We thank the reviewer for allowing us to clarify this mechanism. We emphasize that the terms "insertion" and "deletion" in our AR decoding process refer to **modeling dependencies** between predictions, which is a pseudo-process, not to immediate topological modifications of the graph. The process works as follows:
> >
> > The AR decoder sequentially identifies endpoints of unstable edges. The initial node serves as the starting point for detecting the first unstable edge. In the **deletion stage**, the model identifies an unstable edge connected to the current node by selecting its other endpoint. Importantly, this edge is only marked as unstable for FSTA, not immediately removed from the solution.
> >
> > In the **insertion stage**, the model identifies the starting point of the next unstable edge to detect. This "insertion" simply bridges consecutive unstable edge detections: it establishes the connection between the current unstable edge's endpoint and where to begin searching for the next unstable edge. No edges are actually added to the solution during this stage.
> >
> > Therefore, the initial node never becomes disconnected. The edge connecting to it is merely marked as unstable for subsequent FSTA processing. The alternating deletion-insertion sequence guides the model to systematically identify all unstable edges. We have revised the paper to highlight our presentation accordingly. Thank you for pointing this out!
> >
> > ---
> >
> > > **Q2:** Can the author explain why not evaluate the proposed L2Segment on another two problems, VRPB and 1-VRPPD, as the FSTA is already theoretically verified on these two problems?
> > >
> >
> > Thank you for your comment. The primary goal of providing the theoretical proof for FSTA on VRPB and 1-VRPPD was to demonstrate that our framework can be broadly applied to routing problems with various constraint types, showcasing its theoretical generality beyond just CVRP and VRPTW. We believe our empirical results on CVRP and VRPTW, the two most widely studied VRP variants, already demonstrate the broad applicability of L2Seg and FSTA through extensive experiments across diverse settings (multiple scales, capacity configurations, customer distributions, and realistic datasets). While we have established the theoretical foundations for other routing variants (as detailed in Sections B.1.5 and B.2), implementing and evaluating L2Seg on VRPB and 1-VRPPD would require additional engineering effort. We thus leave them as valuable future work. On the other hand, we provide comprehensive implementation details, including data collection procedures (Section C.4), segment aggregation strategies (Table 6), neural network architectures (Sections 4.1 and D.3), and training/testing protocols, which should enable researchers to implement FSTA on other VRP variants following our guidelines.
> >
> > ---
> >
> > We thank the reviewer again for the insightful comments in shaping our paper!
> >
> > ---
> >
> > **Reference**
> >
> > [1] Li et al. "Learning to delegate for large-scale vehicle routing." *Neurips 2021*
> >
> > [2] Luo et al. “Self-improved learning for scalable neural combinatorial optimization.” *arXiv preprint arXiv:2403.19561, 2024.*

---

> > > ### Comment · Reviewer_SSqq · 2025-11-26
> > >
> > > I thank the authors for their detailed response and the efforts made to improve the manuscript. All of my concerns have been addressed. Given the improvements to the manuscript and the demonstrated potential of L2Seg in Neural Combinatorial Optimization, I am pleased to update my score to 8.

---

> > > > ### Author Response · Authors · 2025-11-27
> > > > **Thank You for Your Support!**
> > > >
> > > > We thank the reviewers for recognizing our responses and raising your scores. We appreciate your insightful questions and are glad that the additional clarifications and experiments helped address your concerns. We will carefully incorporate all your valuable suggestions into our final version. Thank you again for your constructive feedback and support!

---

### Comment · Area_Chair_yq3K · 2025-11-25
**Discussion Period**

The discussion period is now open. Please use the “Official Comments” to engage in discussions about each other's reviews and the authors' rebuttal, and update your assessments or comments as appropriate.

Did the authors' rebuttal adequately address your concerns? We kindly ask that you update your reviews based on these discussions and your evaluation of the rebuttal, even if your overall assessment remains unchanged.

Thank you all for your contributions.

Best regards, AC

---

### Meta-Review · Area_Chair_oJ45 · 2025-12-30

**Summary:**

This work proposes L2Seg, a learning-to-segment framework to tackle vehicle routing problems. Based on the observation that a large portion of the solution will remain stable (unchanged) during the iterative solution search process, L2Seg develops a learning-based First-Segment-Then-Aggregate (FSTA) reduction strategy, which identifies and preserves stable solution segments while dynamically focusing search on unstable parts. In this way, it can significantly accelerate the iterative solvers for large-scale VRPs. L2Seg includes non-autoregressive, autoregressive, and synergistic variants, which can be integrated into various solvers (e.g., LKH-3, LNS, L2D) to tackle different VRP variants with promising results.

The reviewers initially had mixed scores (6,6,4,4) and raised concerns regarding the conceptual novelty, effectiveness, the complexity and plug-and-play nature of L2Seg, the depth of discussion and comparison with related works, the reliability of stability labels, generalization ability, and theoretical contribution. After the rebuttal, most of these concerns have been properly addressed. Based on the author-reviewer discussions:
1) Reviewer SSqq increased their score from 6 to 8.
2) Reviewer Kw4B raised their score from 4 to 6.
3) In both the official comments and a confidential message to the AC, Reviewer L8xN indicated their intention to update their score. Given that their concerns were largely resolved, I believe the updated score could be 6 or 8.
4) I also consider that the rebuttal has satisfactorily addressed the concerns of Reviewer b73r, whose score could therefore be raised from 6 to 8.

As a result, the final score for this work could be either (8,8,8,6) or (8,8,6,6).

I read this paper in detail and agree with the reviewers that this work is generally well-written and easy to follow, the proposed FSTA method is novel and technically sound with good flexibility, and the experimental results are promising. In addition, as the AC for a previous review round of this work, I have observed its significant improvements after two rounds of constructive and very productive rebuttals.

Therefore, I recommend a clear acceptance of this paper and, if capacity permits, nominating it for an oral presentation.

**Reviewer Concerns:**

I believe most concerns have been properly addressed by the rebuttal.

As a minor note, while a comparison with Large Neighborhood Search (LNS) is available in Appendix C.1, the responses regarding L2Seg's relation to LNS (W1 from Reviewer Kw4B) and the search-space restriction aspect (W2 from Reviewer L8xN) could be incorporated into the revised main paper if space permits.

**Reviewer Scores:**

According to the author-reviewer discussion, if the reviewers had been able to participate fully in the discussion,  I believe:
1) Reviewer SSqq increased their score from 6 to 8.
2) Reviewer Kw4B raised their score from 4 to 6.
3) In both the official comments and a confidential message to the AC, Reviewer L8xN indicated their intention to update their score. Given that their concerns were largely resolved, I believe the updated score could be 6 or 8.
4) I also consider that the rebuttal has satisfactorily addressed the concerns of Reviewer b73r, whose score could therefore be raised from 6 to 8.

As a result, the final score for this work could be either (8,8,8,6) or (8,8,6,6).

---

### Decision · Program_Chairs · 2026-01-26

Accept (Oral)